# ACTIVE LEARNING OF DEEP NEURAL NETWORKS VIA GRADIENT-FREE CUTTING PLANES

## ABSTRACT

Active learning methods aim to improve sample complexity in machine learning. In this work, we investigate an active learning scheme via a novel gradient-free cutting-plane training method for ReLU networks of arbitrary depth. We demonstrate, for the first time, that cutting-plane algorithms, traditionally used in linear models, can be extended to deep neural networks despite their nonconvexity and nonlinear decision boundaries. Our results demonstrate that these methods provide a promising alternative to the commonly employed gradient-based optimization techniques in large-scale neural networks. Moreover, this training method induces the first deep active learning scheme known to achieve convergence guarantees. We exemplify the effectiveness of our proposed active learning method against popular deep active learning baselines via both synthetic data experiments and sentimental classification task on real datasets.

## 1 INTRODUCTION

Large neural network models are now core to artificial intelligence systems. After years of development, current large NN training is still dominated by gradient-based methods, which range from basic gradient descent method to more advanced online stochastic methods such as Adam (Kingma & Ba, 2017) and AdamW (Loshchilov & Hutter, 2019). Recent empirical effort has focused on cutting down storage requirement for such optimizers, see (Griewank & Walther, 2000; Zhao et al., 2024); accelerating convergence by adding momentum, see (Xie et al., 2023); designing better step size search algorithms, see (Defazio & Mishchenko, 2023). Despite its popularity, gradient-based methods suffer from sensitivity to hyperparameters and slow convergence. Therefore, researchers are persistently seeking for alternative training schemes for large NN models, including involve zero-order and second-order algorithms.

Cutting-plane method is a classic optimization algorithm and is known for its fast convergence rate. Research on cutting-plane type methods dates back to 1950s when Ralph Gomory (Gomory, 1958) first studied it for integer programming and mixed-integer programming problems. Since then, this method has also been heavily investigated for solving nonlinear problems. Different variations of cutting-plane method emerge, including but not limited to center of gravity cutting-plane method, maximum volume ellipsoid cutting-plane method, and analytic center cutting-plane method, which mainly differ in their center-finding strategy.

Historically, deep NN training and cutting plane methods have developed independently, each with its own audience. In this work, we bridge them for the first time by providing a viable cutting-plane-based deep NN training and active learning scheme. Our method finds optimal neural network weights and actively queries additional training points via a gradient-free cutting plane approach. We show that an active learning scheme based on our newly proposed cutting-plane-based strategy naturally inherits classic convergence guarantees of cutting-plane methods. We present synthetic and real data experiments to demonstrate the effectiveness of our proposed methods.

## 2 NOTATION

We denote the set of integers from 1 to $n$ as $[n]$. We use $\mathcal{B}_p := \{u \in \mathbb{R}^d : \|u\|_p \leq 1\}$ to denote the unit $\ell_p$-norm ball and $\langle \cdot, \cdot \rangle$ to denote the dot product between two vectors. Given a

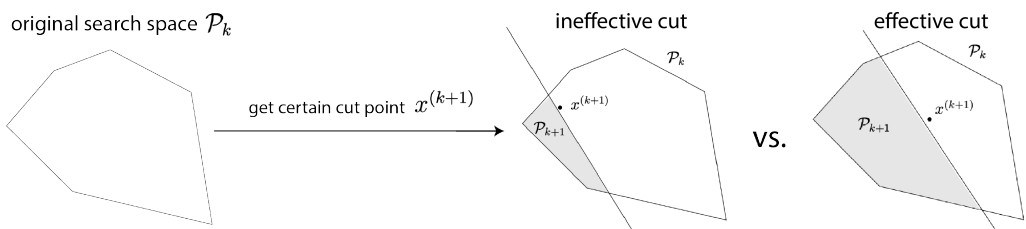

Figure 1: Illustration of a single iteration of general cutting-plane method (Boyd & Vandenberghe, 2007).

hyperplane $\mathcal{H} := \{x : x^T w = 0\}$, we use $\mathcal{H}^+$ ($\mathcal{H}^-$) to denote the *positive* (*negative*) half-space: $\mathcal{H}^{+(-)} := \{x : x^T w \geq (\leq)0\}$. For ReLU, we use notation $(x)_+ = \max(x, 0)$. We take $\mathbb{1}\{x \in S\}$ as the $1/0$-valued indicator function with respect to the set $S$, evaluated at $x$.

## 3 PRELIMINARIES

Classic cutting-plane method's usage in different optimization problems has been heavily studied. To better demonstrate the problem and offer a more self-contained background, we start with describing basic cutting-plane method's workflow below.

**Cutting-Plane Method.** Consider any minimization problem with an objective function $f(\theta)$, where the solution set, denoted as $\Theta$, is a convex set. Cutting-plane method typically assumes the existence of an oracle that, given any input $\theta_0$, either confirms that $\theta_0 \in \Theta$, thereby terminating with $\theta_0$ as a satisfactory solution, or returns a pair $(x, y)$ such that $x^T \theta_0 \leq y$ while $x^T \theta > y$ for all $\theta \in \Theta$. If the cut is "good enough," it allows for the elimination of a large portion of the search space, enabling rapid progress toward the true solution set $\Theta$. The classic convergence results of the cutting-plane method are highly dependent on the quality of the cut in each iteration. For instance, if the center of gravity of the current volume is removed, it guarantees a volume reduction of approximately 63%. Similar results hold for the analytic center and the center of the maximum volume ellipsoid. Figure 1 illustrates a single step of the cutting-plane method, showing how a "good" cut near the center of the current volume induces a much larger volume reduction compared to a "bad" cut near the edge.

**Cutting-Plane-Based Active Learning (AL) with Linear Models (Louche & Ralaivola, 2015).** Cutting-plane method provides a natural active learning framework to localize a set of deep NN classifiers with certain classification margins. Prior work (Louche & Ralaivola, 2015) has studied the use of cutting-plane method in the context of active learning with linear models for binary classification. The setup is the following. One is given a set of unlabeled data $\{x_1, \cdots, x_n\} \in \mathcal{D}$. The authors consider a linear binary classifier $f(x; \theta) := \langle \theta, x \rangle$ with prediction $\text{sign}(f(x; \theta))$ for any input $x$. Define the set of model parameters that correctly classify our dataset as $\mathcal{T}_\mathcal{D}$, which is a set of linear inequalities. The size of parameter set reflects the level of uncertainty in the classifier. Starting at an initial set $\mathcal{T}^0$, the goal is to query additional data points and acquire their labels to reduce the size of $\mathcal{T}^0$ to approach $\mathcal{T}_\mathcal{D}$, which hopefully would have high test accuracy when the generalization error is small. The cutting-plane-based active learning framework developed by Louche & Ralaivola (2015) (Algorithm 3) starts with a localized convex set $\mathcal{T}^0$. The algorithm is presented with a set of unlabeled data. At each step $t$, it performs the following steps: (i) computes the center of the current parameter space $\theta_c^t := \text{center}(\mathcal{T}^t)$; (ii) queries the label $y_t$ for $x_t$ from the unlabeled dataset $\mathcal{D}$ which has minimal prediction margin with respect to $\theta_c^t$; (iii) reduces the parameter space via a cutting-plane in the case of mis-classification: $\mathcal{T}^{t+1} = \mathcal{T}^t \cap \{\theta \mid y_t \langle x_t, \theta \rangle > 0\}$. The algorithm terminates when set $\mathcal{T}^t$ is small enough or maximum iterations or data budget have been reached. This active learning scheme has strong convergence result inherited from classic cutting-plane method, which has been investigated in (Louche & Ralaivola, 2015).

**Disentangling Model Training and Active Learning.** Although Louche & Ralaivola (2015) primarily focuses on the active learning setting, their method implicitly suggests a cutting-plane-based training workflow for linear binary classifiers. To illustrate this, consider a set of training samples $\{(x_1, y_1), \ldots, (x_n, y_n)\}$. Each pair $(x_i, y_i)$ induces a cut on the parameter space $\{\theta \mid y_i \langle \theta, x_i \rangle > 0\}$.

The final center, i.e., $\theta_c = \text{center}(\{\theta \mid y_i\langle\theta, x_i\rangle > 0, i \in [n]\})$, accounts for all such cuts while maintaining the desired property $\text{sign}(f(x; \theta)) = y$ for all training samples. This choice makes $\theta_c$ not only an optimal solution, but also a robust choice, as it remains stable under data perturbations.

By disentangling the model training process from the active learning query strategy described in (Louche & Ralaivola, 2015), we derive a gradient-free cutting-plane-based workflow for training linear binary classifiers. However, this approach has several key limitations: (1) it is restricted to linear models, (2) it requires the data to be linearly separable to ensure an optimal parameter set, and (3) it only supports binary classification tasks. Our current work addresses all three limitations by extending the cutting-plane method to train deep nonlinear neural networks for both classification and regression tasks, without requiring linear separability of the data. Similar to the linear binary classification case, our vanilla training scheme lacks desirable convergence guarantees, as the cuts may occur at the edge of the parameter set. To overcome this, we focus on a cutting-plane-based active learning scheme that enables cuts to be near the center of the parameter set. We provide convergence results for this approach, which, to the best of our knowledge, is the first convergence guarantee for active learning algorithms applied to deep neural networks.

**Outline.** The paper is organized as follows: in Section 4, we adapt the cutting-plane method for nonlinear model training by transforming the nonlinear training process into a linear programming problem. Section 5 introduces the general gradient-free cutting-plane-based training algorithm for deep NNs. In Section 6, we explore the resulting cutting-plane AL framework and prove its convergence. Section 7 demonstrates the practical effectiveness of our proposed training and active learning methods through extensive experiments.

# 4 KEY OBSERVATION: TRAINING ReLU NNS FOR BINARY CLASSIFICATION IS LINEAR PROGRAMMING

The cutting-plane AL scheme proposed by Louche & Ralaivola (2015) (summarized in Section 3) is designed for linear models like $f(x; \theta) = \langle\theta, x\rangle$. Extending this method to nonlinear models, such as a two-layer ReLU network $f(x; \theta) = (x^T\theta_1)_+\theta_2$, with $\theta_1 \in \mathbb{R}^{d\times m}$ and $\theta_2 \in \mathbb{R}^m$, presents additional challenges. Specifically, for a mispredicted data pair $(x_i, y_i)$, determining how to cut the parameter space $(\theta_1, \theta_2)$ is far more complex, whereas in the linear case, the cut is simply $y_i\langle\theta, x_i\rangle > 0$.

To break this bottleneck and extend the cutting-plane-based learning method to nonlinear models, we observe that training a ReLU network for binary classification can be formulated as a linear programming problem. This insight is crucial for extending the learning scheme in (Louche & Ralaivola, 2015) to more complex models. We now develop our core idea of reframing binary classification with ReLU models as linear programs. For clarity, we present our results in two theorems: one for two-layer ReLU networks and another for ReLU networks of arbitrary depth. We focus on the two-layer case in the main paper for detailed discussion, deferring the more abstract general case to Appendix F.2. Since the general case is an extension of the two-layer model, focusing on the two-layer case should provide a clearer understanding of the core concepts.

We start with writing the linear program corresponding to the linear model for binary classification tasks as below,

$$\begin{aligned} \text{find} \quad & \theta \\ \text{s.t.} \quad & y_i\langle\theta, x_i\rangle \geq 1 \quad \forall i. \end{aligned} \tag{1}$$

Note that intuitively, we want $y_i\langle\theta, x_i\rangle > 0$ to be satisfied for our sign prediction. However, the set $\{\theta | y_i\langle\theta, x_i\rangle > 0\}$ is not compact and is thus not compliant with forms of standard linear programs. This may raise technical issues. We observe that our training data is finite, and thus we can always scale $\theta$ to achieve $y_i\langle\theta, x_i\rangle \geq c$ for any positive constant $c$ once $y_i\langle\theta, x_i\rangle > 0$ holds, and $y_i\langle\theta, x_i\rangle \geq c$ for $c > 0$ also guarantees $y_i\langle\theta, x_i\rangle > 0$. We pick $c = 1$ in (1). With a two-layer ReLU model, we obtain the following problem:

$$\begin{aligned} \text{find} \quad & W_1, W_2 \\ \text{s.t.} \quad & y_i(x_i^T W_1)_+ W_2 \geq 1 \quad \forall i. \end{aligned} \tag{2}$$

Before showing that solving (2) is indeed equivalent to solving a linear program, we first introduce the core concept of activation patterns which we will draw on heavily later. For data matrix $X \in$

$\mathbb{R}^{n \times d}$ and any arbitrary vector $u \in \mathbb{R}^d$, we consider the set of diagonal matrices

$$\mathcal{D} := \{\text{diag}(\mathbb{1}\{Xu \geq 0\})\}.$$

We denote the carnality of set $\mathcal{D}$ as $P$, i.e., $P = |\mathcal{D}|$. Thus $\{D_i \in \mathcal{D}, i \in [P]\}$ iterates over all possible activation patterns of ReLU function induced by data matrix $X$. See Definition 3 for more details. With this concept of activation patterns, we can reframe the training of two-layer ReLU model for binary classification as the following linear program:

**Theorem 4.1.** *When $m \geq 2P$, Problem (2) is equivalent to*

$$\text{find} \quad u_i, u_i'$$

$$\text{s.t.} \quad y\left(\sum_{i=1}^{P} D_i X(u_i - u_i')\right) \geq 1, \tag{3}$$

$$(2D_i - I)Xu_i \geq 0,$$

$$(2D_i - I)Xu_i' \geq 0.$$

*Proof.* See Appendix F.1. $\qquad\square$

The high-level rationality behind Theorem 4.1 is the observation that

$$(XW_1)_+ W_2 = \sum_{i=1}^{m} (XW_{1i})_+ W_{2i} = \sum_{i=1}^{m} \text{diag}(\mathbb{1}\{XW_{1i} \geq 0\})XW_{1i}W_{2i}. \tag{4}$$

By defining $K_i := \text{diag}(\mathbb{1}\{XW_{1i} \geq 0\})$ and a set of $\{v_i, v_i'\}$ vectors by setting $v_i := W_{1i}W_{2i}$ when $W_{2i} \geq 0$ and 0 otherwise, $v_i' := -W_{1i}W_{2i}$ when $W_{2i} < 0$ and 0 otherwise, we have $\text{diag}(\mathbb{1}\{XW_{1i} \geq 0\})XW_{1i}W_{2i} = K_i X(v_i - v_i')$. The expression in (4) thus writes

$$(XW_1)_+ W_2 = \sum_{i=1}^{m} K_i X(v_i - v_i') \text{ where } (2K_i - I)Xv_i \geq 0, (2K_i - I)Xv_i' \geq 0.$$

Therefore, if we iterate over all $D_i \in \mathcal{D}$, we are guaranteed to reach each $K_i$. The equivalence in Theorem 4.1 holds in the sense that, which we prove rigorously in Appendix F.1, whenever there is solution $W_1, W_2$ to Problem (2), there is always solution $\{u_i, u_i'\}$ to problem (3) and vice versa. Moreover, when an optimal solution $\{u_i^\star, u_i'^\star\}$ to problem (3) has been found, we can explicitly create an optimal solution $\{W_1^\star, W_2^\star\}$ to Problem (2) from value of $\{u_i^\star, u_i'^\star\}$, see Appendix F.1 for details. Thereafter, for any test point $\tilde{x}$, our sign prediction is simply $\text{sign}((\tilde{x}W_1^\star)_+ W_2^\star)$, which will have the same value as $\text{sign}(\sum_i (\tilde{x}^T u_i^\star)_+ - (\tilde{x}^T u_i'^\star)_+)$.

We emphasize that our reframing of training a ReLU network for binary classification as linear programming does not eliminate the nonlinearity of the ReLU activation. Instead, this approach works because the ReLU activation patterns for a given training dataset are finite. By looping through these activation patterns, we can explicitly enumerate them. At test time, the ReLU nonlinearity is preserved, as our prediction $\text{sign}(\sum_i (\tilde{x}^T u_i)_+ - (\tilde{x}^T u_i')_+)$ depends on the sign of $\tilde{x}^T u_i$ and $\tilde{x}^T u_i'$, ensuring that the expressiveness of the nonlinearity remains intact. A careful reader might note that the number of patterns $P$ increases with the size of the training data, meaning the number of variables in Problem (3) may also grow. Additionally, finding all activation patterns poses a challenge. In Section 7, we demonstrate that subsampling a set of non-duplicate activation patterns performs well in practice. For further grounding, Appendix F.4 outlines an iterative hyperplane filtering method that guarantees the identification of all activation patterns with a reasonable complexity bound.

Now, let us consider ReLU network with $n$ hidden layer for binary classification task

$$\text{find} \quad W_1, W_2, \cdots, W_{n+1}$$

$$\text{s.t.} \quad y \odot ((\cdots (((XW_1)_+ W_2)_+ W_3)_+ W_4 \cdots)_+ W_n)_+ W_{n+1} \geq 1. \tag{5}$$

We extend the activation patterns involved in Theorem 4.1 to $(n+1)$-layer neural networks. Let $W_1 \in \mathbb{R}^{d \times m_1}, W_2 \in \mathbb{R}^{m_1 \times m_2}, W_3 \in \mathbb{R}^{m_2 \times m_3}, \cdots, W_n \in \mathbb{R}^{m_{n-1} \times m_n}, W_{n+1} \in \mathbb{R}^{m_n}$. Define $m_0 := d$ and the activation pattern in $i$-th layer as

$$\mathcal{D}^{(i)} := \left\{\text{diag}(\mathbb{1}\{(\cdots((Xv_1)_+ v_2)_+ v_3 \cdots)_+ v_i \geq 0\}) \middle| v_j \in \mathbb{R}^{m_{j-1} \times m_j} \forall j < i, v_i \in \mathbb{R}^{m_{i-1}}\right\}.$$

We denote the cardinality of set $\mathcal{D}^{(i)}$ as $P_i$, i.e., $P_i = |\mathcal{D}^{(i)}|$. Thus $\{D_j^{(i)} \in \mathcal{D}^{(i)}, j \in [P_i]\}$ iterates over all possible activation patterns at the $i$-th hidden layer. We then reframe Problem (5) as below:

**Theorem 4.2.** *When $m_i \geq \Pi_i^n 2P_i$ for each $i \in [n]$, Problem (5) is equivalent to*

$$
\textit{find} \qquad u_{j_n j_{n-1} j_{n-2} \ldots j_1}^{c_n c_{n-1} c_{n-2} \cdots c_1}
$$

$$
\textit{s.t.} \quad y \odot \sum_{j_n=1}^{P_n} D_{j_n}^{(n)} \left( \mathcal{T}_1^{(n-1)}(D^{(n-1)}) - \mathcal{T}_2^{(n-1)}(D^{(n-1)}) \right) \geq 1 \tag{6}
$$

$$
(2D_{j_i}^{(i)} - I)\mathcal{T}_{c_{n-1} c_{n-2} \cdots c_{i-1}}^{(n-1)(n-2)\cdots(i-1)}(D^{(i-1)}) \geq 0, 2 \leq i \leq n
$$

$$
(2D_{j_1}^{(1)} - I)X u_{j_n j_{n-1} \cdots j_1}^{c_n c_{n-1} \cdots c_1} \geq 0,
$$

*where $c_i \in \{1, 2\}$ and*

$$
\mathcal{T}_{c_{n-1} \cdots c_i}^{(n-1)\cdots(i)}(D^{(i)}) = \sum_{j_i=1}^{P_i} D_{j_i}^{(i)} \left( \mathcal{T}_{c_{n-1} \cdots c_i 1}^{(n-1)\cdots(i)(i-1)}(D^{(i-1)}) - \mathcal{T}_{c_{n-1} \cdots c_i 2}^{(n-1)\cdots(i)(i-1)}(D^{(i-1)}) \right), \forall\, i \leq n-1,
$$

$$
\mathcal{T}_{c_{n-1} c_{n-2} \cdots c_1}^{(n-1)(n-2)\cdots(1)}(D^{(1)}) = \sum_{j_1=1}^{P_1} D_{j_1}^{(1)} X \left( u_{j_n j_{n-1} \cdots j_1}^{1 c_{n-1} c_{n-2} \cdots c_1} - u_{j_n j_{n-1} \cdots j_1}^{2 c_{n-1} c_{n-2} \cdots c_1} \right).
$$

*Proof.* See Appendix F.2. $\qquad\square$

## 5 TRAINING DEEP NEURAL NETWORKS VIA CUTTING-PLANES

---
**Algorithm 1** Training NN with Cutting Plane Method

---
**Input:** $\epsilon_v, T_{max}$
**Initialization:** $\mathcal{T}^0 \leftarrow \mathcal{B}_2, t \leftarrow 0$
**repeat**
    $\theta_c^t \leftarrow \textbf{center}(\mathcal{T}^t)$
    Get new training data $(x_{n_t}, y_{n_t})$
    **if** $y_{n_t} f(x_{n_t}, \theta_c^t) < 0$ **then**
        $\mathcal{T}^{t+1} \leftarrow \mathcal{T}^t \cap \textbf{cut}(x_{n_t}, z)$
        $t \leftarrow t + 1$
    **end if**
**until** $\textbf{vol}(\mathcal{T}^t) \leq \epsilon_v$ or $t \geq T_{max}$
**return** $\theta_c^t$

---

With the linear program reframing of training deep ReLU models in place, we now formally introduce our cutting-plane-based NN training scheme for binary classification. We begin with a feasible set of variables in our linear program (6) that contains the optimal solution. For each training sample $(x_i, y_i)$, we add the corresponding constraints as cuts. At each iteration, we select the center of the current parameter set, stopping when either the validation loss stabilizes or after a fixed number of iterations, similar to the stopping criteria in gradient-based training of large models.

We present our cutting-plane-based NN training algorithm here in main text. For generalizations, such as (i) relaxing the data distribution to remove the linear separability requirement (Appendix E.1), and (ii) extending from classification to regression (Appendix E.2), we refer readers to Appendix E. We emphasize that both the relaxed data constraint and the ability to handle regression tasks are unique to our method and have not been achieved in prior work.

Algorithm 1 presents the general workflow of how we train deep NNs with gradient-free cutting-plane method, which simply adds a cut corresponding to each training data $(x_{n_t}, y_{n_t})$. When the stopping criterion is satisfied, center of current parameter set $\mathcal{T}^t$ is returned. Here we start with the parameter space as the unit 2-norm ball, which is guaranteed to contain optimal parameters due to scale invariance.

After we get the final $\theta_c^t$, for any test point $\tilde{x}$, we can directly compute our sign prediction with $\theta_c^t$. For example, for two-layer ReLU model, $\theta_c^t$ will be of form $\{u_i^t, u_i'^t\}$, our final sign prediction would simply be $\text{sign}(\sum_i (\tilde{x}^T u_i^t)_+ - (\tilde{x}^T u_i'^t)_+)$. For deeper models, the prediction is a bit more complex and is given in equation (17) in Appendix F.2. Notably, the computation of final sign prediction from

value of $\theta_c^t$ always takes a single step, just as one forward pass of original NN model formulation. Moreover, one can also restore optimal NN weights from final $\theta_c^t$, see Appendix F.1 for two-layer case reconstruction of optimal NN parameters and Appendix F.2 for general deep NN models.

Two key functions in our proposed training scheme are the "center" function and the "cut" function, which we detail below:

- *Center*. The "center" function calculates the center of the convex set $\mathcal{T}^t$. There are a couple of notions of centers, such as center of gravity (CG), center of maximum volume ellipsoid (MVE), Chebyshev's center, and analytic center (Boyd & Vandenberghe (2004)). Among these, the analytic center is the easiest to compute and is empirically known to be effective (Goffin et al. (1997), Atlason et al. (2008)). This is the notion of center that we will adopt to compute the query point in our algorithm. See Appendix H.1 for details.

- *Cut*. The "cut" function determines the cutting planes we get from a specific training data $(x_{n_t}, y_{n_t})$. For two-layer model, "cut" function would return the constraint set $\{y_{n_t}(\sum_i D_{in_t} x_{n_t}^T(u_i - u'_i)) \geq 1, (2D_{in_t} - 1)x_{n_t}^T u_i \geq 0, (2D_{in_t} - 1)x_{n_t}^T u'_i \geq 0\}$. For deeper NNs, "cut" function would return constraints listed in Problem (6).

Compared to gradient-based NN training scheme, we take a cutting-plane cut for each data point encountered while gradient-based method employs a gradient descent step corresponding to the data query. Moreover, our training scheme is guaranteed to correctly classify all data points we have ever encountered, while gradient-based method has no such guarantees.

## 6 CUTTING-PLANE-BASED ACTIVE LEARNING AND CONVERGENCE GUARANTEES

### 6.1 ALGORITHM II: CUTTING-PLANE LOCALIZATION FOR ACTIVE LEARNING

Our proposed cutting-plane-based active learning algorithm adapts and extends the generic framework discussed in Section 5 (Algorithm 3). For the sake of simplicity, we present in this section the algorithm specifically for binary classification and with respect to two-layer ReLU NNs. We emphasize that the algorithm can be easily adapted to the case of multi-class and regression, per discussions in Appendix E.2, and for deeper NNs following our reformulation in Theorem 4.2.

Recall the problem formulation for cutting-plane AL with two-layer ReLU NN for binary classification in Equation (3). Given a training dataset $\mathcal{D}$, we use $X_{\mathcal{D}}$ and $y_{\mathcal{D}}$ to denote the slices of $X$ and $y$ at indices $\mathcal{D}$. Moreover, we succinctly denote the prediction function as:

$$f^{\text{two-layer}}(X; \theta) := \sum_{i=1}^{P}(D(S_i)X)_{\mathcal{D}}(u'_i - u_i) = \begin{bmatrix} X_{\mathcal{D}}^1 & -X_{\mathcal{D}}^1 & \dots & X_{\mathcal{D}}^P & -X_{\mathcal{D}}^P \end{bmatrix} \theta, \quad (7)$$

where $\theta = (u'_1, u_1, \dots, u'_P, u_P)$ with $u_i, u'_i \in \mathbb{R}^d$, and $X_{\mathcal{D}}^i$ is a shorthand notation for $(D(S_i)X)_{\mathcal{D}}$. For the further brevity of notation, we denote the ReLU constraints in Equation 3, i.e. $((2D(S_i) - I_n)X)_{\mathcal{D}} u_i \geq 0$, $((2D(S_i) - I_n)X)_{\mathcal{D}} u'_i \geq 0$ for all $i$, as $C(\mathcal{D}), C'(\mathcal{D})$.

With Theorem 4.1 and the linearization of $f^{\text{two-layer}}(X; \theta)$, cutting-plane-based active learning methods become well applicable. As in Algorithm 3, we restrict the parameter space $\Theta$ to be within the unit 2-norm ball: $\mathcal{B}_2 := \{\theta \mid \|\theta\|_2 \leq 1\} = \{\theta \mid \sum_{i=1}^{P}(\|u_i\|_2^2 + \|u'_i\|_2^2) \leq 1\}$. For computing the center of the parameter space at each step for queries, we use the *analytic center* (Definition 1), which is known to be easily computable and has good convergence properties. We refer to Section 6.2 for a more detailed discussion.

**Definition 1** (Analytic center). *The analytic center of polyhedron $\mathcal{P} = \{z \mid a_i^T z \leq b_i, i = 1, ..., m\}$ is given by*

$$AC(\mathcal{P}) := \arg\min_{z} -\sum_{i=1}^{m} \log(b_i - a_i^T z) \quad (8)$$

We are now ready to present the cutting-plane-based active learning algorithm for deep NNs. For breadth of discussion, we present three versions of the active learning algorithms, each corresponding to the following setups:

1. *Cutting-plane AL with query synthesis (Algorithm 2).* The cutting-plane oracle gains access to a query synthesis. Therefore, the cut is always active until we have encountered the optimal classifier(s), at which point the algorithm terminates.

2. *Cutting-plane AL with limited queries (Algorithm 4).* The cutting-plane oracle has access to limited queries. The cut is only performed when the queried candidate mis-classifies the data pair returned by the oracle.

3. *Cutting-plane AL with inexact cuts (Algorithm 5).* The cutting-plane oracle has access to limited queries. However, the algorithm always performs the cut regardless of whether the queried candidate mis-classifies the data pair returned by the oracle.

---

**Algorithm 2** Cutting-plane AL for Binary Classification with Query Synthesis

---

1: $\mathcal{T}^0 \leftarrow \mathcal{B}_2$
2: $t \leftarrow 0$
3: $\mathcal{D}_{\text{AL}} \leftarrow \mathbf{0}$
4: **repeat**
5:     $\theta_c^t \leftarrow \text{center}(\mathcal{T}^t)$
6:     **for** $s$ in $\{1, -1\}$ **do**
7:         $(x_{n_t}, y_{n_t}) \leftarrow \text{QUERY}(\theta_c^t, s)$
8:         **if** $y_{n_t} \cdot f^{\text{two-layer}}(x_{n_t}; \theta_c^t) < 0$ **then**
9:             $\mathcal{D}_{\text{AL}} \leftarrow \text{ADD}(\mathcal{D}_{\text{AL}}, (x_{n_t}, y_{n_t}))$
10:            $\mathcal{T}^{t+1} \leftarrow \mathcal{T}^t \cap \{\theta : y_{n_t} \cdot f^{\text{two-layer}}(x_{n_t}; \theta) \geq 0, \mathcal{C}(\{n_t\}), \mathcal{C}'(\{n_t\})\}$
11:            $t \leftarrow t + 1$
12:         **end if**
13:     **end for**
14: **until** $|\mathcal{D}_{\text{AL}}| \geq n_{\text{budget}}$
15: **return** $\theta_c^t$

---

1: **function** QUERY$(\theta, s)$
2:     $(x, y) \leftarrow \arg\min_{(x_i, y_i) \in \mathcal{D}_{\text{QS}}} s f^{\text{two-layer}}(x_{n_t}; \theta)$
3:     **return** $(x, y)$
4: **end function**

---

For brevity, we present the algorithm for the first setup here and refer the rest to Appendix D. Algorithm 2 summarizes the proposed algorithm under the first setup, where we have used $\mathcal{D}_{\text{QS}}$ to denote the query synthesis. We note that Algorithm 4 for the second setup is obtained simply by changing $\mathcal{D}_{\text{QS}}$ to $\mathcal{D}_{\text{LQ}}$, which denotes limited query.

The general workflow of Algorithm 2 follows that of Algorithm 3 but on the transformed parameter space via the ReLU networks. We highlight here two key differences in our algorithm: (1) in each iteration, for faster empirical convergence, we query twice for the data point that is classified positively and the one that is classified negatively with highest confidence, respectively. If one or both of them turn out to be miss-classifications, this informs the active learner well and we expect a large cut. (2) For two- and three-layer ReLU networks, Ergen & Pilanci (2021b) demonstrate that these models can be reformulated as exact convex programs. This allows the option of incorporating a final convex solver into Algorithm 2, applied after the active learning loop with the data collected thus far. This convex reformulation includes regularization, which can improve the performance of our cutting-plane AL in certain tasks. For more details, see Appendix H.2.

## 6.2 Convergence Guarantees

We give theoretical examination of the convergence properties of Algorithm 2 and 4 with respect to both the center of gravity (cg) and the center of maximum volume ellipsoid (MVE). Analysis of Algorithm 5 for inexact cuts is given in Appendix G.2. As the analysis of MVE closely parallels that of CG, we refer readers to Appendix G.3 for a detailed discussion, in the interest of brevity. For both centers, we measure the convergence speed with respect to the volume of the localization set $\mathcal{T}^t$ and judge the progress in iteration $t$ by the fractional decrease in volume: $\mathbf{vol}(\mathcal{T}^{t+1})/\mathbf{vol}(\mathcal{T}^t)$.

To start, we give the definitions of the center of gravity (Boyd & Vandenberghe, 2004).

**Definition 2** (Center of gravity (CG)). *For a given convex body (i.e. a compact convex set with non-empty interior) $C \subseteq \mathbb{R}^d$, the centroid, or center of gravity of $C$, denoted $\theta_G(C)$, is given by*

$$\theta_G(C) = \frac{1}{\mathbf{vol}(C)} \int_{x \in C} x dx.$$

*Given convex set $C$, we use the abbreviated notation $\theta_G$.*

Our analysis on the center of gravity relies on an important proposition (Proposition 1) given by Grünbaum (1960), which guarantees that in each step of cutting via a hyperplane passing through the

centroid of the convex body, a fixed portion of the feasible set is eliminated. A recursive application of this proposition shows that after $t$ steps from the initial step, we obtain the following volume inequality: $\mathbf{vol}(\mathcal{T}_t) \leq (1 - 1/e)^t \mathbf{vol}(\mathcal{T}_0) \approx (0.63)^t \mathbf{vol}(\mathcal{T}_0)$.

Observe that our proposed cutting-plane-based active learning method in Algorithm 2 and 4 uses a modified splitting, where the weight vector in Proposition 1 is substituted by a mapping of the parameters $\theta$ to the feature space via function $f^{\text{two-layer}}$, which depends on point $x_{n_t}$ returned by the oracle at the step, along with the associated linear constraints $\mathcal{C}(\{n_t\}), \mathcal{C}'(\{n_t\})$. Since $f^{\text{two-layer}}$ is linear in $\theta$ as we recall that

$$f^{\text{two-layer}}(x_{n_t}; \theta) = [x_{n_t}^1 - x_{n_t}^1 \ ... \ x_{n_t}^P - x_{n_t}^P]\theta,$$

the set defined by $\{\theta | y_{n_t} \cdot f^{\text{two-layer}}(x_{n_t}; \theta) \geq 0\}$ forms a half-space in the parameter space. Additionally, since the constraints $\mathcal{C}(\{n_t\}), \mathcal{C}'(\{n_t\})$ are linear in $\theta$, the cutting set $\{\theta : y_{n_t} \cdot f^{\text{two-layer}}(x_{n_t}; \theta) \geq 0, \mathcal{C}(\{n_t\}), \mathcal{C}'(\{n_t\})\}$ in Algorithm 2 and 4 defines a convex polyhedron. This change suggests a non-trivial modification of the results given in Proposition 1. The following theorem is our contribution.

**Theorem 6.1** (Convergence with Center of Gravity). *Let $\mathcal{T} \subseteq \mathbb{R}^d$ be a convex body and let $\theta_G$ denote its center of gravity. The polyhedron cut given in Algorithm 2 and Algorithm 4 (assuming that the cut is active), i.e.,*

$$\mathcal{T} \cap \{\theta : y_n \cdot f^{\text{two-layer}}(x_n; \theta) \geq 0, \mathcal{C}(\{n\}), \mathcal{C}'(\{n\})\},$$

*where coupling $(x_n, y_n)$ is the data point returned by the cutting-plane oracle after receiving queried point $\theta_G$, partitions the convex body $\mathcal{T}$ into two subsets:*

$$\mathcal{T}_1 := \{\theta \in \mathcal{T} : y_n \cdot f^{\text{two-layer}}(x_n; \theta) \geq 0, \mathcal{C}(\{n\}), \mathcal{C}'(\{n\})\}$$
$$\mathcal{T}_2 := \{\theta \in \mathcal{T} : y_n \cdot f^{\text{two-layer}}(x_n; \theta) < 0, \ or \ \neg\mathcal{C}(\{n\}), \neg\mathcal{C}'(\{n\})\},$$

*where $\neg$ denotes the complement of a given set. Then $\mathcal{T}_1$ satisfies the following inequality:*

$$\mathbf{vol}(\mathcal{T}_1) < \left(1 - \frac{1}{e}\right) \cdot \mathbf{vol}(\mathcal{T}).$$

*Proof.* See Appendix G.1. $\square$

# 7 EXPERIMENTS

We validate our proposed training and active learning methods through extensive experiments, comparing them with various popular baselines from `scikit-activeml` (Kottke et al., 2021) and `DeepAL` (Huang, 2021). Synthetic data experiments are presented in Section 7.1, and real data experiments are presented in Section 7.2. An overview of each baseline is given in Appendix H.3. For implementation details and additional results, refer to Appendix H.

## 7.1 SYNTHETIC DATA EXPERIMENTS

In this section, we present small scale numerical experiments to verify the performance of our algorithm on both classification and regression tasks.

**Binary Classification on Synthetic Spiral.** We use a synthetic dataset of two intertwined spirals with positive and negative labels, respectively (see generation details in Appendix H.4). We generate 100 spiral data points with a 4:1 train-test split. Table 3 in Appendix H.4 presents the train and test accuracy of our cutting-plane AL (Algorithm 4) compared to popular deep AL baselines, with all methods evaluated with a query budget of 20 points (25% of train data). Our method achieved perfect accuracy on both sets, outperforming all baselines. Notably, the strong performance of our cutting-plane AL extends to the 3-layer case, achieving train/test accuracies of 0.71/0.60, while using only a fraction of neurons per layer (57 and 34, resp.) compared to the two-layer cutting-plane AL, which used 623 neurons. This result is illustrated in the corresponding decision boundary plot in Figure 2, where the 3-layer cutting-plane AL is one of the few methods to capture the spiral's rough shape despite using smaller embeddings, while the two-layer cutting-plane AL, with the same network

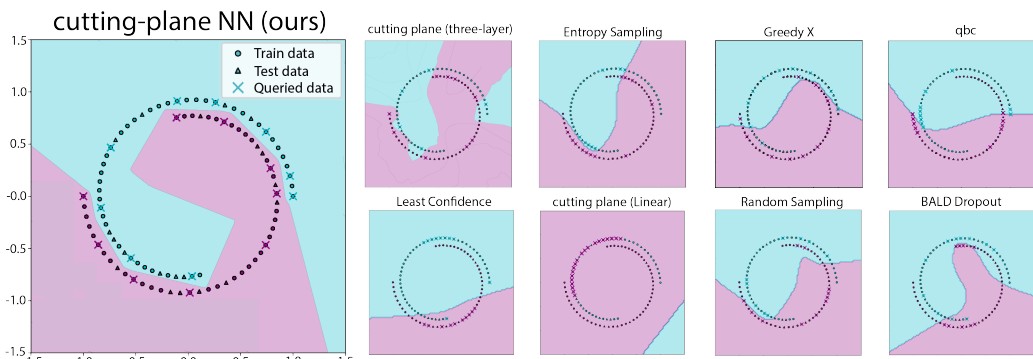

Figure 2: Decision boundaries for binary classification on the spiral dataset for the cutting-plane AL method using a two-layer ReLU neural network, alongside various deep AL baselines. For compactness, we also include the decision boundaries for the cutting-plane AL method with a three-layer ReLU network in the collage to demonstrate its feasibility. For fairness of comparison, we use the same two-layer ReLU network structure and embedding size of 623 for all methods. We enforce the same hyperparameters for all deep AL baselines and select the best performing number of training epochs at 2000 and a learning rate at 0.001 to ensure optimal performance. See Appendix H.3 for details.

structure as the baselines, precisely traces the spiral. We emphasize that the superior performance of our cutting-plane AL remains consistent across different random seeds. As shown in the error-bar plot in Figure 15, our approach reliably converges to the optimal classifier faster than all the tested baselines in the number of queries.

**Quadratic Regression.** We evaluate the performance of our method (Algorithm 6) against the same seven baselines used in the spiral task, along with two additional popular regression AL methods from `scikit-activeml`: greedy sampling in target space (GreedyT) and KL divergence maximization (kldiv), on a simple quadratic regression task. We generate 100 noise-free data points from the function $y = x^2$ and apply the AL methods on a 4:1 train-test split with a query budget of 20 points. Table 5 in Appendix H.5 shows the root mean square error (RMSE) (see Definition 5), and left of Figure 3 provides a visualization of the final predictions of our method compared with a selection of baselines. We refer readers to Figure 18 for the complete version. Our cutting-plane AL achieves the lowest train/test RMSE of 0.01/0.01, representing an over 80%/75% reduction compared to the next best-performing baseline.This superior performance remains consistent across random seeds, as demonstrated in the train/test error-bar plots right of Figure 3, where our method consistently converges faster to the optimal classifier in terms of the number of queries.

### 7.2 Sentiment Classification for Real Datasets Using LLM Embeddings

To demonstrate the viability of our method when applied to tackle real life tasks, we also explore the concatenation of microsoft Phi-2 (Javaheripi et al., 2023) model with our two-layer ReLU binary classifier for sentiment classification task on IMDB (Maas et al., 2011) movie review datasets. Specifically, the dataset contains $50k$ movie reviews collected online with each comment accompanied with binary labels where $+1$ denoting a positive review and $-1$ denoting a negative review. See Appendix H.6 for several training data examples. We test our proposed active learning algorithm conjuncted with our cutting-plane training scheme as described in Section 5 and Section 6 respectively. For our experiment, we first collect all last layer Phi-2 embeddings (corresponding to last token), which is of size $d = 2560$, for our training and testing movie reviews as our feature vectors. We follow the implementation details in Appendix H.1 and sample $P = 500$ activation patterns to cater for the large dimension feature space.

Figure 4 demonstrates our experimental results involving our method and various baselines. The left most figure compares the classification accuracy between our two-layer ReLU model and the linear model studied in (Louche & Ralaivola, 2015). As it can be seen, our model achieves higher accuracy within same query budget compared to the linear classifier, which demonstrates the effectiveness of using a nonlinear model in this task. The middle plot compares our active learning method and the random sampling method, both are with two-layer ReLU model. As expected, our active sampling scheme identifies important data points in each iteration, which helps to train a better network within same query budget. The right most plot compares between our newly-introduced

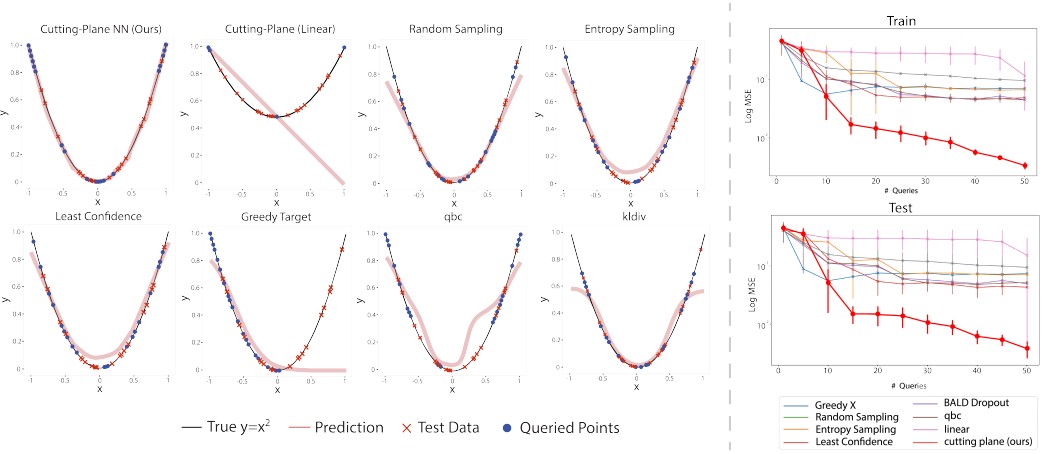

Figure 3: Left: Predictions for the quadratic regression task using the cutting-plane AL method with a two-layer ReLU neural network, alongside various deep AL baselines. For brevity, we present only the most representative examples here and refer to Figure 18 in Appendix H.5 for the full result. Implementation details can be found in Appendix H.5. We note that the linear cutting-plane AL for regression (Algorithm 7) becomes infeasible when solving for the next center after the fourth query. This failure is expected, see Appendix D. Thus the reported prediction for this method is based on 4 queries, while all other methods use 20 queries. Right: Logarithm of mean test/train RMSE across seeds (0-4) versus the number of queries for the two-layer cutting-plane AL and various baselines. The linear cutting-plane method is omitted from the plot due to its infeasibilty.

cutting-plane-based NN training scheme and classic stochastic gradient descent (SGD) method. For SGD baselines, we take one gradient step corresponding to each data query. For NN trained with SGD, people usually use batched data for gradient computation, thus within as few as 15 query points, it is reasonable that SGD does not make good progress. On the contrary, our cutting-plane training scheme achieves higher accuracy with this tiny training budget.

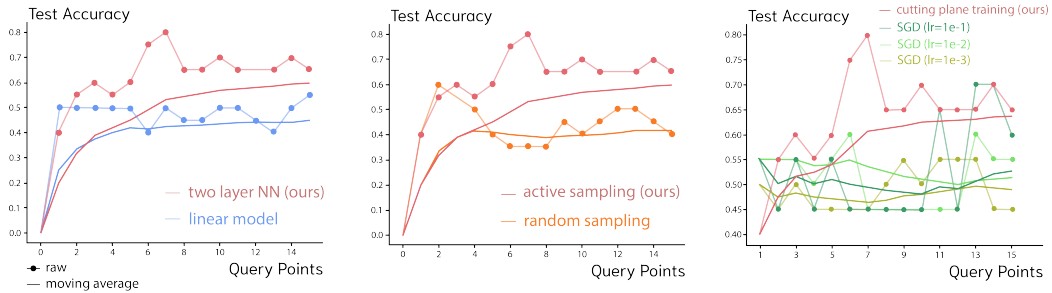

Figure 4: Sentiment analysis on IMDB movie review dataset with two-layer ReLU model. We take Phi-2 embedding as our training features and compare with various baselines. The result shows that the introduction of non-linearity improves upon linear model performance, our active sampling scheme effectively identifies valuable training points compared to random sampling, and our cutting-plane training scheme is more effective than SGD in this setting. See Section 7.2 for details. Linear and our reframed two-layer models are initialized to predict zero while two-layer NN trained with SGD has random weight initialization, thus starting from non-zero prediction.

## 8 CONCLUSION AND LIMITATION

In this work, we introduce a novel cutting-plane-based method for deep neural network training. We also explore an active learning scheme built on our proposed training framework. Despite its novelty, our current implementation has several key limitations that hinder its competitiveness with large-scale models trained using gradient-based methods: (1) our hyperplane subsampling process is not exhaustive; (2) our implementation relies on CPU-based convex program solver and is inefficient for large-scale problems with many variables; (3) our approach has so far been applied only to classification and regression tasks. **Due to space constraints, the prior work section is deferred to Appendix A and detailed conclusion and limitations section is deferred to Appendix B.**

## 9 REPRODUCIBILITY STATEMENT

We state that all the theoretical and experimental results presented in this paper are obtained with reproducibility in mind. Detailed proofs of the theoretical results from Sections 4 and 6 are provided in Appendices F and G, respectively. Furthermore, additional results, including the relaxation of certain constraints (e.g., data distribution requirements) and key generalizations (such as the extension from classification to regression), are discussed in Appendix E. To reproduce the experimental results, we provide detailed implementation procedures, including data generation and numerical solver deployment, in Appendix H. Our code has been submitted as a zip file, along with instructions for reproducing the plots presented in the main text.

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

# Appendix

## Table of Contents

## A  PRIOR WORK DISCUSSION

We emphasize the novelty of our work as the first to introduce both cutting-plane-based training schemes and cutting-plane-based active learning methods to deep neural network models. To the best of our knowledge, this work also provides the first convergence guarantees among existing active learning methods for nonlinear neural networks. As discussed in Section 3, the most closely related work is (Louche & Ralaivola, 2015), which also integrates the concepts of cutting-plane methods, active learning, and machine learning for binary classification tasks. We are not aware of any other studies that closely align with our approach. However, we review additional works that, while less directly connected, share some overlap with our research in key areas.

**Cutting-Plane Method.**  Cutting-plane methods are first introduced by Gomory (1958) for linear programming. Though being considered as ineffective after it was first introduced, it has been later shown by Balas et al. (1993) to be empirically useful when combined with branch-and-bound methods. Cutting-plane method is now heavily used in different commercial MILP solvers. Commonly employed cutting planes for solving convex programs are tightly connected to gradient information, which is also the underline logic for well-known Kelley's cutting-plane method (Kelley, 1960). Specifically, consider any minimization problem with objective $f(\theta)$ whose solution set, we denote as $\Theta$, is a convex set. Gradient-based cutting-plane method usually assumes the existence of an oracle such that given any input $\theta_0$, it either accepts $\theta_0 \in \Theta$ - and thus terminates with a satisfactory solution $\theta_0$ being found - or it returns a pair $(x, y)$ such that $x^T \theta_0 \leq y$ while $x^T \theta > y$ for any $\theta \in \Theta$, i.e., we receive a cutting plane that cuts between the current input $\theta_0$ and the desired solution set $\Theta$. Such cut is given by subgradient of $f$ at query point $\theta_0$. Consider any subgradient $g \in \nabla f(\theta_0)$, we have the inequality $f(\theta) \geq f(\theta_0) + g^T(\theta - \theta_0)$, which then raises the cut $g^T(\theta - \theta_0) \leq 0$ for minimization problem. Recent development of cutting-plane methods involves those designed for convex-concave games (Jiang et al., 2020), combinatorial optimization (Lee et al., 2015), and also application to traditional machine learning tasks such as regularized risk minimization, multiple kernel learning, and MAP inference in graphical models (Franc et al., 2011).

**Active Learning.**  With the fast growth of current deep learning model sizes, more and more data is in need for training an effective deep NN model. Compared to just feeding the model with a set of randomly-selected training samples, how to select the most informative data for each training iteration becomes a valuable question. Efficient data sampling algorithm is thus increasingly important, especially for current RL-based language model training scheme. With the increasing power of current pretrained models, researchers found that since most of daily questions can be addressed correctly by the model already, these questions are valueless for further boosting the language models' capacity. Therefore, actively selecting questions that LLM cannot address correctly is key to current LLM training (Lightman et al., 2023). Based on different learning settings, active learning strategies can be divided into: stream-based selective sampling used when the data is generated continuously (Woodward & Finn, 2017); pool-based sampling used when a pool of unlabeled data is presented (Gal et al., 2017b); query synthesis methods used when new samples can be generated for labeling. Based on different information measuring schemes, active learning algorithms can be further divided into uncertainty sampling (Lewis & Gale, 1994b) which selects data samples to reduce prediction uncertainty, query-by-committee sampling which involves multiple models for data selection, diversity-weighted method which selects the most diverse data sample, and expected error reduction method (Mussmann et al., 2022) which selects samples to best reduce models' expected prediction error.

**Convex NN.**  We note that the idea of introducing hyperplane arrangements to derive equivalent problem formulation for deep NN training task has also been investigated in prior convexification of neural network research. For example, Ergen & Pilanci (2021b) has exploited this technique to derive a convex program which is equivalent to two-layer ReLU model training task. More developments involving (Ergen & Pilanci, 2021a) which derives convex programs equivalent to training three-layer CNNs and (Zhang & Pilanci, 2024) which derives convex programs for diffusion models. However, those work neither derive linear programs as considered in our case, nor did they connect such reframed problems to active learning platform. On the contrast, they mainly focus on solving NN training problem by solving the equivalent convex program (directly) they have derived via convex program solver such as CVXPY (Diamond & Boyd, 2016).

## B  LIMITATIONS AND CONCLUSION

In this work, we introduce a novel cutting-plane-based method for deep neural network training, which, for the first time, enables the application of this approach to nonlinear models. Additionally, our new training scheme removes previous restrictions on the training data distribution and extends the method beyond binary classification to general regression tasks. We also explore an active learning scheme built on our proposed training framework, which inherits convergence guarantees from classic cutting-plane methods. Through both synthetic and real data experiments, we demonstrate the practicality and effectiveness of our training and active learning methods. In summary, our work introduces a novel, gradient-free approach to neural network training, demonstrating for the first time the feasibility of applying the cutting-plane method to neural networks, while also offering the first deep active learning method with convergence guarantees.

Despite its novelty, our current implementation has several key limitations that hinder its competitiveness with large-scale models trained using gradient-based methods. First, although we employ subsampling of activation patterns and propose an iterative filtering scheme (see Appendix F.4), the subsampling process is not exhaustive, which impacts model performance, especially with high-dimensional data. Refining the activation pattern sampling strategy could significantly improve results. Second, we rely on analytic center retrieval during training, which we solve using CVXPY. However, this solution is CPU-bound and becomes inefficient for large-scale problems with many variables. Developing a center-finding algorithm that leverages GPU parallelism is crucial to unlocking the full potential of our training method. Finally, while current large language model (LLM) training often involves cross-entropy loss, our approach has so far been applied only to classification and regression tasks. Extending our method to handle more diverse loss functions presents an exciting avenue for future research.

## C  KEY DEFINITIONS AND DEFERRED THEOREMS

### C.1  KEY DEFINITION

A notion central to the linear programming reformulation which enables the feasibility of our proposed cutting-plane based AL method is the notion of *hyperplane arrangement*. It has been briefly introduced in Section 4. We now give a formal definition.

**Definition 3** (Hyperplane Arrangement). *A hyperplane arrangement for a dataset $X \in \mathbb{R}^{n \times d}$, where $X$ contains $x_i$ in its rows, is defined as the collection of sign patterns generated by the hyperplanes. Let $\mathcal{A}$ denote the set of all possible hyperplane arrangement patterns:*

$$\mathcal{A} := \cup\{\text{sign}(Xw) : w \in \mathbb{R}^d\}, \tag{9}$$

*where the* sign *function is applied elementwise to the product $Xw$.*

*The number of distinct sign patterns in $\mathcal{A}$ is finite, i.e., $|\mathcal{A}| < \infty$. We define a subset $\mathcal{S} \subseteq \mathcal{A}$, representing the collection of sets corresponding to the positive signs in each element of $\mathcal{A}$:*

$$\mathcal{S} := \{\cup_{h_i=1}\{i\} : h \in \mathcal{A}\}. \tag{10}$$

*The cardinality of $\mathcal{S}$, denoted by $P$, represents the number of regions in the partition of $\mathbb{R}^d$ created by hyperplanes passing through the origin and orthogonal to the rows of $X$, or more birefly, $P$ is the number of regions formed by the hyperplane arrangement.*

As the readers may see in the definition, $P$, the number of regions formed by the hyperplane arrangement, increases with both the dimension and the size of the dataset $X$. To reduce computational costs of our cutting-plane AL algorithm, as we will discuss in Appendix H.1, we sample the number of regions in the partition instead of using the entire $\mathcal{S}$.

In addition to center of gravity (Definition 2) and analytic center (Definition 1), another widely-used notion for center is the center of maximum volume inscribed ellipsoid (MVE), which we also referenced in the main text. We hereby give its definition (Boyd & Vandenberghe, 2004).

**Definition 4** (Maximum volume inscribed ellipsoid (MVE)). *Given a convex body $C \subseteq \mathbb{R}^d$, the maximum volume inscribed ellipsoid inside $C$ is found by solving the below optimization problem:*

$$\begin{aligned} maximize \quad & \log \det B \\ subject\ to \quad & \sup_{\|u\|_2 \leq 1} \mathbb{1}_C(\varepsilon(u)) \leq 0, \end{aligned} \tag{11}$$

*where we have parametrized the ellipsoid as the image of the unit ball under an affine transformation:*

$$\varepsilon(u) := \{Bu + c : \|u\|_2 \leq 1\}$$

*with $c \in \mathbb{R}^d$ and $B \in S_{++}^n$. The optimal value of the variable $u$ in the convex program in 11 gives the center of the maximum volume inscribed ellipsoid of the convex body $C$, which is affine invariant. We denote it as $\theta_M(C)$, or $\theta_M$ as abbreviation.*

A metric important to our evaluation for the regression prediction in Section 7 is the notion of root mean square error, or RMSE.

**Definition 5** (Root Mean Square Error). *The root mean square error (RMSE) is a metric used to measure the difference between predicted values and the actual values in a regression model. For a set of predicted values $\hat{y}_i$ and true values $y_i$, the RMSE is given by:*

$$RMSE = \sqrt{\frac{1}{n} \sum_{i=1}^{n} (\hat{y}_i - y_i)^2}$$

*where $n$ is the total number of data points, $\hat{y}_i$ is the predicted value, and $y_i$ is the actual value. RMSE provides an estimate of the standard deviation of the prediction errors (residuals), giving an overall measure of the model's accuracy.*

### C.2 Key Theorems

**Key Theorems in Section 6.2.** We present the cornerstone theorem to our results in Section 6.2 given by Grünbaum (1960) on the bounds relating to convex body partitioning.

**Proposition 1** (Grünbaum's Inequality). *Let $\mathcal{T} \in \mathbb{R}^d$ be a convex body (i.e. a compact convex set) and let $\theta_G$ denote its center of gravity. Let $\mathcal{H} = \{x \in \mathbb{R}^d : w^T(\theta - \theta_G) = 0\}$ be an arbitrary hyperplane passing through $\theta_G$. This plane divides the convex body $\mathcal{T}$ in the two subsets:*

$$\mathcal{T}_1 := \{\theta \in \mathcal{T} : w^T\theta \geq w^T\theta_G\},$$
$$\mathcal{T}_2 := \{\theta \in \mathcal{T} : w^T\theta < w^T\theta_G\}.$$

*Then the following relations hold for $i = 1, 2$:*

$$\textbf{vol}(\mathcal{T}_i) \leq (1 - (\frac{d}{d+1})^{1/d})\textbf{vol}(\mathcal{T}) \leq (1 - 1/e)\textbf{vol}(\mathcal{T}). \tag{12}$$

*Proof.* See proof of Theorem 2 in Grünbaum (1960). □

## D Deferred Algorithms

### D.1 More on Cutting-Plane AL for Binary Classification

#### D.1.1 Linear Model

Algorithm 3 describes the original cutting-plane-based learning algorithm proposed by Louche & Ralaivola (2015). This algorithm, despite having pioneered in bridging the classic cutting-plane optimization algorithm with active learning for the first time, remains limited to linear decision boundary classification and can only be applied to shallow machine learning models. We have demonstrated its inability to handle nonlinear decision boundaries and simple regression tasks in Section 7, where the linearity of the final decision boundary and prediction returned by the cutting-plane AL algorithm is evident. Nevertheless, Algorithm 3 establishes a crucial foundation for the development of our proposed cutting-plane active learning algorithms (Algorithm 2, 5, 4, and 6).

---

**Algorithm 3** Generic (Linear) Cutting-Plane AL

---

1: $\mathcal{T}^0 \leftarrow \mathcal{B}_2$
2: $t \leftarrow 0$
3: **repeat**
4:     $\theta_c^t \leftarrow \text{center}(\mathcal{T}^t)$
5:     $x_{n_t}, y_{n_t} \leftarrow \text{QUERY}(\mathcal{T}^t, \mathcal{D})$
6:     **if** $y_{n_t}\langle\theta_c^t, x_{n_t}\rangle < 0$ **then**
7:         $\mathcal{T}^{t+1} \leftarrow \mathcal{T}^t \cap \{z : y_{n_t}\langle z, x_{n_t}\rangle \geq 0\}$
8:         $t \leftarrow t + 1$
9:     **end if**
10: **until** $\mathcal{T}^t$ is small enough
11: **return** $\theta_c^t$

---

1: **function** QUERY($\mathcal{T}, \mathcal{D}$)
2:     Sample $M$ points $s_1, \ldots, s_M$ from $\mathcal{T}$
3:     $g \leftarrow \frac{1}{M}\sum_{k=1}^M s_k$
4:     $x \leftarrow \arg\min_{x_i \in \mathcal{D}}\langle g, x_i\rangle$
5:     $y \leftarrow$ get label from an expert
6:     **return** $x, y$
7: **end function**

---

### D.1.2 NN MODEL WITH LIMITED QUERIES

We begin with Algorithm 4, which summarizes our proposed cutting-plane active learning algorithm under the second setup discussed in Section 6.1. In this case, the cutting-plane oracle has access to limited queries provided by the user and, in contrast to its query synthesis alternative (Algorithm 2), only makes the cut if the queried center mis-classifies the returned data point from the oracle. Hence, the convergence speed associated with Algorithm 4 hinges on how often the algorithm queries a center which incorrectly classifies the returned queried points before it reaches the optimal classifier. Therefore, the performance of this algorithm in terms of convergence speed depends not only on the geometry of the parameter version space but also on the effectiveness of the `Query` function to identify points from the limited dataset which gives the most informative evaluation of the queried center.

While Algorithm 4 still maintains similar rate and convergence guarantees as Algorithm 2, to optimize the empirical performance of Algorithm 4 (see discussions in Section 6.2), we therefore modify the `Query` function to query twice, once for minimal margin and once for maximal margin, to maximize the chances of the oracle returned points in correctly identifying mis-classification of the queried center. This modification greatly aids the performance of Algorithm 4, allowing the algorithm to make effective classification given very limited data. This is demonstrated in our experiment results in Section 7.

### D.1.3 NN MODEL WITH INEXACT CUT

Algorithm 5 summarizes our proposed cutting-plane active learning algorithm under the third setup metioned in Section 6.1. Under this scenario, the cutting-plane oracle has access to limited queries. However, in contrast to Algorithm 4, this cutting-plane AL algorithm always performs the cut regardless of whether the queried center mis-classifies the data point returned by the oracle. This is an interesting extension to consider. On the one hand, it can possibly speed up Algorithm 4 in making a decision boundary as the cut is effective in every iteration. On the other hand, however, the algorithm's lack of discern for the correctness of the queried candidate presents a non-trivial challenge to evaluate its convergence rate and whether it still maintains convergence guarantees. It turns out that we can still ensure convergence in the case of Algorithm 5, and the convergence rate can be quantified by measuring the "inexactness" of the cut in relation to a cut which directly passes through the queried center. We refer the readers to a detailed discussion on this matter in Appendix G.2. We would also like to emphasize that Algorithm 2, 5, and 4, although written for binary clas-

---

**Algorithm 4** Cutting-plane AL for Binary Classification with Limited Queries

---

1: $\mathcal{T}^0 \leftarrow \mathcal{B}_2$
2: $t \leftarrow 0$
3: $\mathcal{D}_{\text{AL}} \leftarrow \mathbf{0}$
4: **repeat**
5:     $\theta_c^t \leftarrow \text{center}(\mathcal{T}^t)$
6:     **for** $s$ in $\{1, -1\}$ **do**
7:         $(x_{n_t}, y_{n_t}) \leftarrow \text{QUERY}(\mathcal{T}^t, \mathcal{D} \setminus \mathcal{D}_{\text{AL}}, s)$
8:         **if** $y_{n_t} \cdot f^{\text{two-layer}}(x_{n_t}; \theta_c^t) < 0$ **then**
9:             $\mathcal{D}_{\text{AL}} \leftarrow \text{ADD}(\mathcal{D}_{\text{AL}}, (x_{n_t}, y_{n_t}))$
10:            $\mathcal{T}^{t+1} \leftarrow \mathcal{T}^t \cap \{\theta : y_{n_t} \cdot f^{\text{two-layer}}(x_{n_t}; \theta) \geq 0, \mathcal{C}(\{n_t\}), \mathcal{C}'(\{n_t\})\}$
11:            $t \leftarrow t + 1$
12:         **end if**
13:     **end for**
14: **until** $|\mathcal{D}_{\text{AL}}| \geq n_{\text{budget}}$
15: **return** $\theta_c^t$

---

1: **function** QUERY$(\theta, s)$
2:     $(x, y) \leftarrow \arg\min_{(x_i, y_i) \in \mathcal{D}_{\text{QS}}} s f^{\text{two-layer}}(x_{n_t}; \theta)$
3:     **return** $(x, y)$
4: **end function**

---

sification, can be easily extended to multi-class by using, for instance, the "one-versus-all" strategy. See Appendix E.2 for details.

While it may be intuitive for the optimal performance of the cutting-plane AL algorithms to translate from binary classification to the multi-class case, it is not entirely evident for us to expect similar performance of the algorithms for regression tasks, where the number of classes $K \to \infty$. What is surprising is that our cutting-plane AL still maintains its optimal performance on regression tasks, as evidenced by the synthetic toy example using quadratic regression in Section 7. Nevertheless, one can argue that the result is not so surprising after all as it is to be expected in theory due to intuition explained in Appendix E.2.

---

**Algorithm 5** Cutting-plane AL for Binary Classification with Inexact Cutting

---

1: $\mathcal{T}^0 \leftarrow \mathcal{B}_2$
2: $t \leftarrow 0$
3: $\mathcal{D}_{\text{AL}} \leftarrow \mathbf{0}$
4: **repeat**
5:     $\theta_c^t \leftarrow \text{center}(\mathcal{T}^t)$
6:     **for** $s$ in $\{1, -1\}$ **do**
7:         $(x_{n_t}, y_{n_t}) \leftarrow \text{QUERY}(\mathcal{T}^t, \mathcal{D} \setminus \mathcal{D}_{\text{AL}}, s)$
8:         $\mathcal{D}_{\text{AL}} \leftarrow \text{ADD}(\mathcal{D}_{\text{AL}}, (x_{n_t}, y_{n_t}))$
9:         $\mathcal{T}^{t+1} \leftarrow \mathcal{T}^t \cap \{\theta : y_{n_t} \cdot f^{\text{two-layer}}(x_{n_t}; \theta) \geq 0, \mathcal{C}(\{n_t\}), \mathcal{C}'(\{n_t\})\}$
10:         $t \leftarrow t + 1$
11:     **end for**
12: **until** $|\mathcal{D}_{\text{AL}}| \geq n_{\text{budget}}$
13: **return** $\theta_c^t$

---

1: **function** QUERY$(\theta, s)$
2:     $(x, y) \leftarrow \arg\min_{(x_i, y_i) \in \mathcal{D}_{\text{QS}}} s f^{\text{two-layer}}(x_{n_t}; \theta)$
3:     **return** $(x, y)$
4: **end function**

---

### D.2 MORE ON CUTTING-PLANE AL FOR REGRESSION

#### D.2.1 NN MODEL WITH LIMITED QUERIES

To generalize our classification cutting-plane AL algorithm to regression tasks, we need to make some adaptations to the cutting criterion and to how the cuts are being made. As the main body of the algorithm is the same across different setups (e.g. query synthesis, limited query, and inexact cuts) except for minor changes as the reader can see in Algorithm 2, 4, and 5, we only present the cutting-plane AL algorithm for regression under limited query. Algorithm 6 summarizes our proposed algorithm for training regression models via cutting-plane active learning. Here $\epsilon > 0$ is a threshold value for the $L_2-$norm error chosen by the user. Observe that the new $L_2$-norm cut of step

$$\mathcal{T}^{t+1} \leftarrow \mathcal{T}^t \cap \{\theta : \|y_{n_t} - f^{\text{two-layer}}(x_{n_t}; \theta)\|_2 \leq \epsilon, \mathcal{C}(\{n_t\}), \mathcal{C}'(\{n_t\})\}$$

consists simply of two linear cuts:

$$-\epsilon \leq y_{n_t} - f^{\text{two-layer}}(x_{n_t}; \theta) \leq \epsilon.$$

We can hence still ensure that the version space remains convex after each cut.

---

**Algorithm 6** Cutting-plane AL for Regression with Limited Queries

1: $\mathcal{T}^0 \leftarrow \mathcal{B}_2$
2: $t \leftarrow 0$
3: $\mathcal{D}_{\text{AL}} \leftarrow \mathbf{0}$
4: **repeat**
5:      $\theta_c^t \leftarrow \text{center}(\mathcal{T}^t)$
6:      **for** $s$ in $\{1, -1\}$ **do**
7:          $(x_{n_t}, y_{n_t}) \leftarrow \text{QUERY}(\mathcal{T}^t, \mathcal{D} \setminus \mathcal{D}_{\text{AL}}, s)$
8:          **if** $\|y_{n_t} - f^{\text{two-layer}}(x_{n_t}; \theta_c^t)\|_2 > \epsilon$ **then**
9:              $\mathcal{D}_{\text{AL}} \leftarrow \text{ADD}(\mathcal{D}_{\text{AL}}, (x_{n_t}, y_{n_t}))$
10:             $\mathcal{T}^{t+1} \leftarrow \mathcal{T}^t \cap \{\theta : \|y_{n_t} - f^{\text{two-layer}}(x_{n_t}; \theta)\|_2 \leq \epsilon, \mathcal{C}(\{n_t\}), \mathcal{C}'(\{n_t\})\}$
11:             $t \leftarrow t + 1$
12:          **end if**
13:      **end for**
14: **until** $|\mathcal{D}_{\text{AL}}| \geq n_{\text{budget}}$
15: **return** $\theta_c^t$

---

1: **function** QUERY$(\theta, s)$
2:      $(x, y) \leftarrow \arg\min_{(x_i, y_i) \in \mathcal{D}_{\text{QS}}} s f^{\text{two-layer}}(x_{n_t}; \theta)$
3:      **return** $(x, y)$
4: **end function**

---

#### D.2.2 LINEAR MODEL

Following the adaptation of our cutting-plane AL from classification to regression tasks, we similarly attempt to adapt the original linear cutting-plane AL (Algorithm 3) for regression. Algorithm 7 introduces an $\epsilon > 0$ threshold to account for the $L_2$-norm error between the predicted value and the actual target. This threshold controls both when a cut is made and the size of the cut. We applied this version of the algorithm (Algorithm 7) in the quadratic regression experiment detailed in Section 7. However, for nonlinear regression data—such as quadratic regression—this approach will necessarily fail. The prediction model $\langle \theta, x \rangle$ in Algorithm 7 is linear, whereas the underlying data distribution follows a nonlinear relationship, i.e., $y = x^2$. As a result, once the query budget starts accumulating nonlinearly distributed data points, no linear predictor can satisfy the regression task's requirements for small values of $\epsilon$.

In particular, after a certain number of iterations $t$, the cut

$$\mathcal{T}^t \cap \{\theta : \|y_{n_t} - \langle \theta, x_{n_t} \rangle\|_2 \leq \epsilon\}$$

will eliminate the entire version space (i.e., $\mathcal{T}^{t+1} = \emptyset$). This occurs because the error between the linear prediction and the nonlinear true values (e.g., $y = x^2$) cannot be reduced sufficiently, disqualifying all linear predictors. This is precisely what we observed in our quadratic regression example

in Section 7. After four queries, the algorithm reported infeasibility in solving for the version space center, as the remaining version space had been reduced to an empty set. This infeasibility is a direct consequence of the mismatch between the linear model's capacity and the data's nonlinear nature.

---

**Algorithm 7** Linear Cutting-Plane AL for Regression

---

1: $\mathcal{T}^0 \leftarrow \mathcal{B}_2$
2: $t \leftarrow 0$
3: $\mathcal{D}_{\text{AL}} \leftarrow \mathbf{0}$
4: **repeat**
5: $\quad \theta_c^t \leftarrow \text{center}(\mathcal{T}^t)$
6: $\quad x_{n_t}, y_{n_t} \leftarrow \text{QUERY}(\mathcal{T}^t, \mathcal{D})$
7: $\quad$ **if** $\|y_{n_t} - \langle \theta_c^t, x_{n_t} \rangle\|_2 > \epsilon$ **then**
8: $\quad\quad \mathcal{T}^{t+1} \leftarrow \mathcal{T}^t \cap \{\theta : \|y_{n_t} - \langle \theta, x_{n_t} \rangle\|_2 \leq \epsilon\}$
9: $\quad\quad t \leftarrow t + 1$
10: $\quad$ **end if**
11: **until** $|\mathcal{D}_{\text{AL}}| \geq n_{\text{budget}}$
12: **return** $\theta_c^t$

---

1: **function** QUERY($\mathcal{T}, \mathcal{D}$)
2: $\quad$ Sample $M$ points $s_1, \ldots, s_M$ from $\mathcal{T}$
3: $\quad g \leftarrow \frac{1}{M} \sum_{k=1}^{M} s_k$
4: $\quad x \leftarrow \arg\min_{x_i \in \mathcal{D}} \langle g, x_i \rangle$
5: $\quad y \leftarrow$ get label from an expert
6: $\quad$ **return** $x, y$
7: **end function**

---

### D.3 Minimal Margin Query Strategy

Here we note that in our main algorithm 2, we always query points with highest prediction confidence by setting

$$(x, y) \leftarrow \arg\min_{(x_i, y_i) \in \mathcal{D}_{\text{QS}}} s f^{\text{two-layer}}(x_{n_t}; \theta).$$

However, we indeed allow more custom implementation of query selection. For example, an alternative approach is to select data with minimal prediction margin, i.e.

$$(x, y) \leftarrow \arg\min_{(x_i, y_i) \in \mathcal{D}_{\text{QS}}} |f^{\text{two-layer}}(x_{n_t}; \theta)|.$$

We experiment with this query strategy in our real dataset experiments.

## E   Key Generalization to Cutting-Plane AL

In this section, we discuss two important generalizations of our cutting-plane AL method: (i). the relaxation in data distribution requirement and (ii). the extension from classification to regression.

### E.1 Relaxed Data Distribution Requirement

For linear classifier $f(x; \theta) = x^T \theta$, the training data is expected to be linearly separable for an optimal $\theta^\star$ to exist. However, this constraint on training data distribution is too restrictive in real scenarios. With ReLU model, due to its uniform approximation capacity, the training data is not required to be linearly separable as long as there is a continuous function $h$ such that $\text{sign}(h(x)) = y$ for all $(x, y)$ pairs. Due to the discrete nature of sampled data points, this is always satisfiable, thus we can totally remove the prerequisite on training data. For sake of completeness, we provide a version of uniform approximation capacity of ReLU below, with an extended discussion about a sample compression perspective of our cutting-plane based model training scheme thereafter. Given the fact that two-layer model has weaker approximation capacity compared to deeper models, we here consider only two-layer model without loss of generality. Uniform approximation capacity of single hidden layer NN has been heavily studied (Chen & Chen, 1993; Chui & Li, 1992; Costarelli

et al., 2013; Cotter, 1990; Cybenko, 1989; ichi Funahashi, 1989; Gallant & White, 1988; Hornik, 1991; Mhaskar & Micchelli, 1992; Leshno et al., 1993; Pinkus, 1999), here we present the version of ReLU network which we find most explicit and close to our setting.

**Theorem E.1.** *(Theorem 1.1 in (Shen et al., 2022)) For any continuous function $f \in C([0,1]^d)$, there exists a two-layer ReLU network $\phi = (xW_1 + b_1)_+ W_2 + b_2$ such that*

$$\|f - \phi\|_{L^p([0,1]^d)} \in \mathcal{O}\left(\sqrt{d}\omega_f\left((m^2 \log m)^{-1/d}\right)\right)$$

*where $\omega_f(\cdot)$ is the modulus of continuity of $f$.*

The above result can be extended to any $f \in C([-R, R]^d)$, see Theorem 2.5 in Shen et al. (2022) for discussion. Therefore, for any fixed dimension $d$, as we increase number of hidden neurons, we are guaranteed to approximate $f$ a.e. (under proper condition on $\omega_f$). Thus the assumption on training data can be relaxed to existence of some $\{W_1, b_1, W_2, b_2\}$ such that $\mathrm{sign}(\phi(x)) = y$, which happens almost surely. A minor discrepancy here is that the $\phi$ being considered in above Theorem incorporates the bias terms $b_1, b_2$ while our Theorem 4.1 considers two-layer NN of form $(xW_1)_+ W_2$. We note here that $(xW_1)_+ W_2$ is essentially the same as the one with layer-wise bias. To see this, we can append the data $x$ by a 1-value entry to accommodate for $b_1$. After this modification, we can always separate a neuron of form $(XW_{1i})_+ W_{2i}$, set $W_{1(d+1)} = 1$ and zeros elsewhere, and $W_{2i}$ would then accommodate for bias $b_2$. Therefore, our relaxed requirement on training data still persists. For deeper ReLU NNs, similar uniform approximation capability has been established priorly and furthermore the general NN form we considered in Theorem 4.2 incorporates the form with layer-wise bias and thus we still have such relaxation on training data distribution, see Appendix F.3 for more explanation.

### E.2 FROM CLASSIFICATION TO REGRESSION

Algorithm 2 shows how we train deep NN for binary classification task with cutting-plane method. Nevertheless, it can be easily extended to multi-class using, for instance, the so-called "one-versus-all" classification strategy. To illustrate, to extend the binary classifier to handle $K$ classes under this approach, we decompose the multi-class problem into $K$ binary subproblems. Specifically, for each class $\mathcal{C}_k$, we define a binary classification task as:

$$\text{Classify between } \mathcal{C}_k \text{ and } \bigcup_{i \neq k} \mathcal{C}_i,$$

where we classify $\mathcal{C}_k$ against all other classes combined as a single class. This creates $K$ binary classification problems, each corresponding to distinguishing one class from the rest.

In fact, our cutting-plane AL can be even applied to the case of regression, where the number of classes $K \to \infty$. The core intuition behind this is still the uniform approximation capability of nonlinear ReLU model. Given any training data $x$ with its label $y$, we want a model $f(x; \theta)$ to be able to predict $y$ exactly. Here the data label is no longer limited to plus or minus one and can be any continuous real number. For linear model $f(x; \theta) = x^T \theta$ considered in (Louche & Ralaivola, 2015), train such a predictor is almost impossible since it is highly unlikely there exists such a $\theta$ for real dataset. However, with our ReLU model, we are guaranteed there is a set of NN weights that would serve as a desired predictor due to its uniform approximation capacity.

Therefore, for regression task, the training algorithm will be exactly the same as Algorithm 3 instead that the original classification cut $y_{n_t} f(x_{n_t}; \theta) \geq 1$ will be replaced with $f(x_{n_t}; \theta) = y_{n_t}$. All other activation pattern constraints are leaved unchanged. A minor discrepancy here is that though theoretically sounding, the strict inequality $f(x_{n_t}; \theta) = y_{n_t}$ may raise numerical issues in practical implementation. Thus, we always include a trust region as $y_{n_t} - \epsilon \leq f(x_{n_t}; \theta) \leq y_{n_t} + \epsilon$ with some small $\epsilon$ for our experiments. See Algorithm 6 for our implementation details.

## F DEFERRED PROOFS AND EXTENSIONS IN SECTION 4

### F.1 PROOF OF THEOREM 4.1

We prove the equivalence in two directions, we first show that if there exists $W_1, W_2$ to $y \odot ((XW_1)_+ W_2) \geq 1$, then we can find solution $\{u_i, u_i'\}$ to Problem (3); we then show that when

there is solution $\{u_i, u_i'\}$ to Problem (3) and the number of hidden neurons $m \geq 2P$, then there exists $W_1, W_2$ such that $y \odot ((XW_1)_+ W_2) \geq 1$. We last show that given solution $\{u_i, u_i'\}$ to Problem (3), after finding correspondent $W_1, W_2$, the prediction for any input $\tilde{x}$ can be simply computed by $\sum_{i=1}^{P}(\tilde{x}u_i)_+ - (\tilde{x}u_i')_+$. We now start with our first part and assume the existence of $W_1, W_2$ to $y \odot ((XW_1)_+ W_2) \geq 1$, i.e.,

$$y \odot \sum_{j=1}^{m} (XW_{1j})_+ W_{2j} \geq 1,$$

from which we can derive

$$y \odot \sum_{j=1}^{m} D_j^1 X W_{1j} W_{2j} \geq 1, \tag{13}$$

where $D_j^1 = \text{diag}(\mathbb{1}\{XW_{1j} \geq 0\}) \in \mathbb{R}^{n \times n}$. Now consider set of pairs of $\{u_j, u_j'\}$ given by $u_j = W_{1j}W_{2j}, u_j' = 0$ for $j \in \{j|W_{2j} \geq 0\}$ and $u_j = 0, u_j' = -W_{1j}W_{2j}$ for $j \in \{j|W_{2j} < 0\}$. We thus have by (13)

$$y \odot \sum_{j=1}^{m} D_j^1 X(u_j - u_j') \geq 1, (2D_j^1 - I)Xu_j \geq 0, (2D_j^1 - I)Xu_j' \geq 0.$$

The only discrepancy between our set of $\{u_j, u_j'\}$ pairs and our desired solution to Problem (3) is we want to match $D_j^1$'s in equation (13) to $D_i$'s in Problem (3). This is achieved by observing that whenever we have $D_a^1 = D_b^1 = D_{(a,b)}$ for some $a, b \in [m]$, we can merge them as

$$D_a^1 X(u_a - u_a') + D_b^1 X(u_b - u_b')$$
$$= D_{(a,b)} X((u_a + u_b) - (u_a' + u_b'))$$
$$= D_{(a,b)} X(u_{a+b} - u_{a+b}'),$$

with $(2D_{(a+b)} - I)Xu_{a+b} \geq 0$ and $(2D_{(a+b)} - I)Xu_{a+b}' \geq 0$ still hold. We can keep this merging for all activation patterns $\{D_j^1 | j \in [m]\}$. We are guaranteed to get

$$y \odot \sum_{j=1}^{\tilde{m}} \tilde{D}_j X(\tilde{u}_j - \tilde{u}_j') \geq 1, (2\tilde{D}_j - I)X\tilde{u}_j \geq 0, (2\tilde{D}_j - I)X\tilde{u}_j', \geq 0$$

where all $\tilde{D}_j, j \in [\tilde{m}]$ are different. Note since our $\{D_i | i \in [P]\}$ in Problem (3) loop over all possible activation patterns corresponding to $X$, it is always the case that $\tilde{m} \leq P$ and $\tilde{D}_j = D_i$ for some $i \in [P]$. Thus we get a solution $\{u_i, u_i'\}$ to Problem (3) by setting $u_i = \tilde{u}_k, u_i' = \tilde{u}_k'$ when $D_i = \tilde{D}_k$ for some $k \in [\tilde{m}]$. Otherwise we simply set $u_i = u_i' = 0$. This completes our proof of the first direction, we now turn to prove the second direction and assume that there is solution $\{u_i, u_i'\}$ to Problem (3) as well as the number of hidden neurons $m \geq 2P$. We aim to show that there exists $W_1, W_2$ such that $y \odot ((XW_1)_+ W_2) \geq 1$. Since we have

$$y \odot \sum_{i=1}^{P} (D_i X(u_i - u_i')) \geq 1, (2D_i - I)Xu_i \geq 0, (2D_i - I)Xu_i' \geq 0,$$

we are able to derive the below inequality by setting $v_i = u_i, \alpha_i = 1$ for $i \in [P]$ and $v_i = u_{i-P}', \alpha_i = -1$ for $i \in [P+1, 2P]$,

$$y \odot \sum_{i=1}^{2P} (Xv_i)_+ \alpha_i \geq 1.$$

Therefore, consider $W_1 \in \mathbb{R}^{d \times m}$ defined by $W_{1j} = v_j$ for $j \in [2P]$ and $W_{1j} = 0$ for any $j > 2P$, $W_2 \in \mathbb{R}^m$ defined by $W_{2j} = \alpha_j$ for $j \in [2P]$ and $W_{2j} = 0$ for any $j > 2P$. Then we achieve

$$y \odot ((XW_1)_+ W_2) = y \odot \sum_{i=1}^{2P} (Xv_i)_+ \alpha_i \geq 1,$$

as desired. Lastly, once a solution $\{u_i, u_i'\}$ to Problem (3) is given, we can find corresponding $W_1, W_2$ according to our analysis of second direction above. Then for any input $\tilde{x}$, the prediction given by $(\tilde{x}W_1)_+ W_2$ is simply $\sum_{i=1}^{P}(\tilde{x}u_i)_+ - (\tilde{x}u_i')_+$.

### F.2 PROOF OF THEOREM 4.2

Similar to proof of Theorem 4.1, we carry out the proof in two directions. We prove first that if there exists solution to Problem (5), then there is also solution to Problem (6). We then show that whenever there exists solution to Problem (6), there is also solution to Problem (5). Concerned with that the notation complexity of $(n+1)$-layer NN might introduce difficulty to follow the proof, we start with showing three-layer case as a concrete example, we then move on to $(n+1)$-layer proof which is more arbitrary.

**Proof for three-layer.** The three-layer ReLU model being considered is of the form $((XW_1)_+W_2)_+W_3$. The corresponding linear program is given by

find $\quad u_{ij}, u'_{ij}, v_{ij}, v'_{ij}$

s.t. $\quad y \odot \sum_{j=1}^{P_2} D_j^{(2)} \left( \sum_{i=1}^{P_1} D_i^{(1)} X(u_{ij} - u'_{ij}) - \sum_{i=1}^{P_1} D_i^{(1)} X(v_{ij} - v'_{ij}) \right) \geq 1,$

$\quad (2D_i^{(1)} - I)Xu_{ij} \geq 0, (2D_i^{(1)} - I)Xu'_{ij} \geq 0, (2D_i^{(1)} - I)Xv_{ij} \geq 0, (2D_i^{(1)} - I)Xv'_{ij} \geq 0,$

$\quad (2D_j^{(2)} - I) \left( \sum_{i=1}^{P_1} D_i^{(1)} X(u_{ij} - u'_{ij}) \right) \geq 0, (2D_j^{(2)} - I) \left( \sum_{i=1}^{P_1} D_i^{(1)} X(v_{ij} - v'_{ij}) \right) \geq 0,$

which is a rewrite of (6) with $n = 2$. Firstly, assume there exists solution $\{W_1, W_2, W_3\}$ to the problem $y \odot ((XW_1)_+W_2)_+W_3 \geq 1$. We want to show there exists $\{u_{ij}, u'_{ij}, v_{ij}, v'_{ij}\}$ solves the above problem. Note that by $y \odot ((XW_1)_+W_2)_+W_3 \geq 1$, we get

$$y \odot \left( \sum_{i=1}^{m_1} (XW_{1i})_+W_{2i} \right)_+ W_3 \geq 1.$$

Let $K_i$ denote $\text{sign}(XW_{1i})$, i.e, $(2K_i - I)XW_{1i} \geq 0$, we can write

$$y \odot \left( \sum_{i=1}^{m_1} K_i XW_{1i}W_{2i} \right)_+ W_3 \geq 1.$$

Expand on the outer layer neurons, we get

$$y \odot \sum_{j=1}^{m_2} \left( \sum_{i=1}^{m_1} K_i XW_{1i}W_{2ij} \right)_+ W_{3j} \geq 1.$$

Construct $c_{ij} = W_{1i}W_{2ij}$ whenever $W_{2ij} \geq 0$ and 0 otherwise, $c'_{ij} = -W_{1i}W_{2ij}$ whenever $W_{2ij} < 0$ and 0 otherwise, we can write

$$y \odot \sum_{j=1}^{m_2} \left( \sum_{i=1}^{m_1} K_i X(c_{ij} - c'_{ij}) \right)_+ W_{3j} \geq 1, (2K_i - I)Xc_{ij} \geq 0, (2K_i - I)Xc'_{ij} \geq 0.$$

Denote $\text{sign}(\sum_{i=1}^{m_1} K_i X(c_{ij} - c'_{ij}))$ as $K_j^{(2)}$, we thus have

$$y \odot \sum_{j=1}^{m_2} K_j^{(2)} \left( \sum_{i=1}^{m_1} K_i X(c_{ij} - c'_{ij}) \right) W_{3j} \geq 1,$$

with $(2K_i - I)Xc_{ij} \geq 0, (2K_i - I)Xc'_{ij} \geq 0, (2K_j^{(2)} - I)(\sum_{i=1}^{m_1} K_i X(c_{ij} - c'_{ij})) \geq 0$. We construct $\{d_{ij}, d'_{ij}, e_{ij}, e'_{ij}\}$ by setting $d_{ij} = c_{ij}W_{3j}, d'_{ij} = c'_{ij}W_{3j}$ when $W_{3j} \geq 0$ and 0 otherwise, setting $e_{ij} = -c_{ij}W_{3j}, e'_{ij} = -c'_{ij}W_{3j}$ when $W_{3j} < 0$ and 0 otherwise, and we will arrive at

$$y \odot \sum_{j=1}^{m_2} K_j^{(2)} \left( \sum_{i=1}^{m_1} K_i X(d_{ij} - d'_{ij}) - \sum_{i=1}^{m_1} K_i X(e_{ij} - e'_{ij}) \right) \geq 1,$$

where

$$(2K_i - I)Xd_{ij} \geq 0, (2K_i - I)Xd'_{ij} \geq 0, (2K_i - I)Xe_{ij} \geq 0, (2K_i - I)Xe'_{ij} \geq 0$$

$$(2K_j^{(2)} - I)\sum_{i=1}^{m_1} K_i X(d_{ij} - d'_{ij}) \geq 0, (2K_j^{(2)} - I)\sum_{i=1}^{m_1} K_i X(e_{ij} - e'_{ij}) \geq 0,$$

which is already of form we want. We are left with matching $m_i$ with $P_i$ for $i \in \{1, 2\}$ and matching $D_j^{(2)}$ to $K_j^{(2)}$, $D_i^{(1)}$ to $K_i$, $\{u_{ij}, u'_{ij}, v_{ij}, v'_{ij}\}$ to $\{d_{ij}, d'_{ij}, e_{ij}, e'_{ij}\}$. We achieve this by observing that whenever there is duplicate $K_{j_1}^{(2)} = K_{j_2}^{(2)}$, we can merge the corresponding terms as

$$K_{j_1}^{(2)} \left(\sum_{i=1}^{m_1} K_i X(d_{ij_1} - d'_{ij_1}) - \sum_{i=1}^{m_1} K_i X(e_{ij_1} - e'_{ij_1})\right) + K_{j_2}^{(2)} \left(\sum_{i=1}^{m_1} K_i X(d_{ij_2} - d'_{ij_2}) - \sum_{i=1}^{m_1} K_i X(e_{ij_2} - e'_{ij_2})\right)$$

$$= K_{(j_1,j_2)}^{(2)} \left(\sum_{i=1}^{m_1} K_i X\left((d_{ij_1} + d_{ij_2}) - (d'_{ij_1} + d'_{ij_2})\right) - \sum_{i=1}^{m_1} K_i X\left((e_{ij_1} + e_{ij_2}) - (e'_{ij_1} + e'_{ij_2})\right)\right)$$

$$= K_{(j_1,j_2)}^{(2)} \left(\sum_{i=1}^{m_1} K_i X(d_{i(j_1+j_2)} - d'_{i(j_1+j_2)}) - \sum_{i=1}^{m_1} K_i X(e_{i(j_1+j_2)} - e'_{i(j_1+j_2)})\right),$$

where $K_{(j_1,j_2)}^{(2)} := K_{j_1}^{(2)}, d_{i(j_1+j_2)} := d_{ij_1} + d_{ij_2}, d'_{i(j_1+j_2)} = d'_{ij_1} + d'_{ij_2}, e_{i(j_1+j_2)} = e_{ij_1} + e_{ij_2},$ $e'_{i(j_1+j_2)} = e'_{ij_1} + e'_{ij_2}$. We have as constraints

$$(2K_{(j_1,j_2)}^{(2)} - I)\left(\sum_{i=1}^{m_1} K_i X(d_{i(j_1+j_2)} - d'_{i(j_1+j_2)})\right) \geq 0,$$

$$(2K_{(j_1,j_2)}^{(2)} - I)\left(\sum_{i=1}^{m_1} K_i X(e_{i(j_1+j_2)} - e'_{i(j_1+j_2)})\right) \geq 0,$$

$$(2K_i - I)Xd_{i(j_1+j_2)} \geq 0, (2K_i - I)Xd'_{i(j_1+j_2)} \geq 0,$$

$$(2K_i - I)Xe_{i(j_1+j_2)} \geq 0, (2K_i - I)Xe'_{i(j_1+j_2)} \geq 0.$$

Keep such merging until all $K_j^{(2)}$ are different, we arrive at

$$y \odot \sum_{j=1}^{\tilde{m}_2} \overline{K}_j^{(2)} \left(\sum_{i=1}^{m_1} K_i X(\overline{d}_{ij} - \overline{d'_{ij}}) - \sum_{i=1}^{m_1} K_i X(\overline{e}_{ij} - \overline{e}'_{ij})\right) \geq 1,$$

with

$$(2K_i - I)X\overline{d}_{ij} \geq 0, (2K_i - I)X\overline{d}'_{ij} \geq 0, (2K_i - I)X\overline{e}_{ij} \geq 0, (2K_i - I)X\overline{e}'_{ij} \geq 0,$$

$$(2\overline{K}_j^{(2)} - I)\left(\sum_{i=1}^{m_1} K_i X(\overline{d}_{ij} - \overline{d}'_{ij})\right) \geq 0, (2\overline{K}_j^{(2)} - I)\left(\sum_{i=1}^{m_1} K_i X(\overline{e}_{ij} - \overline{e}'_{ij})\right) \geq 0,$$

where $\tilde{m}_2 \leq P_2, \overline{K}_j^{(2)} \in \{D^{(2)}\}$ and $\overline{K}_j^{(2)}$ all different. We now proceed to match $K_i$ and $D_i^{(1)}$. Consider $\sum_{i=1}^{m_1} K_i X(\overline{d}_{ij} - \overline{d}'_{ij})$, if $K_v = K_q$ for some $v, q$, we can merge them as

$$K_v X(\overline{d}_{vj} - \overline{d}'_{vj}) + K_q X(\overline{d}_{qj} - \overline{d}'_{qj}) = K_{(v,q)} X(\overline{d}_{(v+q)j} - \overline{d}'_{(v+q)j}),$$

where $K_{(v,q)} := K_v, \overline{d}_{(v+q)j} := \overline{d}_{vj} + \overline{d}_{qj}, \overline{d}'_{(v+q)j} := \overline{d}'_{vj} + \overline{d}'_{qj}$. The constraints are $(2K_{(v,q)} - I)X\overline{d}_{(v+q)j} \geq 0, (2K_{v,q} - I)X\overline{d}'_{(v+q)j} \geq 0$, and we still have

$$(2\overline{K}_j^{(2)} - I)\left(\sum_{i \in [m_1], i \neq v, i \neq q} K_i X(\overline{d}_{ij} - \overline{d}_{ij'}) + K_{(v,q)} X(\overline{d}_{(v+q)j} - \overline{d}'_{(v+q)j})\right) \geq 0.$$

Continue such merging and also for $\{\overline{e}_{ij}, \overline{e}'_{ij}\}$, we finally arrive at

$$y \odot \sum_{j=1}^{\tilde{m}_2} \overline{K}_j^{(2)} \left( \sum_{i=1}^{\tilde{m}_1} \hat{K}_i X(\hat{\overline{d}}_{ij} - \hat{\overline{d}}'_{ij}) - \sum_{i=1}^{\tilde{m}_1} \hat{K}_i X(\hat{\overline{e}}_{ij} - \hat{\overline{e}}'_{ij}) \right) \geq 1,$$

with

$$(2\hat{K}_i - I)X\hat{\overline{d}}_{ij} \geq 0, (2\hat{K}_i - I)X\hat{\overline{d}}'_{ij} \geq 0, (2\hat{K}_i - I)X\hat{\overline{e}}_{ij} \geq 0, (2\hat{K}_i - I)X\hat{\overline{e}}'_{ij} \geq 0,$$

$$(2\overline{K}_j^{(2)} - I) \left( \sum_{i=1}^{\tilde{m}_1} \hat{K}_i X(\hat{\overline{d}}_{ij} - \hat{\overline{d}}'_{ij}) \right) \geq 0, (2\overline{K}_j^{(2)} - I) \left( \sum_{i=1}^{\tilde{m}_1} \hat{K}_i X(\hat{\overline{e}}_{ij} - \hat{\overline{e}}'_{ij}) \right) \geq 0.$$

Now, for any $D_j^{(2)} \notin \{\overline{K}^{(2)}\}$, we set all $u_{ij} = u'_{ij} = v_{ij} = v'_{ij} = 0$. For $D_j^{(2)} = \overline{K}_{j'}^{(2)}$, for any $D_i^{(1)} \notin \{\hat{K}\}$, we set all $u_{ij} = u'_{ij} = v_{ij} = v'_{ij} = 0$. For $D_j^{(2)} = \overline{K}_{j'}^{(2)}, D_i^{(1)} = \hat{K}_{i'}$, we set $u_{ij} = \hat{\overline{d}}_{i'j'}, u'_{ij} = \hat{\overline{d}}'_{i'j'}, v_{ij} = \hat{\overline{e}}_{i'j'}, v'_{ij} = \hat{\overline{e}}'_{i'j'}$. Then we get exactly

$$y \odot \left( \sum_{j=1}^{P_2} D_j^{(2)} \left( \sum_{i=1}^{P_1} D_i^{(1)} X(u_{ij} - u'_{ij}) - \sum_{i=1}^{P_1} D_i^{(1)} X(v_{ij} - v'_{ij}) \right) \right) \geq 1,$$

and

$$(2D_i^{(1)} - I)Xu_{ij} \geq 0, (2D_i^{(1)} - I)Xu'_{ij} \geq 0, (2D_i^{(1)} - I)Xv_{ij} \geq 0, (2D_i^{(1)} - I)Xv'_{ij} \geq 0,$$

$$(2D_j^{(2)} - I) \left( \sum_{i=1}^{P_1} D_i^{(1)} X(u_{ij} - u'_{ij}) \right) \geq 0, (2D_j^{(2)} - I) \left( \sum_{i=1}^{P_1} D_i^{(1)} X(v_{ij} - v'_{ij}) \right) \geq 0,$$

which completes the proof of our first direction. We now turn on to prove the second direction, assume there exists $u_{ij}, u'_{ij}, v_{ij}, v'_{ij}$ such that

$$y \odot \left( \sum_{j=1}^{P_2} D_j^{(2)} \left( \sum_{i=1}^{P_1} D_i^{(1)} X(u_{ij} - u'_{ij}) - \sum_{i=1}^{P_1} D_i^{(1)} X(v_{ij} - v'_{ij}) \right) \right) \geq 1,$$

with

$$(2D_i^{(1)} - I)Xu_{ij} \geq 0, (2D_i^{(1)} - I)Xu'_{ij} \geq 0, (2D_i^{(1)} - I)Xv_{ij} \geq 0, (2D_i^{(1)} - I)Xv'_{ij} \geq 0,$$

$$(2D_j^{(2)} - I) \left( \sum_{i=1}^{P_1} D_i X(u_{ij} - u'_{ij}) \right) \geq 0, (2D_j^{(2)} - I) \left( \sum_{i=1}^{P_1} D_i X(v_{ij} - v'_{ij}) \right) \geq 0.$$

We want to show that there exists $W_1, W_2, W_3$ such that

$$y \odot ((XW_1)_+ W_2)_+ W_3 \geq 1.$$

We are able to derive

$$y \odot \left( \sum_{j=1}^{P_2} \left( \sum_{i=1}^{P_1} D_i X(u_{ij} - u'_{ij}) \right)_+ - \left( \sum_{i=1}^{P_1} D_i X(v_{ij} - v'_{ij}) \right)_+ \right) \geq 1,$$

and furthermore

$$y \odot \left( \sum_{j=1}^{P_2} \left( \sum_{i=1}^{P_1} (Xu_{ij})_+ - (Xu'_{ij})_+ \right)_+ - \left( \sum_{i=1}^{P_1} (Xv_{ij})_+ - (Xv'_{ij})_+ \right)_+ \right) \geq 1.$$

We thus construct $\{\kappa_{ij}, \alpha_{ij}\}$ by setting $\kappa_{ij} = u_{ij}, \alpha_{ij} = 1$ for $i \in [P_1], \kappa_{ij} = u'_{(i-P_1)j}, \alpha_{ij} = -1$ for $i \in [P_1 + 1, 2P_1]$. We similarly construct $\{\kappa'_{ij}, \alpha'_{ij}\}$ with $\{v_{ij}, v'_{ij}\}$, we thus get

$$y \odot \left( \sum_{j=1}^{P_2} \left( \sum_{i=1}^{2P_1} (X\kappa_{ij})_+ \alpha_{ij} \right)_+ - \left( \sum_{i=1}^{2P_1} (X\kappa'_{ij})_+ \alpha'_{ij} \right)_+ \right) \geq 1.$$

We construct $\{u_j\}$ by setting $u_j = 1$ for $j \in [P_2]$ and $u_j = -1$ for $j \in [P_2 + 1, 2P_2]$. We construct also $\{\tilde{\kappa}_{ij}, \tilde{\alpha}_{ij}\}$ such that $\tilde{\kappa}_{ij} = \kappa_{ij}, \tilde{\alpha}_{ij} = \alpha_{ij}$ for $i \in [2P_1], j \in [P_2]$, $\tilde{\kappa}_{ij} = \kappa'_{i(j-P_2)}, \tilde{\alpha}_{ij} = \alpha'_{i(j-P_2)}$ for $i \in [2P_1], j \in [P_2 + 1, 2P_2]$, we thus get

$$ y \odot \left( \sum_{j=1}^{2P_2} \left( \sum_{i=1}^{2P_1} (X\tilde{\kappa}_{ij})_+ \tilde{\alpha}_{ij} \right)_+ u_j \right) \geq 1. $$

Therefore, we arrive at $y \odot ((XW_1)_+ W_2)_+ W_3 \geq 1$ by focusing on $j \leq 2P_2, i \leq 4P_1 P_2$, i.e., setting parameters with $(i, j)$ indices exceeding these thresholds to be all zero. We then set $W_{1i} = \tilde{\kappa}_{ab}$ for $a = \lfloor \frac{i-1}{2P_1} \rfloor + 1, b = (i-1)\%2P_1 + 1, W_{2ij} = \tilde{\alpha}_{(i-1)\%2P_1+1}$ for $i \in [(j-1) * 2P_1 + 1, j * 2P_1]$ and 0 otherwise, $W_{3j} = u_j$.

**Proof for $n + 1$-layer.** We now provide proof for ReLU model of arbitrary depth. The logic follows the three-layer case proof above, despite the notation now represents $n + 1$-layer model for arbitrary $n$. We first assume that there exists $W_1, W_2, \cdots, W_{n+1}$ that satisfies problem (5). We want to show that there exists $\{a_{j_n \cdots j_1}^{c_n \cdots c_1}\}$ which satisfies problem (6). We first span the inner most neuron to get

$$ y \odot \left( \cdots \left( \left( \left( \sum_{i_1=1}^{m_1} (XW_{1i_1})_+ W_{2i_1} \right)_+ W_3 \right)_+ W_4 \right)_+ W_5 \cdots \right)_+ W_{n+1} \geq 1. $$

Denote $\text{sign}(XW_{1i})$ as $K_{i_1}^{(1)}$, i.e., $(2K_{i_1}^{(1)} - I)XW_{1i_1} \geq 0$. We can then rewrite the above inequality as

$$ y \odot \left( \cdots \left( \left( \left( \sum_{i_1=1}^{m_1} K_{i_1}^{(1)} XW_{1i_1} W_{2i_1} \right)_+ W_3 \right)_+ W_4 \right)_+ W_5 \cdots \right)_+ W_{n+1} \geq 1. $$

We then expand the second last inner layer as

$$ y \odot \left( \cdots \left( \left( \sum_{i_2=1}^{m_2} \left( \sum_{i_1=1}^{m_1} K_{i_1}^{(1)} XW_{1i_1} W_{2i_1 i_2} \right)_+ W_{3i_2} \right)_+ W_4 \right)_+ W_5 \cdots \right)_+ W_{n+1} \geq 1. $$

We construct $\{b_{i_1 i_2}^{(1)}, b'^{(1)}_{i_1 i_2}\}$ such that $b_{i_1 i_2}^{(1)} = W_{1i_1} W_{2i_1 i_2}$ when $W_{2i_1 i_2} \geq 0$ and 0 otherwise, $b'^{(1)}_{i_1 i_2} = -W_{1i_1} W_{2i_1 i_2}$ when $W_{2i_1 i_2} < 0$ and 0 otherwise. Therefore we get

$$ y \odot \left( \cdots \left( \left( \sum_{i_2=1}^{m_2} \left( \sum_{i_1=1}^{m_1} K_{i_1}^{(1)} X \left( b_{i_1 i_2}^{(1)} - b'^{(1)}_{i_1 i_2} \right) \right)_+ W_{3i_2} \right)_+ W_4 \right)_+ W_5 \cdots \right)_+ W_{n+1} \geq 1, $$

with constraints $(2K_{i_1}^{(1)} - I)Xb_{i_1 i_2}^{(1)} \geq 0, (2K_{i_1}^{(1)} - I)Xb'^{(1)}_{i_1 i_2} \geq 0$. Let $K_{i_2}^{(2)}$ denote $\text{sign}(\sum_{i_1=1}^{m_1} K_{i_1}^{(1)} X(b_{i_1 i_2}^{(1)} - b'^{(1)}_{i_1 i_2}))$, i.e., $(2K_{i_2}^{(2)} - I)(\sum_{i_1=1}^{m_1} K_{i_1}^{(1)} X(b_{i_1 i_2}^{(1)} - b'^{(1)}_{i_1 i_2})) \geq 0$, we thus have

$$ y \odot \left( \cdots \left( \left( \sum_{i_2=1}^{m_2} K_{i_2}^{(2)} \left( \sum_{i_1=1}^{m_1} K_{i_1}^{(1)} X \left( b_{i_1 i_2}^{(1)} - b'^{(1)}_{i_1 i_2} \right) \right) W_{3i_2} \right)_+ W_4 \right)_+ W_5 \cdots \right)_+ W_{n+1} \geq 1, $$

with constraints

$$ (2K_{i_1}^{(1)} - I)Xb_{i_1 i_2}^{(1)} \geq 0, (2K_{i_1}^{(1)} - I)Xb'^{(1)}_{i_1 i_2} \geq 0, (2K_{i_2}^{(2)} - I) \left( \sum_{i_1=1}^{m_1} K_{i_1}^{(1)} X(b_{i_1 i_2}^{(1)} - b'^{(1)}_{i_1 i_2}) \right) \geq 0. $$

Expand one more hidden layer

$$ y \odot \left( \cdots \left( \sum_{i_3=1}^{m_3} \left( \sum_{i_2=1}^{m_2} K_{i_2}^{(2)} \left( \sum_{i_1=1}^{m_1} K_{i_1}^{(1)} X \left( b_{i_1 i_2}^{(1)} - b'^{(1)}_{i_1 i_2} \right) \right) W_{3i_2 i_3} \right)_+ W_{4i_3} \right)_+ W_5 \cdots \right)_+ W_{n+1} \geq 1. $$

Construct $\{b^{(11)}_{i_1 i_2 i_3}, b'^{(11)}_{i_1 i_2 i_3}\}$ by setting $b^{(11)}_{i_1 i_2 i_3} = b^{(1)}_{i_1 i_2} W_{3 i_2 i_3}$ and $b'^{(11)}_{i_1 i_2 i_3} = b'^{(1)}_{i_1 i_2} W_{3 i_2 i_3}$ when $W_{3 i_2 i_3} \geq 0$ and 0 otherwise. Construct $\{b^{(12)}_{i_1 i_2 i_3}, b'^{(12)}_{i_1 i_2 i_3}\}$ by setting $b^{(12)}_{i_1 i_2 i_3} = -b^{(1)}_{i_1 i_2} W_{3 i_2 i_3}$ and $b'^{(12)}_{i_1 i_2 i_3} = -b'^{(1)}_{i_1 i_2} W_{3 i_2 i_3}$ when $W_{3 i_2 i_3} < 0$ and 0 otherwise. Let $K^{(3)}_{i_3}$ denotes $\text{sign}(\sum^{m_2}_{i_2=1} K^{(2)}_{i_2} (\sum^{m_1}_{i_1=1} K^{(1)}_{i_1} X(b^{(1)}_{i_1 i_2} - b'^{(1)}_{i_1 i_2})) W_{3 i_2 i_3})$. Thus we have

$$
y \odot \left( \cdots \left( \sum^{m_3}_{i_3=1} K^{(3)}_{i_3} \left( \sum^{m_2}_{i_2=1} K^{(2)}_{i_2} \left( \sum^{m_1}_{i_1=1} K^{(1)}_{i_1} X \left( b^{(11)}_{i_1 i_2 i_3} - b'^{(11)}_{i_1 i_2 i_3} \right) \right) \right. \right. \right.
$$

$$
\left. \left. \left. - \sum^{m_2}_{i_2=1} K^{(2)}_{i_2} \left( \sum^{m_1}_{i_1=1} K^{(1)}_{i_1} X \left( b^{(12)}_{i_1 i_2 i_3} - b'^{(12)}_{i_1 i_2 i_3} \right) \right) \right) W_{4 i_3} \right)_+ W_5 \cdots \right)_+ W_{n+1} \geq 1,
$$

with constraints

$(2K^{(1)}_{i_1} - I) X b^{(11)}_{i_1 i_2 i_3} \geq 0, (2K^{(1)}_{i_1} - I) X b'^{(11)}_{i_1 i_2 i_3} \geq 0$

$(2K^{(1)}_{i_1} - I) X b^{(12)}_{i_1 i_2 i_3} \geq 0, (2K^{(1)}_{i_1} - I) X b'^{(12)}_{i_1 i_2 i_3} \geq 0,$

$(2K^{(2)}_{i_2} - I) \left( \sum^{m_1}_{i_1=1} K^{(1)}_{i_1} X \left( b^{(11)}_{i_1 i_2 i_3} - b'^{(11)}_{i_1 i_2 i_3} \right) \right) \geq 0, (2K^{(2)}_{i_2} - I) \left( \sum^{m_1}_{i_1=1} K^{(1)}_{i_1} X \left( b^{(12)}_{i_1 i_2 i_3} - b'^{(12)}_{i_1 i_2 i_3} \right) \right) \geq 0,$

$(2K^{(3)}_{i_3} - I) \left( \sum^{m_2}_{i_2=1} K^{(2)}_{i_2} \left( \sum^{m_1}_{i_1=1} K^{(1)}_{i_1} X \left( b^{(11)}_{i_1 i_2 i_3} - b'^{(11)}_{i_1 i_2 i_3} \right) \right) - \sum^{m_2}_{i_2=1} K^{(2)}_{i_2} \left( \sum^{m_1}_{i_1=1} K^{(1)}_{i_1} X \left( b^{(12)}_{i_1 i_2 i_3} - b'^{(12)}_{i_1 i_2 i_3} \right) \right) \right) \geq 0.$

For cleanness, we introduce the following notation, for any $c_i \in \{1,2\}$ and $2 \leq s \leq n-1$,

$$
\mathcal{T}^{(n-1)(n-2)\cdots(s)}_{c_{n-1} c_{n-2} \cdots c_s}(K^{(s)}) = \sum^{m_s}_{i_s=1} K^{(s)}_{i_s} \left( \mathcal{T}^{(n-1)(n-2)\cdots(s)(s-1)}_{c_{n-1} c_{n-2} \cdots c_s [c_{s-1}=1]}(K^{(s-1)}) - \mathcal{T}^{(n-1)(n-2)\cdots(s)(s-1)}_{c_{n-1} c_{n-2} \cdots c_s [c_{s-1}=2]}(K^{(s-1)}) \right).
$$

When $s = 1$,

$$
\mathcal{T}^{(n-1)(n-2)\cdots(2)(1)}_{c_{n-1} c_{n-2} \cdots c_2 c_1}(K^{(1)}) = \sum^{m_1}_{i_1=1} K^{(1)}_{i_1} X \left( b^{(c_{n-1} c_{n-2} \cdots c_1)}_{i_n i_{n-1} \cdots i_1} - b'^{(c_{n-1} c_{n-2} \cdots c_1)}_{i_n i_{n-1} \cdots i_1} \right).
$$

Proceed with the above splitting, under the newly defined notation, we will get

$$
y \odot \sum^{m_n}_{i_n=1} K^{(n)}_{i_n} \left( \mathcal{T}^{(n-1)}_1(K^{(n-1)}) - \mathcal{T}^{(n-1)}_2(K^{(n-1)}) \right) \geq 1
$$

with constraints

$$
(2K^{(s)}_{i_s} - I) \mathcal{T}^{(n-1)(n-2)\cdots(s)(s-1)}_{c_{n-1} c_{n-2} \cdots c_s c_{s-1}}(K^{(s-1)}) \geq 0, \forall 2 \leq s \leq n-1,
$$

$$
(2K^{(1)}_{i_1} - I) X b^{(c_{n-1} c_{n-2} \cdots c_1)}_{i_n i_{n-1} \cdots i_1} \geq 0, (2K^{(1)}_{i_1} - I) X b'^{(c_{n-1} c_{n-2} \cdots c_1)}_{i_n i_{n-1} \cdots i_1} \geq 0.
$$

Note we already have the form of (6) by combine $\{b^{(c_{n-1} c_{n-2} \cdots c_1)}_{i_n i_{n-1} \cdots i_1}\}$ and $\{b'^{(c_{n-1} c_{n-2} \cdots c_1)}_{i_n i_{n-1} \cdots i_1}\}$ into $\{b^{(c_n c_{n-1} c_{n-2} \cdots c_1)}_{i_n i_{n-1} \cdots i_1}\}$, the only thing left is to match $\{K^{(s)}\}$ with $\{D^{(n)}\}$, we do this by recursion. Consider any layer $l$, assume all $\{K^{(s)}\}, s \leq l$ can be matched with $\{D^{(n)}\}, n \leq l$. Now we consider the $(l+1)$-th layer. Note that all $K^{(l+1)}_{i_{l+1}} \in \{D^{(l+1)}\}$ for any $i_{l+1} \in [m_{l+1}]$. If there is duplicate neuron activation patterns, i.e., $K^{(l+1)}_a = K^{(l+1)}_b$ for some $a \neq b, a, b \in [m_{l+1}]$. Then we merge all lower-level neurons corresponding to $K^{(l+1)}_a$ and $K^{(l+1)}_b$ by summing up the corresponding (with respect to $i_1, i_2, \ldots, i_l$ indices) $b$ vectors. Both the layer output and ReLU sign constraints will be preserved for all layers up to $(l+1)$-th layer, and we thus get, after the merging, a new set of $\{\tilde{K}^{(l+1)}\}$ and $\{\tilde{b}\}$ that matches the problem (6) up to layer $(l+1)$, where we just set all parameters to be zero for any $j_{l+1} \in [P_{l+1}]$ such that $D_{j_{l+1}} \notin \{\tilde{K}^{(l+1)}\}$. Now, we only need to verify that the last inner layer's neuron can be matched. By the symmetry between $b^{(c_{n-1} c_{n-2} \cdots c_1)}_{i_n i_{n-1} \cdots i_1}$ terms, consider without loss of generality the neuron

$$
\sum^{m_1}_{i_1=1} K^{(1)}_{i_1} X b^{[c_{n-1}=1][c_{n-2}=1]\cdots[c_1=1]}_{i_n i_{n-1} \cdots i_1}, \tag{14}
$$

which we want to match with

$$\sum_{j_1=1}^{P_1} D_{j_1}^{(1)} X u_{j_n j_{n-1} \cdots j_1}^{[c_n=1][c_{n-1}=1][c_{n-2}=1]\cdots[c_1=1]}. \tag{15}$$

Note by construction we know $K_{i_1}^{(1)} \in \{D^{(1)}\}$ for any $i_1 \in [m_1]$. If there is any duplicate neurons $K_c^{(1)} = K_d^{(1)}$ for some $c \neq d, c, d \in [m_1]$. We denote $K_{(c,d)}^{(1)} = K_c^{(1)} = K_d^{(1)}$. Let $b_{i_n i_{n-1} \cdots [i_1 \leftarrow (c,d)]}^{11\cdots 1} = b_{i_n i_{n-1} \cdots [i_1=c]}^{11\cdots 1} + b_{i_n i_{n-1} \cdots [i_1=d]}^{11\cdots 1}$. Then we merge $K_c^{(1)}$ and $K_d^{(1)}$ by replacing them with only one copy of $K_{(c,d)}^{(1)}$, and set the corresponding $b$ vector to be $b_{i_n i_{n-1} \cdots [i_1=(c,d)]}$. Continue this process until there is no duplicate neurons in the last inner layer. We are guaranteed to get a set of $\{\tilde{K}_{i_1}^{(1)}\}$ and corresponding $\{\tilde{b}_{i_n i_{n-1} \cdots i_1}^{11\cdots 1}\}$ such that $\tilde{K}_{i_1}^{(1)}$ belong to $\{D^{(1)}\}$ and are all different. Now assume all outer layers have already been merged by the scheme of summing up corresponding $b$ vectors mentioned above. Using $\{\hat{m}_s, \hat{K}^{(s)}, \hat{b}\}$ to represent the new set of parameters. Then $i_s \in [\hat{m}_s]$ with $\hat{m}_s \leq P_s$. The expressions (14) and (15) can be matched by setting $u_{j_n j_{n-1} \cdots j_1}^{111\cdots 1} = \hat{b}_{i_n i_{n-1} \cdots i_1}^{11\cdots 1}$ for $(j_s, i_s)$ pairs satisfying $D_{j_s}^{(s)} = \hat{K}_{i_s}^{(s)}$, and setting to zero if $D_{j_s}^{(s)} \notin [\hat{K}^{(s)}]$. This completes our first direction proof.

We now assume that there exists $\{u_{j_n j_{n-1} \cdots j_1}^{c_n c_{n-1} \cdots c_1}\}$ which satisfies problem (6), our goal is to find $(n+1)$-layer NN weights $W_1, W_2, \cdots, W_{n+1}$ satisfying (5). Since our $\{u_{j_n j_{n-1} \cdots j_1}^{c_n c_{n-1} \cdots c_1}\}$ satisfies all plane arrangement constraints in (6), we thus have, based on the inner most layer's activation pattern constraint and splits $\{u_{j_n j_{n-1} \cdots j_1}^{c_n c_{n-1} \cdots c_1}\}$ into $u_{j_n j_{n-1} \cdots j_1}^{c_{n-1} c_{n-2} \cdots c_1} := u_{j_n j_{n-1} \cdots j_1}^{[c_n=1]c_{n-1}c_{n-2}\cdots c_1}, u'_{j_n j_{n-1} \cdots j_1}^{c_{n-1} c_{n-2} \cdots c_1} := u_{j_n j_{n-1} \cdots j_1}^{[c_n=2]c_{n-1}c_{n-2}\cdots c_1}$,

$$y \odot \sum_{j_n=1}^{P_n} D_{j_n}^{(n)} \left( \sum_{j_{n-1}=1}^{P_{n-1}} D_{j_{n-1}}^{(n-1)} \left( \cdots \sum_{j_1=1}^{P_1} (X u_{j_n \cdots j_1}^{1\cdots 1})_+ - (X u'^{1\cdots 1}_{j_n \cdots j_1})_+ \cdots \right) - \right.$$
$$\left. \sum_{j_{n-1}=1}^{P_{n-1}} D_{j_{n-1}}^{(n-1)} \left( \cdots \sum_{j_1=1}^{P_1} (X u_{j_n \cdots j_1}^{2\cdots 1})_+ - (X u'^{2\cdots 1}_{j_n \cdots j_1})_+ \cdots \right) \right) \geq 1.$$

Thus we can find some $v_{j_n j_{n-1} \cdots j_2 j_1} \in \mathbb{R}^d, S_{j_1}^{(1)}, S_{j_2}^{(2)}, \cdots, S_{j_n}^{(n)} \in \{-1, 1\}$ such that

$$\sum_{j_n=1}^{P_n} D_{j_n}^{(n)} \left( \mathcal{T}_1^{(n-1)}(D^{(n-1)}) - \mathcal{T}_2^{(n-1)}(D^{(n-1)}) \right)$$

$$= \sum_{j_n=1}^{2P_n} \left( \sum_{j_{n-1}=1}^{2P_{n-1}} \left( \sum_{j_{n-2}=1}^{2P_{n-2}} \left( \cdots \sum_{j_2=1}^{2P_2} \left( \sum_{j_1=1}^{2P_1} (X v_{j_n j_{n-1} \cdots j_1})_+ + S_{j_1}^{(1)} \right)_+ \right. \right. \right. \tag{16}$$
$$\left. \left. \left. S_{j_2}^{(2)} \cdots \right)_+ S_{j_{n-2}}^{(n-2)} \right)_+ S_{j_{n-1}}^{(n-1)} \right)_+ S_{j_n}^{(n)}.$$

Though the process of finding $\{v_{j_n j_{n-1} \cdots j_1}, S_{j_i}^{(i)}\}$ has been outlined in three-layer case proof above and can be extended to $(n+1)$-layer case, we present here an outline of finding such

$\{v_{j_n j_{n-1} \cdots j_1}, S_{j_i}^{(i)}\}$ for five-layer NN for demonstration. For $n = 5$, we have the following

$$
\sum_{j_4=1}^{P_4} \left( \left( \sum_{j_3=1}^{P_3} \left( \sum_{j_2=1}^{P_2} \left( \sum_{j_1=1}^{P_1} (Xa_{j_4 j_3 j_2 j_1}^{111})_+ - (Xa_{j_4 j_3 j_2 j_1}^{'111})_+ \right)_+ - \left( \sum_{j_1=1}^{P_1} (Xa_{j_4 j_3 j_2 j_1}^{112})_+ - (Xa_{j_4 j_3 j_2 j_1}^{112})_+ \right) \right)_+ \right)_+
$$

$$
- \left( \sum_{j_2=1}^{P_2} \left( \sum_{j_1=1}^{P_1} (Xa_{j_4 j_3 j_2 j_1}^{121})_+ - (Xa_{j_4 j_3 j_2 j_1}^{'121})_+ \right)_+ - \left( \sum_{j_1=1}^{P_1} (Xa_{j_4 j_3 j_2 j_1}^{122})_+ - (Xa_{j_4 j_3 j_2 j_1}^{'122})_+ \right) \right)_+ \right)_+
$$

$$
- \left( \sum_{j_3=1}^{P_3} \left( \sum_{j_2=1}^{P_2} \left( \sum_{j_1=1}^{P_1} (Xa_{j_4 j_3 j_2 j_1}^{211})_+ - (Xa_{j_4 j_3 j_2 j_1}^{'211})_+ \right)_+ - \left( \sum_{j_1=1}^{P_1} (Xa_{j_4 j_3 j_2 j_1}^{212})_+ - (Xa_{j_4 j_3 j_2 j_1}^{'212})_+ \right) \right)_+ \right)_+
$$

$$
- \left( \sum_{j_2=1}^{P_2} \left( \sum_{j_1=1}^{P_1} (Xa_{j_4 j_3 j_2 j_1}^{221})_+ - (Xa_{j_4 j_3 j_2 j_1}^{'221})_+ \right)_+ - \left( \sum_{j_1=1}^{P_1} (Xa_{j_4 j_3 j_2 j_1}^{222})_+ - (Xa_{j_4 j_3 j_2 j_1}^{'222})_+ \right) \right)_+ \right)_+ \right).
$$

Let $v_{j_4 j_3 j_2 j_1}^{111} = \{a_{j_4 j_3 j_2 j_1}^{111}\} \cup \{a_{j_4 j_3 j_2 j_1}^{'111}\}$ and similarly construct $v_{j_4 j_3 j_2 j_1}^{112}, v_{j_4 j_3 j_2 j_1}^{121}, v_{j_4 j_3 j_2 j_1}^{122},$ $v_{j_4 j_3 j_2 j_1}^{211}, v_{j_4 j_3 j_2 j_1}^{212}, v_{j_4 j_3 j_2 j_1}^{221}, v_{j_4 j_3 j_2 j_1}^{222}$. Let $S_{j_1}^{111}, S_{j_1}^{112}, S_{j_1}^{121}, S_{j_1}^{122}, S_{j_1}^{211}, S_{j_1}^{212}, S_{j_1}^{221}, S_{j_1}^{222}$ to be 1 for $j_1 \in [P_1]$ and to be $-1$ for $j_1 \in [P_1 + 1, 2P_1]$. Thus, the above expression is equivalent to

$$
\sum_{j_4=1}^{P_4} \left( \left( \sum_{j_3=1}^{P_3} \left( \sum_{j_2=1}^{P_2} \left( \sum_{j_1=1}^{2P_1} (Xv_{j_4 j_3 j_2 j_1}^{111})_+ S_{j_1}^{111} \right) - \left( \sum_{j_1=1}^{2P_1} (Xv_{j_4 j_3 j_2 j_1}^{112})_+ S_{j_1}^{112} \right) \right)_+ \right)_+
$$

$$
- \left( \sum_{j_2=1}^{P_2} \left( \sum_{j_1=1}^{2P_1} (Xv_{j_4 j_3 j_2 j_1}^{121})_+ S_{j_1}^{121} \right) - \left( \sum_{j_1=1}^{2P_1} (Xv_{j_4 j_3 j_2 j_1}^{122})_+ S_{j_1}^{122} \right) \right)_+ \right)_+
$$

$$
- \left( \sum_{j_3=1}^{P_3} \left( \sum_{j_2=1}^{P_2} \left( \sum_{j_1=1}^{2P_1} (Xv_{j_4 j_3 j_2 j_1}^{211})_+ S_{j_1}^{211} \right) - \left( \sum_{j_1=1}^{2P_1} (Xv_{j_4 j_3 j_2 j_1}^{212})_+ S_{j_1}^{212} \right) \right)_+ \right)_+
$$

$$
- \left( \sum_{j_2=1}^{P_2} \left( \sum_{j_1=1}^{2P_1} (Xv_{j_4 j_3 j_2 j_1}^{221})_+ S_{j_1}^{221} \right) - \left( \sum_{j_1=1}^{2P_1} (Xv_{j_4 j_3 j_2 j_1}^{222})_+ S_{j_1}^{222} \right) \right)_+ \right)_+ \right).
$$

Let $v_{j_4 j_3 j_2 j_1}^{11} = \{v_{j_4 j_3 j_2 j_1}^{111}\} \cup \{v_{j_4 j_3 j_2 j_1}^{112}\}$ and $\tilde{S}_{j_1}^{11} = S_{j_1}^{111}$. Let $S_{j_2}^{11} = 1$ for $j_2 \in [P_2]$ and $S_{j_2}^{11} = -1$ for $j_2 \in [P_2 + 1, 2P_2]$. Similarly construct $v_{j_4 j_3 j_2 j_1}^{12}, \tilde{S}_{j_1}^{12}, S_{j_2}^{12}, v_{j_4 j_3 j_2 j_1}^{21}, \tilde{S}_{j_1}^{21}, S_{j_2}^{21},$ $v_{j_4 j_3 j_2 j_1}^{22}, \tilde{S}_{j_1}^{22}, S_{j_2}^{22}$. Thus the above expression is equivalent to

$$
\sum_{j_4=1}^{P_4} \left( \left( \sum_{j_3=1}^{P_3} \left( \sum_{j_2=1}^{2P_2} \left( \sum_{j_1=1}^{2P_1} (Xv_{j_4 j_3 j_2 j_1}^{11})_+ \tilde{S}_{j_1}^{11} \right)_+ S_{j_2}^{11} \right) - \left( \sum_{j_2=1}^{2P_2} \left( \sum_{j_1=1}^{2P_1} (Xv_{j_4 j_3 j_2 j_1}^{12})_+ \tilde{S}_{j_1}^{12} \right)_+ S_{j_2}^{12} \right) \right)_+ \right)_+
$$

$$
\left( \sum_{j_3=1}^{P_3} \left( \sum_{j_2=1}^{2P_2} \left( \sum_{j_1=1}^{2P_1} (Xv_{j_4 j_3 j_2 j_1}^{21})_+ \tilde{S}_{j_1}^{21} \right)_+ S_{j_2}^{21} \right) - \left( \sum_{j_2=1}^{2P_2} \left( \sum_{j_1=1}^{2P_1} (Xv_{j_4 j_3 j_2 j_1}^{22})_+ \tilde{S}_{j_1}^{22} \right)_+ S_{j_2}^{22} \right) \right)_+ \right).
$$

Let $v_{j_4 j_3 j_2 j_1}^1 = \{v_{j_4 j_3 j_2 j_1}^{11}\} \cup \{v_{j_4 j_3 j_2 j_1}^{12}\}$, $v_{j_4 j_3 j_2 j_1}^2 = \{v_{j_4 j_3 j_2 j_1}^{21}\} \cup \{v_{j_4 j_3 j_2 j_1}^{22}\}$. Let $\hat{\tilde{S}}_{j_1}^{(1)} = \tilde{S}_{j_1}^{11}$, $\hat{\tilde{S}}_{j_1}^{(2)} = \tilde{S}_{j_1}^{21}$. Let $\hat{S}_{j_2}^{(1)} = S_{j_2}^{11}, \hat{S}_{j_2}^{(2)} = S_{j_2}^{21}$. Let further $S_{j_3}^{(1)}, S_{j_3}^{(2)}$ to take value 1 for $j_3 \in [P_3]$ and

take value $-1$ for $j_3 \in [P_3 + 1, 2P_3]$. Therefore the above expression is equivalent to

$$\sum_{j_4=1}^{P_4} \left( \left( \left( \sum_{j_3=1}^{2P_3} \left( \sum_{j_2=1}^{2P_2} \left( \sum_{j_1=1}^{2P_1} (Xv^1_{j_4 j_3 j_2 j_1}) + \hat{\tilde{S}}^{(1)}_{j_1} \right)_+ \hat{S}^{(1)}_{j_2} \right)_+ S^{(1)}_{j_3} \right)_+ \right. \right.$$
$$\left. \left. - \left( \sum_{j_3=1}^{2P_3} \left( \sum_{j_2=1}^{2P_2} \left( \sum_{j_1=1}^{2P_1} (Xv^2_{j_4 j_3 j_2 j_1}) + \hat{\tilde{S}}^{(2)}_{j_1} \right)_+ \hat{S}^{(2)}_{j_2} \right)_+ S^{(2)}_{j_3} \right)_+ \right) \right).$$

We once again repeat the above procedure for the outer most layer and we arrive

$$\sum_{j_4=1}^{P_4} \left( \sum_{j_3=1}^{2P_3} \left( \sum_{j_2=1}^{2P_2} \left( \sum_{j_1=1}^{2P_1} (Xv^1_{j_4 j_3 j_2 j_1}) + \hat{\tilde{S}}^{(1)}_{j_1} \right)_+ \hat{S}^{(1)}_{j_2} \right)_+ S^{(1)}_{j_3} \right)_+ S'_{j_4},$$

where $S'_{j_4} = 1$ for $j_4 \in [P_4]$ and $-1$ otherwise. This completes our construction of $\{v_{j_n j_{n-1} \cdots j_1}, S^{(i)}_{j_i}\}$ for $n = 5$. Now, we are left with matching the following two expressions

$$\sum_{j_n=1}^{2P_n} \left( \sum_{j_{n-1}=1}^{2P_{n-1}} \left( \sum_{j_{n-2}=1}^{2P_{n-2}} \left( \cdots \sum_{j_2=1}^{2P_2} \left( \sum_{j_1=1}^{2P_1} (Xv_{j_n j_{n-1} \cdots j_1}) + S^{(1)}_{j_1} \right)_+ S^{(2)}_{j_2} \cdots \right)_+ S^{(n-2)}_{j_{n-2}} \right)_+ S^{(n-1)}_{j_{n-1}} \right)_+ S^{(n)}_{j_n},$$

and

$$\sum_{i_n=1}^{m_n} \left( \sum_{i_{n-1}=1}^{m_{n-1}} \left( \sum_{i_{n-2}=1}^{m_{n-2}} \left( \cdots \sum_{i_2=1}^{m_2} \left( \sum_{i_1=1}^{m_1} (XW_{1 i_1}) + W_{2 i_2 i_1} \right)_+ W_{3 i_3 i_2} \cdots \right)_+ W_{(n-1) i_{n-1} i_{n-2}} \right)_+ W_{n i_n i_{n-1}} \right)_+ W_{(n+1) i_n}.$$

This can be done by setting all weights corresponding to indices $\{i_n > 2P_n, i_{n-1} > 4P_n P_{n-1}, \cdots, i_k > \Pi^n_{c=k} 2P_c\}$ to be all zeros, then for the outer most layer, we set $W_{(n+1) i_n} = S^{(n)}_{i_n}$, for $2 \leq k \leq n$, we set $W_{k i_k i_{k-1}} = S^{(k-1)}_{((i_{k-1}-1)\%2P_{k-1})+1}$ for $i_{k-1} \in [(i_k-1)*2P_{k-1}+1, i_k * 2P_{k-1}]$ and $0$ otherwise. For the inner most layer, we set $W_{1 i_1} = v_{j_n j_{n-1} \cdots j_1}$ with

$$j_n = \lfloor (i_1 - 1)/\Pi^{n-1}_{k=1} 2P_k \rfloor + 1,$$
$$j_{n-1} = \lfloor ((i_1 - 1)\%\Pi^{n-1}_{k=1} 2P_k)/\Pi^{n-2}_{k=1} 2P_k \rfloor + 1,$$
$$j_{n-2} = \lfloor (((i_1 - 1)\%\Pi^{n-1}_{k=1} 2P_k)\%\Pi^{n-2}_{k=1} 2P_k)/\Pi^{n-3}_{k=1} 2P_k \rfloor + 1,$$
$$\cdots$$
$$j_2 = \lfloor ((\cdots)\%\Pi^2_{k=1} 2P_k)/2P_1 \rfloor + 1,$$
$$j_1 = ((\cdots)\%\Pi^2_{k=1} 2P_k)\%2P_1 + 1.$$

Given any test point $\tilde{x} \in \mathbb{R}^d$, the final prediction can be computed by

$$\tilde{y} = \sum_{j_n=1}^{P_n} \left( \mathcal{T}^{(n-1)}_1 (D^{(n-1)}) \right)_+ - \left( \mathcal{T}^{(n-1)}_2 (D^{(n-1)}) \right)_+, \tag{17}$$

where

$$\mathcal{T}^{(n-1)\cdots(i)}_{c_{n-1}\cdots c_i} (D^{(i)}) = \sum_{j_i=1}^{P_i} \left( \mathcal{T}^{(n-1)\cdots(i)(i-1)}_{c_{n-1}\cdots c_i 1} (D^{(i-1)}) \right)_+ - \left( \mathcal{T}^{(n-1)\cdots(i)(i-1)}_{c_{n-1}\cdots c_i 2} (D^{(i-1)}) \right)_+,$$

$$\mathcal{T}^{(n-1)(n-2)\cdots(1)}_{c_{n-1} c_{n-2}\cdots c_1} (D^{(1)}) = \sum_{j_1=1}^{P_1} \left( \tilde{x}^T a^{1 c_{n-1}\cdots c_1}_{j_n j_{n-1}\cdots j_1} \right)_+ - \left( \tilde{x}^T a^{2 c_{n-1}\cdots c_1}_{j_n j_{n-1}\cdots j_1} \right)_+.$$

### F.3 EXPLANATION OF BIAS TERM FOR GENERAL CASE

To show that the $n$-layer NN of form $((\cdots((XW_1)_+W_2)_+W_3\cdots)_+W_{n-1})_+W_n$ preserves the same approximation capacity as the biased version $((\cdots((XW_1 + b_1)_+W_2 + b_2)_+W_3 + b_3\cdots)_+W_{n-1} + b_{n-1})_+W_n + b_n$, we show that the biased version is incorporated in the form $((\cdots((XW_1)_+W_2)_+W_3\cdots)_+W_{n-1})_+W_n$ when the constraint on number of hidden neurons is mild. First note that we can always append the data matrix $X$ with a column of ones to incorporate the inner most bias $b_1$. Then for each outer layer, we can always have an inner neuron to be a pure bias neuron with value one. Then the corresponding outer neuron weight would serve as an outer layer bias.

### F.4 ACTIVATION PATTERN SUBSAMPLING AND ITERATIVE FILTERING.

Here we detail more about our hyperplane selection scheme. Take two-layer ReLU model for example, in order to find the activation pattern $\mathcal{D}$ corresponding to the hidden layer, one needs to exhaust the set $\{\mathrm{diag}(\mathbb{1}\{Xu \geq 0\})\}$ for all $u \in \mathbb{R}^d$. In our experiments, we adopt a heuristic subsampling procedure, i.e., we usually set a moderate number of hidden neurons $m_1$, and we sample a set of Gaussian random vectors $\{u_1, u_2, \cdots, u_{n_1}\}$ with some random $n_1 > m_1$. Then we take the set $\{\mathrm{diag}(\mathbb{1}\{Xu_i \geq 0\})|i \in [n_1]\}$. If $|\{\mathrm{diag}(\mathbb{1}\{Xu_i \geq 0\})|i \in [n_1]\}| > m_1$, we take a subset of $m_1$ activation patterns there, if not, we increase $n_1$ and redo all prior steps until we hit some satisfactory $n_1$. This heuristic method always works well in our experiments, see Appendix H.1 for more about our implementation details.

A more rigorous way which exhausts all possible activation patterns can be done via an iterative filtering procedure. We demonstrate here for a toy example. Consider still two-layer model as before, when we are given a data set $X$ of size $n$, a loose upper bound on $|\mathcal{D}|$ is given by $2^n$, i.e., each piece of data can take either positive and negative values and they are all independent. Thus one can find all possible $D_i$'s by solving

$$\mathrm{find} \qquad \sum_{i}^{2^n} \|u_i\|_2$$

$$\mathrm{s.t.} \qquad (2D_i - I)Xu_i \geq 0, \|u_i\|_2 \leq 1$$

where $D_i$'s loop over all $2^n$ possibilities. Then the $D_i$'s correspond to non-zero $u_i$'s in the solution are feasible plane arrangements. However, this method induces $2^n * d$ variables. A more economic way to find all feasible arrangements is to do an iterative filtering with each newly added data. When there is only one non-zero data point $X_1$, there always exists $u_1, u_2$ vectors such that $X_1^T u_1 > 0$ and $X_1^T u_2 < 0$. Thus $\mathcal{D}_1 = \{[1], [0]\}$ represent all possible sign patterns for this single training data. After the second data point has been added, we know

$$\mathcal{D}_2 \subseteq \left\{ \begin{bmatrix} 1 & 0 \\ 0 & 1 \end{bmatrix}, \begin{bmatrix} 1 & 0 \\ 0 & 0 \end{bmatrix}, \begin{bmatrix} 0 & 0 \\ 0 & 1 \end{bmatrix}, \begin{bmatrix} 0 & 0 \\ 0 & 0 \end{bmatrix} \right\}.$$

Therefore, an upper bound on cardinality of $\mathcal{D}_2$ is given by $2^2 = 4$. However, this upper bound might be pessimistic, for example, if $X_1 = X_2$, then we would expect

$$\mathcal{D}_2 = \left\{ \begin{bmatrix} 1 & 0 \\ 0 & 1 \end{bmatrix}, \begin{bmatrix} 0 & 0 \\ 0 & 0 \end{bmatrix} \right\} \subset \left\{ \begin{bmatrix} 1 & 0 \\ 0 & 1 \end{bmatrix}, \begin{bmatrix} 1 & 0 \\ 0 & 0 \end{bmatrix}, \begin{bmatrix} 0 & 0 \\ 0 & 1 \end{bmatrix}, \begin{bmatrix} 0 & 0 \\ 0 & 0 \end{bmatrix} \right\}.$$

Since $X_1^T u = X_2^T u$ for any $u$, they will always have the same sign pattern. Things get more complicated with more data points, say for $n$ data points $X_1, X_2, \cdots, X_n$, it is possible that sign pattern of $X_n^T u$ can be determined by sign patterns of $X_1^T u, X_2^T u, \cdots, X_{n-1}^T u$ when there are linear dependency between the data points, which happens more often with larger set of training data. Thus the true cardinality of $\mathcal{D}_n$ might be far smaller than $2^n$. Indeed, one can show that, with $r$ denoting rank of the training data matrix consisting of the first $n$ data points (Pilanci & Ergen, 2020; Stanley et al., 2004),

$$|\mathcal{D}_n| \leq 2r \left( \frac{e(n-1)}{r} \right)^r,$$

which can be much smaller than our pessimistic bound $2^n$ especially when training data has small rank. To stay close to the optimal cardinality and avoid solving an optimization problem with $2^n * d$

number of variables, what one can do is to find the feasible plane arrangements iteratively at the time when each single data point is added. The underline logic is that plane arrangement patterns which are infeasible to $X_{[1:t-1]}$ will also be infeasible to $X_{[1:t]}$. Therefore, assume one has already found the optimal sign patterns $\mathcal{D}_{t-1}$ for the first $t-1$ training samples which has cardinality $c_{t-1}$, when $X_t$ arrives, one needs to solve the following auxiliary problem

$$\text{find} \quad \sum_i^{2c_{t-1}} \|u_i\|_2$$

$$\text{s.t.} \quad \left(2\begin{bmatrix} D_{(t-1)i} & 0 \\ 0 & 0 \end{bmatrix} - I\right) X u_i \geq 0, i \in [1, \cdots, c_{t-1}]$$

$$\left(2\begin{bmatrix} D_{(t-1)i-c_{t-1}} & 0 \\ 0 & 1 \end{bmatrix} - I\right) X u_i \geq 0, i \in [c_{t-1}+1, \cdots, 2c_{t-1}]$$

$$\|u_i\|_2 \leq 1,$$

which has only $(2*c_{t-1})*d$ number of variables and can be far fewer than $2^t*d$. This scheme can also be done lazily each fixed $T$ iterations, one just add all $\{0,1\}$-patterns for the last $T$ data points.

# G   DEFERRED PROOFS AND EXTENSIONS IN SECTION 6

## G.1   DEFERRED PROOF OF THEOREM 6.1

*Proof.* Notice that the polyhedron cut is simply three consecutive cuts with three hyperplanes:

- $\mathcal{H}_1 := \{\theta : y_n \cdot f^{\text{two-layer}}(x_n^T; \theta) = 0\}$;

- $\mathcal{H}_2 := \{\theta^e : (2D(S_i) - I)_n x_n^T \theta^e = 0\}$;

- $\mathcal{H}_3 := \{\theta^o : (2D(S_i) - I)_n x_n^T \theta^o = 0\}$,

where $\theta^o$ ($\theta^e$) denotes the reduced $\theta$ vector containing only the odd (even) indices. Since the cuts imposed by the linear inequality constraints $\mathcal{C}_n$ and $\mathcal{C}'_n$ via hyper-planes $\mathcal{H}_2, \mathcal{H}_3$ only reduce the remaining set $\mathcal{T}$, examining the volume remained after just the cut via $\mathcal{H}_1$ suffices for the convergence analysis as it bounds $\textbf{vol}(\mathcal{T}_1)$ from above.

Assuming the cut is active, it follows that $\theta_G$ misclassifies $(x_n, y_n)$, so $y_n \cdot f^{\text{two-layer}}(x_n; \theta) < 0$. Thus, the cut is deep and $\theta_G$ is in the interior of $\mathcal{T}_2$. By Proposition 1, it follows that

$$\textbf{vol}(\mathcal{T} \cap \mathcal{H}_1^+) < \textbf{vol}(\mathcal{T} \cap \mathcal{H}_G^+) \leq (1 - 1/e)\textbf{vol}(\mathcal{T}),$$

where $\mathcal{H}_G$ is any hyperplane that goes through $\theta_G$ which is parallel to $\mathcal{H}_1$. Therefore, at each step $t$ where the cut is active, at least volume of magnitude $\frac{1}{e}\textbf{vol}(\mathcal{T}^t)$ is cut away. The volume of $\mathcal{T}^t$ after $t$ iterations is bounded by:

$$\textbf{vol}(\mathcal{T}^t) < (1 - 1/e)^t \cdot \textbf{vol}(\mathcal{T}^0).$$

Then as $t \to \infty$, $(1 - 1/e)^t \to 0$, it follows that $\textbf{vol}(\mathcal{T}^t)$ converges to zero and the feasible region shrinks to point(s).

Notice that since $\mathcal{T}$ is a convex body and $\{\mathcal{T}^t\}$ is a nested decreasing sequence of convex set with $\mathcal{T}^{t+1} \subseteq \mathcal{T}^t$, by the finite intersection property, the intersection of all $\mathcal{T}^t$ sets is non-empty and contains the optimal solution $\theta^*$:

$$\cap_{t=0}^{\infty} \mathcal{T}^t = \{\theta^*\}.$$

It remains to justify that what the intersection contains is indeed the optimal solution. To see this, observe that the problem that Algorithm 4 is simply a feasibility problem. Since every time the convex set shrinks by intersecting with the constraint set containing feasibility criteria given each new acquired data point $(x_{n_t}, y_{n_t})$, i.e.,

$$\{\theta \in \mathcal{T} : y_n \cdot f^{\text{two-layer}}(x_n; \theta) \geq 0, \mathcal{C}(\{n\}), \mathcal{C}'(\{n\})\},$$

the set shrinks to a finer set that satisfies increasingly more constraints posed by additional acquired data points. This monotonic improvement with the shrinking feasible region implies that the sequence of classifiers $\theta^t$ converges to the set of optimal classifiers $\{\theta^*\}$. □

### G.2 EXTENSION TO ALGORITHM 4: THE CASE OF INEXACT CUTS AND CONVERGENCE

In this section, we discuss convergence results of Algorithm 5 under the third setup described in Section 6.1.

Note that the cut made by both Algorithm 4 and Algorithm 2 are *exact* in the sense that cuts are only made when the computed center mis-classifies the data points returned by the oracle and is therefore always discarded via the cutting hyperplanes. Algorithm 5, on the other hand, implements *inexact* cut: cuts are always made regardless of whether the queried center mis-classifies. Nice convergence rate similar to that of Theorem 6.1 can still be guaranteed, as we shall see in the following theorem, as long as the cut is made within a certain neighborhood of the centroid at each step.

First, let us quantify the inexactness of the cut. At each iteration, given a cutting hyperplane $\mathcal{H}_a$ of normal vector $a$, which passes through the origin by design, and the computed center $\theta$, the Euclidean distance between $\theta$ and the hyperplane is given by

$$h = \|\theta\|_2 |\cos(\alpha)|, \tag{18}$$

where

$$\alpha = \arccos(\frac{\theta \cdot a}{\|\theta\|_2 \|a\|_2}) \tag{19}$$

is the angle between $\theta$ and the normal vector $a$. For simple notation, let us denote the angle between $\theta$ and the cutting plane $\mathcal{H}_a$ as $\beta$. Then

$$\beta = \frac{\pi}{2} - \alpha. \tag{20}$$

Next, we introduce an important extension to Proposition 1 given in Louche & Ralaivola (2015).

**Proposition 2** (Generalized Partition of Convex bodies)**.** *Let $\mathcal{T} \subseteq \mathbb{R}^d$ be a convex body. Let $\mathcal{H}_a$ be a hyperplane of normal vector $a$. We define the positive (negative) halfspace $\mathcal{T}^+$ (resp. $\mathcal{T}^-$) of $\mathcal{T}$ with respect to $\mathcal{H}_a$ as*

$$\mathcal{T}^+ := \mathcal{T} \cap \{\theta \in \mathbb{R}^d : \langle a, \theta \rangle \geq 0\}$$
$$\mathcal{T}^- := \mathcal{T} \cap \{\theta \in \mathbb{R}^d : \langle a, \theta \rangle < 0\}$$

*The following holds true: if $\theta_G + \Lambda a \in \mathcal{T}^+$ then*

$$\boldsymbol{vol}(\mathcal{T}^+)/\boldsymbol{vol}(\mathcal{T}) \geq e^{-1}(1 - \lambda)^d,$$

*where*

$$\Lambda = \lambda \Theta_d \frac{\boldsymbol{vol}(\mathcal{T}) H_{\mathcal{T}^+}}{R^d H_{\mathcal{T}^-}},$$

*with $\lambda \in \mathbb{R}$ an arbitrary real such that $\lambda \leq 1$, $\Theta_d$ a constant depending only on $d$, $R$ the radius of the $(d-1)$-dimensional ball $\mathcal{B}_2$ of volume $\boldsymbol{vol}(\mathcal{B}_2) := \boldsymbol{vol}(\mathcal{T} \cap \{\theta \in \mathbb{R}^d : \langle a, \theta \rangle = 0\})$ and*

$$H_{\mathcal{T}^+} := \max_{b \in \mathcal{T}^+} b^T a \quad (resp. \ H_{\mathcal{T}^-} := \min_{b \in \mathcal{T}^-} b^T a).$$

Note that by symmetry of partitioned convex body with respect to the centroid (an application of Proposition 1), one can similarly establish a bound in the case that

$$\theta_G + \Lambda a \in \mathcal{T}^-,$$

where $\Lambda$ is defined as in Proposition 2. Then

$$\mathbf{vol}(\mathcal{T}^-)/\mathbf{vol}(\mathcal{T}) \geq e^{-1}(1 - \lambda)^d. \tag{21}$$

Proposition 2 generalizes Grünbaum's inequality in Proposition 1 by allowing a cut that is of distance (greater than or equal to) $\Lambda$ along the direction of the normal vector $a$ from the actual center of gravity. To apply Proposition 2, we need to establish an equivalence relation between our quantifiation of the inexactness through the Euclidean distance $h$ (or angle $\alpha$ and or $\beta$) with $\Lambda$.

We are now ready to bound the volume reduction factor for inexact cuts with respect to the center gravity in Algorithm 5.

**Theorem G.1** (Convergence with Inexact Cuts of Center of Gravity). *Let $\mathcal{T} \subseteq \mathbb{R}^d$ be a convex body and let $\theta_G$ denote its center of gravity. Given oracle returned data point $(x_n, y_n)$, define hyperplane*

$$\mathcal{H}_a := \{\theta \in \mathbb{R}^d : y_n \cdot f^{\text{two-layer}}(x_n; \theta) = 0\},$$

*where*

$$a := y_n \cdot [x_n^1 - x_n^1 \ ... \ x_n^P - x_n^P]$$

*is its associated normal vector. Then $\mathcal{H}_a$ divides $\mathcal{T}$ into a positive half-space $\mathcal{T}^+$ and a negative half-space $\mathcal{T}^-$,*

$$\mathcal{T}^+ := \mathcal{T} \cap \{\theta \in \mathbb{R}^d : \langle a, \theta \rangle \geq 0\}$$
$$\mathcal{T}^- := \mathcal{T} \cap \{\theta \in \mathbb{R}^d : \langle a, \theta \rangle < 0\},$$

*and $h := \frac{|\theta_G^T a|}{\|a\|_2}$ is the Euclidean distance between $\theta_G$ and $\mathcal{H}_a$. The inexact polyhedron cut given in Algorithm 5 partitions the convex body $\mathcal{T}$ into two subsets:*

$$\mathcal{T}_1 := \{\theta \in \mathcal{T} : \mathcal{T}^+, \mathcal{C}(\{n\}), \mathcal{C}'(\{n\})\}$$
$$\mathcal{T}_2 := \{\theta \in \mathcal{T} : \mathcal{T}^-, \text{ or } \neg\mathcal{C}(\{n\}), \neg\mathcal{C}'(\{n\})\},$$

*where $\neg$ denotes the complement of a given set. The following holds:*

$$\boldsymbol{vol}(\mathcal{T}_1) < (1 - e^{-1}(1 - \tilde{\lambda})^d)\boldsymbol{vol}(\mathcal{T}),$$

*with*

$$\tilde{\lambda} = -\frac{R^d H_{\mathcal{T}^-}}{\Theta_d \boldsymbol{vol}(\mathcal{T}) H_{\mathcal{T}^+}} h > 0,$$

*where $\Theta_d$ is a constant depending only on $d$, $R$ is the radius of the $(d-1)$-dimensional ball $\mathcal{B}_2$ of volume $\boldsymbol{vol}(\mathcal{B}_2) := \boldsymbol{vol}(\mathcal{T} \cap \mathcal{H}_a)$ and*

$$H_{\mathcal{T}^+} := \max_{\theta \in \mathcal{T}^+} \theta^T a \quad (\text{resp. } H_{\mathcal{T}^-} := \min_{\theta \in \mathcal{T}^-} \theta^T a).$$

*Proof.* Observe that at each iteration if $\theta_G$ mis-classifies the oracle returned data point $(x_n, y_n)$, then we have the same analysis as in Theorem 6.1. In such a case, we have guaranteed accelerated cuts with a strictly better volume reduction rate bound than in the case when cuts are made exactly through the centroid. This is because in this case the centroid is contained in the interior of the eliminated region. On the other hand, when the cut is made such that the centroid is contained in the interior of the positive half-space, we cut away smaller volume than when we make the cut through the centroid. Therefore, to obtain a lower bound on the volume reduction rate for Algorithm 5, it suffices to examine only the latter case, i.e. when $\theta_G$ is contained in interior of the positive half-space.

Suppose that $\theta_G \in \mathcal{T}^+ \setminus \mathcal{H}_a$. Define real number $\Lambda$ to be such that

$$\Lambda < -h := -\frac{|\theta^T a|}{\|a\|_2},$$

then since $\theta_G - ha \in \mathcal{H}_a$, it follows that

$$\theta_G + \Lambda a \in \mathcal{T}^-.$$

By Proposition 2 and symmetry, it follows that

$$\boldsymbol{vol}(\mathcal{T}^-)/\boldsymbol{vol}(\mathcal{T}) \geq e^{-1}(1 - \lambda)^d,$$

where

$$\lambda = \Lambda(\Theta_d \frac{\boldsymbol{vol}(\mathcal{T}) H_{\mathcal{T}^+}}{R^d H_{\mathcal{T}^-}})^{-1}$$

is a real such that $\lambda \leq 1$, $\Theta_d$ is a constant depending only on $d$, $R$ is the radius of the $(d-1)$-dimensional ball $\mathcal{B}_2$ of volume $\boldsymbol{vol}(\mathcal{B}_2) := \boldsymbol{vol}(\mathcal{T} \cap \mathcal{H}_a)$ and

$$H_{\mathcal{T}^+} := \max_{\theta \in \mathcal{T}^+} \theta^T a \quad (\text{resp. } H_{\mathcal{T}^-} := \min_{\theta \in \mathcal{T}^-} \theta^T a).$$

Since

$$\Lambda = \lambda \Theta_d \frac{\mathbf{vol}(\mathcal{T})H_{\mathcal{T}^+}}{R^d H_{\mathcal{T}^-}} < -h,$$

or equivalently

$$0 < \lambda < -\frac{R^d H_{\mathcal{T}^-}}{\Theta_d \mathbf{vol}(\mathcal{T})H_{\mathcal{T}^+}}h,$$

it follows that

$$\mathbf{vol}(\mathcal{T}^-) \geq e^{-1}(1-\lambda)^d \mathbf{vol}(\mathcal{T}) > e^{-1}(1-\tilde{\lambda})^d \mathbf{vol}(\mathcal{T}),$$

or equivalently

$$\mathbf{vol}(\mathcal{T}^+) < (1 - e^{-1}(1-\tilde{\lambda})^d)\mathbf{vol}(\mathcal{T}),$$

where

$$\tilde{\lambda} = -\frac{R^d H_{\mathcal{T}^-}}{\Theta_d \mathbf{vol}(\mathcal{T})H_{\mathcal{T}^+}}h.$$

Since

$$\mathcal{T}_1 := \{\theta \in \mathcal{T} : \mathcal{T}^+, \mathcal{C}(\{n\}), \mathcal{C}'(\{n\})\} \subseteq \mathcal{T}^+,$$

we have that

$$\mathbf{vol}(\mathcal{T}_1) \leq \mathbf{vol}(\mathcal{T}^+) < (1 - e^{-1}(1-\tilde{\lambda})^d)\mathbf{vol}(\mathcal{T}).$$

To justify that the derived ratio $\mathbf{vol}(\mathcal{T}_1)/\mathbf{vol}(\mathcal{T})$ is valid, i.e. $\mathbf{vol}(\mathcal{T}_1)/\mathbf{vol}(\mathcal{T}) \in [0,1]$, it suffices to show that $e^{-1}(1-\tilde{\lambda})^d \in [0,1]$, or $0 < \tilde{\lambda} \leq 1$.

Define coefficient

$$c(\mathcal{H}_a, \mathcal{T}, d) := \Theta_d \frac{\mathbf{vol}(\mathcal{T})H_{\mathcal{T}^+}}{R^d H_{\mathcal{T}^-}}.$$

Since by definition $H_{\mathcal{T}^+} > 0$, $H_{\mathcal{T}^-} < 0$, with $\Theta_d, \mathbf{vol}(\mathcal{T}), R^d > 0$, it follows that $c(\mathcal{H}_a, \mathcal{T}, d) < 0$.

Since $\tilde{\lambda} := -c(\mathcal{H}_a, \mathcal{T}, d)^{-1}h$ and $h > 0$, we have that $\tilde{\lambda} > 0$. Next, to see $\tilde{\lambda} \leq 1$, observe that since we have

$$\Lambda = \lambda c(\mathcal{H}_a, \mathcal{T}, d) < -h,$$

it follows that

$$c(\mathcal{H}_a, \mathcal{T}, d) < -h \leq -\lambda^{-1}h,$$

where we have used the fact that $0 < \lambda \leq 1$. It follows that

$$\tilde{\lambda} := -c(\mathcal{H}_a, \mathcal{T}, d)^{-1}h < h^{-1}h = 1.$$

Hence, $0 < \tilde{\lambda} < 1$. Then by Theorem 6.1, the volume reduction ratio $1 - e^{-1}(1-\tilde{\lambda})^d > 1 - e^{-1}$ offers a strict upper bound for each iteration of Algorithm 5, regardless of whether the centroid $\theta_G$ is contained in the positive half-space or the negative half-space. $\square$

Given Theorem G.1, at each step $t$ in Algorithm 5, at least a volume of magnitude $e^{-1}(1 - \tilde{\lambda})^d \mathbf{vol}(\mathcal{T}^t)$ is cut away. The volume of $\mathcal{T}^t$ after $t$ iterations is bounded by:

$$\mathbf{vol}(\mathcal{T}^t) < (1 - e^{-1}(1-\tilde{\lambda})^d)^t \cdot \mathbf{vol}(\mathcal{T}^0).$$

Since $0 < \tilde{\lambda} < 1$, we have $0 < 1 - e^{-1}(1-\tilde{\lambda})^d \leq 1$. So as $t \to \infty$, $(1 - e^{-1}(1-\tilde{\lambda})^d)^t \to 0$. It follows that $\mathbf{vol}(\mathcal{T}^t)$ converges to zero as $t \to \infty$, and the feasible region shrinks to the optimal point(s), as in Theorem 6.1.

Theorem G.1 allows us to quantify the volume reduction rate as a function of the inexactness of cuts through the Euclidean distance between the centroid and the cutting hyperplane, which is monitored by the query sampling method. In particular, in the case of inexact cuts in Algorithm 5 with a minimal margin query sampling scheme, the volume reduction rate converges asymptotically in the number of training data supplied to the cutting-plane oracle to the rate in Proposition 1. This result is stated in the following corollary.

**Corollary 1** (Asymptotic Convergence of Inexact Cuts with Center of Gravity). *Assume that data points $x_i$'s are uniformly distributed across input space. Let the number of training data supplied to the cutting-pane oracle in Algorithm 5 goes to infinity, i.e. $|\mathcal{D}| \to \infty$. Then the volume of $\mathcal{T}^t$ after $t$ iterations with the polyhedron cut in Algorithm 5 under minimal margin query sampling method converges asymptotically to bound*

$$\boldsymbol{vol}(\mathcal{T}^t) < (1 - e^{-1})^t \cdot \boldsymbol{vol}(\mathcal{T}^0).$$

*Proof.* Let us begin with analyzing a single iteration with convex body $\mathcal{T} \subset \mathbb{R}^d$ and its centroid $\theta_G$. Notice that as $|\mathcal{D}|$ increases, data points $x_i$'s become more densely distributed across the input space. So for any $\epsilon > 0$, there exists a sufficiently large $N$ such that for all $|\mathcal{D}| > N$, there exists at least one data point $x_i \in \mathcal{D}$, for which $|f^{\text{two-layer}}(x_i; \theta_G) := [x_i^1 - x_i^1 \; ... \; x_i^P - x_i^P] \cdot \theta_G| < \epsilon$. As $|\mathcal{D}| \to \infty$, $\epsilon$ can be made arbitrarily small, implying that

$$\arg\min_{x_i \in \mathcal{D}_x} |f^{\text{two-layer}}(x_i; \theta_G)| \to 0.$$

This suggests that the Euclidean distance between $\theta_G$ and the oracle returned cutting plane $\mathcal{H}_a$ diminishes to 0 as $|\mathcal{D}| \to \infty$.

By Theorem G.1, given an inexact polyhedron cut in Algorithm 5 distance $h$ away from the centroid, the volume of the remaining set is upper bounded by the following:

$$\mathbf{vol}(\mathcal{T}_1) \leq (1 - e^{-1}(1 - \tilde{\lambda})^d)\mathbf{vol}(\mathcal{T}),$$

where $\tilde{\lambda} := -c(\mathcal{H}_a, \mathcal{T}, d)^{-1}h$ and

$$c(\mathcal{H}_a, \mathcal{T}, d) := \Theta_d \frac{\mathbf{vol}(\mathcal{T})H_{\mathcal{T}^+}}{R^d H_{\mathcal{T}^-}}.$$

Then to show that ratio $(1 - e^{-1}(1 - \tilde{\lambda})^d) \to 1 - e^{-1}$, or equivalently $\tilde{\lambda} \to 0$, as $h \to 0$, it suffices to show that the coefficient term $-c(\mathcal{H}_a, \mathcal{T}, d) = |c(\mathcal{H}_a, \mathcal{T}, d)|$ is lower bounded by a constant $M_{\min}$ so that its inverse is upper bounded. This is easy to show. First observe that dimension dependent constant $\Theta_d$ and $\mathbf{vol}(\mathcal{T})$ is fixed and finite regardless of the cutting plane $\mathcal{H}_a$. Because convex body $\mathcal{T}$ is by definition closed, variables $R, |H_{\mathcal{T}^-}|$ are bounded from above. By the same reasoning, $H_{\mathcal{T}^+}$ is bounded from below and away from 0, for if $H_{\mathcal{T}^+} = 0$, it follows that $\mathbf{vol}(\mathcal{T}^+) = 0$, suggesting the termination and convergence of the algorithm to the optimal solution(s). Hence, there exists a constant $M_{\min}$ such that

$$\min_{\mathcal{H}_{a_i}} |c(\mathcal{H}_{a_i}, \mathcal{T}, d)| \geq M_{\min}.$$

It follows that

$$0 \leq \lim_{h \to 0} \tilde{\lambda} := \lim_{h \to 0} -c(\mathcal{H}_a, \mathcal{T}, d)^{-1}h \leq \lim_{h \to 0} M_{\min}^{-1}h = 0.$$

So as $|\mathcal{D}| \to \infty$, we have that $h \to 0$, which results in $\tilde{\lambda} \to 0$. Therefore, the volume of $\mathcal{T}^t$ after $t$ iterations in Algorithm 5 follows:

$$\lim_{|\mathcal{D}| \to \infty} \mathbf{vol}(\mathcal{T}^t) < \lim_{|\mathcal{D}| \to \infty} (1 - e^{-1}(1 - \tilde{\lambda})^d)^t \cdot \mathbf{vol}(\mathcal{T}^0) = (1 - e^{-1})^t \cdot \mathbf{vol}(\mathcal{T}^0).$$

$\square$

### G.3 CONVERGENCE W.R.T. CENTER OF THE MAXIMUM VOLUME ELLIPSOID

Recall the definition for the center of MVE in Definition 4. We will now show that similar convergence rate as that of the center of gravity can be achieved under center of MVE as well.

**Theorem G.2** (Convergence with Center of MVE). *Let $\mathcal{T} \subseteq \mathbb{R}^d$ be a convex body and let $\theta_M$ denote its center of the maximum volume inscribed ellipsoid. The polyhedron cut given in Algorithm 4 (assuming that the cut is active) and Algorithm 2, i.e.,*

$$\mathcal{T} \cap \{\theta : y_n \cdot f^{\text{two-layer}}(x_n; \theta) \geq 0, \mathcal{C}(\{n\}), \mathcal{C}'(\{n\})\},$$

*where coupling $(x_n, y_n)$ is the data point returned by the cutting-plane oracle after receiving queried point $\theta_M$, partitions the convex body $\mathcal{T}$ into two subsets as in Theorem 6.1. Then $\mathcal{T}_1$ satisfies the following inequality:*

$$\boldsymbol{vol}(\mathcal{T}_2) < (1 - \frac{1}{d}) \cdot \boldsymbol{vol}(\mathcal{T}).$$

*Proof.* The proof follows similarly as the proof to Theorem 6.1. For the MVE cutting-plane method, the bound on the volume reduction factor is given by:

$$\frac{\mathbf{vol}(\mathcal{T}^{k+1})}{\mathbf{vol}(\mathcal{T}^{k})} \leq 1 - \frac{1}{d},$$

where $k$ denotes the iteration (Boyd & Vandenberghe, 2004). Since $f^{\text{two-layer}}(x_n; \theta)$ is linear in $\theta$ and by the design of the algorithm, the center $\theta_M$ is always contained in the interior of the discarded set (or equivalently, the cut is deep), proof for the bound $(1 - \frac{1}{d}) \cdot \mathbf{vol}(\mathcal{T})$ follows similarly as the proof of Theorem 6.1. $\qquad\square$

## H   Experiments Supplementals

### H.1   Implementation Details

In this section, we provide an overview of the implementation details. For specific aspects, such as baseline implementation and the cutting-plane AL method for regression, we direct readers to the corresponding sections in Appendix H.

In our experiments, we follow exactly the same algorithm workflow as in Algorithm 1 for our model training and Algorithm 4 for active learning with limited queries, which is the version we used conducting all of our experiments in Section 7. In the training of Algorithm 4, we implement the "center" function for analytic center retrieval due to its simple computation formula (see Definition 1). Since the center retrieval problem is of convex minimization form, we solve it with CVXPY (Diamond & Boyd, 2016) and default to MOSEK (MOSEK ApS, 2024) as our solver. In our experiments, MOSEK is able to handle all encountered convex optimization programs, given that they are feasible. We involve all training data cuts while we note that some cut dropping method may help alleviate computation overhead, i.e., one may keep the last several cuts only, see Section 2.5 in (Parshakova et al., 2023) for a discussion on constraint dropping. For activation pattern generation, for two-layer model experiment, say we want $P = m$ patterns, we always generate a set of standard Gaussian vectors $\{u_i, i \in [n]\}$ for some $n > m$, compute the induced pattern set $\{\mathcal{D}_i = \text{diag}(\mathbb{1}\{Xu_i \geq 0\})\}$, and take $m$ non-duplicate patterns out of it. Furthermore, in our implementation of Algorithm 4, we included a few additional checks for computational efficiency, like skipping the centering step when no cut has been performed before, or book-keeping of discarded data points that should not be re-considered immediately. For IMDB experiments, we randomly pick 50 training data points and 20 test data points for ease of computation, and we use minimal margin query strategy described in Appendix D.3. Through all our data experiments, random seed is fixed to be 0 except for error bar plots, where we always take random seeds $\{0, 1, 2, 3, 4\}$.

### H.2   Final Solve Regularization in Cutting-plane AL

In this section, we discuss an optional feature to our cutting-plane AL method via the two-or-three layer ReLU network: the inclusion of a final convex solve which solves for the optimal parameter of the equivalent convex program to the two-or-three layer ReLU network using the data acquired by the AL thus far.

**Convexifying a Two-layer ReLU Network.**   To introduce this final convex solve, we first briefly overview the work of Pilanci & Ergen (2020). We will focus on the two-layer ReLU network case, as the case for three-layer can be easily extended by referencing the exact convex reformulation of the three-layer ReLU network given in the paper by Ergen & Pilanci (2021b).

Pilanci & Ergen (2020) first introduced a finite dimensional, polynomial-size convex program that globally solves the training problem for two-layer fully connected ReLU networks. The convexification of ReLU networks can be summarized into a two-step strategy:

1. Project original feature to higher dimensions using convolutional hyperplane arrangements.
2. Convexify using convex regularizers, such as convex variable selection models.

First, as per Step 1), we briefly recall and define a notion of hyperplane arrangements for the neural network in Definition 3 and we can rewrite the ReLU constraint using the partitioned regions by the

hyperplanes:

$$(2D(S) - I_n)Xw \geq 0 \tag{22}$$

Next, per Step 2), we define the primal problem and give its exact finite-dimensional convex formulation. Given data matrix $X \in \mathbb{R}^{n \times d}$, we consider a two-layer network $f(X; \theta) : \mathbb{R}^{n \times d} \to \mathbb{R}^n$ with $m$ neurons:

$$f(X; \theta) = \sum_{j=1}^{m} (Xw_j)_+ \alpha_j, \tag{23}$$

where $w_j \in \mathbb{R}^d$, $\alpha_j \in \mathbb{R}$ are weights for hidden and output layers respectively, and $\theta = \{w_j, \alpha_j\}_{j=1}^m$. Supplied additionally with a label vector $y \in \mathbb{R}^n$ and a regularization parameter $\beta > 0$, the primal problem of the 2-layer fully connected ReLU network is the following:

$$p^* := \min_{\{w_j, \alpha_j\}_{j=1}^m} \frac{1}{2} \|f(X; \theta) - y\|_2^2 + \frac{\beta}{2} \sum_{j=1}^{m} (\|w_j\|_2^2 + \alpha_j^2). \tag{24}$$

The above non-convex objective is transformed into an equivalent convex program. For a detailed derivation, please refer to Pilanci & Ergen (2020). We will give an overview of the key steps here. First, re-represent the above optimization problem as an equivalent $\ell_1$ penalized minimization,

$$p^* = \min_{\|w_j\|_2 \leq 1 \forall j \in [m]} \min_{\{\alpha_j\}_{j=1}^m} \frac{1}{2} \|f(X; \theta) - y\|_2^2 + \beta \sum_{j=1}^{m} |\alpha_j|. \tag{25}$$

Using strong duality, the exact *semi-infinite* convex program to objective 24 is obtained:

$$p^* = \max_{v \in \mathbb{R}^n \text{ s.t. } |v^T (Xw)_+| \leq \beta \forall w \in \mathcal{B}_2} -\frac{1}{2} \|y - v\|_2^2 + \frac{1}{2} \|y\|_2^2. \tag{26}$$

Under certain regularity conditions in Pilanci & Ergen (2020), the equivalent finite-dimensional convex formulation is the following:

$$p^* = \min_{\{u_i, u'_i\}_{i=1}^P, u_i, u'_i \in \mathbb{R}^d \forall i} \frac{1}{2} \| \sum_{i=1}^{P} D(S_i) X(u'_i - u_i) - y\|_2^2$$
$$+ \beta (\sum_{i=1}^{P} (\|u_i\|_2 + \|u'_i\|_2)) \tag{27}$$

subject to the constraints that

$$(2D(S_i) - I_n)Xu_i \geq 0, \ (2D(S_i) - I_n)Xu'_i \geq 0, \ \forall i.$$

Here, $\beta$ is a regularization parameter, and we denote $\{u_i^*, u'^*_i\}_{i=1}^P$ as the solution to objective 27.

To see that the convex program outlined in 27 is indeed an exact reformulation of the two-layer ReLU network, we run the convex solve on the spiral dataset and the quadratic regression dataset in Section 7 and compare it with standard stochastic gradient descent technique. We also note here that for the sake of brevity, we will be referring to the solving of the exact convex program with respect to the two-or-three layer ReLU networks interchangeably as "convex solve" or "final solve".

First, using the same spiral dataset of randomly selected 80 points from 100 points Spiral ($k_1 = 13, k_2 = 0.5, n_{\text{shape}} = 50$) (detailed in Appendix H.4) and $\beta = 0.001$, the convex solve achieves an objective value of 0.0008, while stochastic gradient descent levels out at an objective value greater than 0.001. The final decision boundary on the spiral using the convex solve (left) and SGD (right) is shown in Figure 5, where we have used marker "x" to indicate the train points passed into each solver and triangle marker to indicate test points. While both methods capture the spiral perfectly in this case, the advantage of the convex solve becomes more evident given more complex (e.g. in terms of shape) data. This is demonstrated in Figure 6, where we have used a spiral dataset of randomly selected 80 points from 100 points Spiral ($k_1 = 13, k_2 = 0.5, n_{\text{shape}} = 100$). Similar results hold for the regression case. The convex solve for the two-layer ReLU network returns an objective value of $4.142 \times 10^{-5}$. This aligns with the result of Figure 7 as the final prediction of the convex solves aligns perfectly with the actual quadratic function.

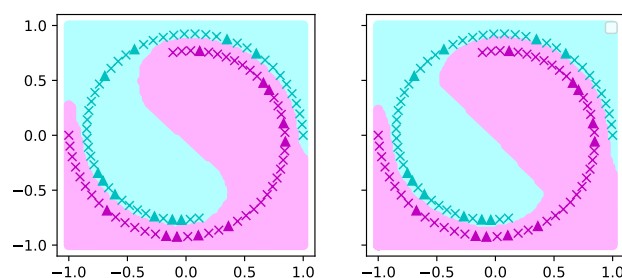

Figure 5: Decision boundary made by convex solve (left) and stochastic gradient descent (right) on the Spiral Dataset of $n_{\text{shape}} = 50$.

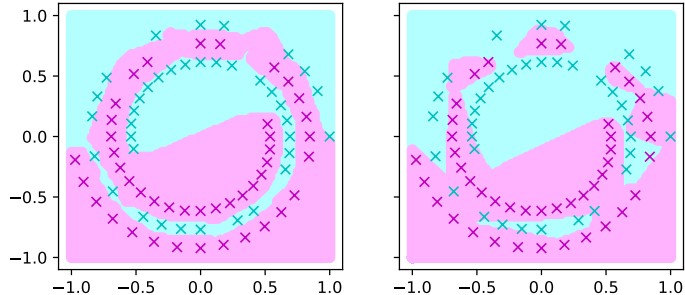

Figure 6: Decision boundary made by convex solve (left) and stochastic gradient descent (right) on the Spiral Dataset of $n_{\text{shape}} = 100$.

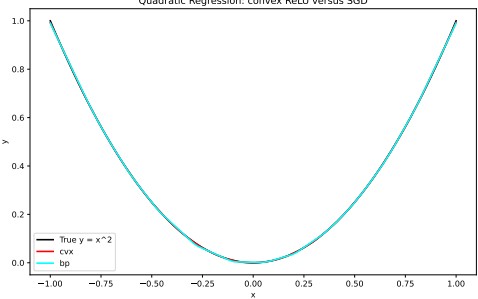

Figure 7: Prediction made by convex solve (red) and stochastic gradient descent (cyan) on the regression dataset.

**Integration of the Final Convex Solve.** Now we talk about how we can incorporate the convex program in 27 into our cutting-plane AL method to potentially aid its performance. We first provide some justifications.

As we have mentioned in Section 6, since the prediction function

$$\sum_{i=1}^{P} D(S_i)X(u'_i - u_i) := f^{\text{two-layer}}(X;\theta) = [X^1 - X^1 \dots X^P - X^P]\theta,$$

is linear in $\theta := (u'_1, u_1, \dots, u'_P, u_P)$ with $u_i, u'_i \in \mathbb{R}^d$ along with the ReLU cuts, we preserve the convexity of the parameter space after each cut. And hence, the final solve becomes well applicable. To introduce this step, we use the short-hand notations in Section 6 and also, for the sake of brevity, we denote the exact convex objective in Equation 27 as the following:

$$f^{\text{obj}}(\mathcal{D};\theta,\beta) = \min_{\theta} \frac{1}{2} \| \sum_{i=1}^{P} (D(S_i)X)_{\mathcal{D}}(u'_i - u_i) - y_{\mathcal{D}}\|_2^2$$

$$+ \beta(\sum_{i=1}^{P}(\|u_i\|_2 + \|u'_i\|_2)), \tag{28}$$

where $X_{\mathcal{D}}$ and $y_{\mathcal{D}}$ are the slices of $X$ and $y$ at indices $\mathcal{D}$.

---

**Algorithm 8** Cutting-plane AL for Binary Classification with Limited Queries using Final Solve

1: $\mathcal{T}^0 \leftarrow \mathcal{B}_2$
2: $t \leftarrow 0$
3: $\mathcal{D}_{\text{AL}} \leftarrow \mathbf{0}$
4: **repeat**
5:     $\theta^t \leftarrow \text{center}(\mathcal{T}^t)$
6:     **for** $s$ in $\{1, -1\}$ **do**
7:         $(x_{n_t}, y_{n_t}) \leftarrow \text{QUERY}(\mathcal{T}^t, \mathcal{D} \setminus \mathcal{D}_{\text{AL}}, s)$
8:         **if** $y_{n_t} \cdot f^{\text{2layer}}(x_{n_t};\theta^t) < 0$ **then**
9:             $\mathcal{D}_{\text{AL}} \leftarrow \text{ADD}(\mathcal{D}_{\text{AL}}, (x_{n_t}, y_{n_t}))$
10:          $\mathcal{T}^{t+1} \leftarrow \mathcal{T}^t \cap \{\theta : y_{n_t} \cdot f^{\text{2layer}}(x_{n_t};\theta) \geq 0, \mathcal{C}(\{n_t\}), \mathcal{C}'(\{n_t\})\}$
11:          $t \leftarrow t + 1$
12:         **end if**
13:     **end for**
14: **until** $|\mathcal{D}_{\text{AL}}| \geq n_{\text{budget}}$
15: $\theta^t \leftarrow \text{SOLVE}(f^{\text{obj}}(\mathcal{D}_{\text{AL}};\theta^t,\beta), \{C(\mathcal{D}_{\text{AL}}), C'(\mathcal{D}_{\text{AL}})\})$
16: **return** $\theta^t$

---

Algorithm 8 shows how the final convex solve is incorporated into our cutting-plane AL method via the two-layer ReLU network with limited queries. Other variations such as query synthesis (Algorithm 2), inexact cuts (Algorithm 5), as well as regression (Algorithm 6) can be adapted in the same way. In Algorithm 8, SOLVE() is a convex solve that solves the exact convex formulation in Equation 27 with all the selected training data pairs $(x_i, y_i) \mid i \in \mathcal{D}_{\text{AL}}$ from the active learning loop. In our implementation, we use CVXPY and default to solver CLARABEL (Goulart & Chen, 2024).

Upon examining Algorithm 8 and the objective function of the equivalent convex program $f^{\text{obj}}(\mathcal{D}_{\text{AL}};\theta^t,\beta)$ in 27, a key difference is the introduction of a regularization term, i.e. $\beta(\sum_{i=1}^{P}(\|u_i\|_2 + \|u'_i\|_2))$. In more complex tasks, the inclusion of the final convex solve, and thus an additional regularization, tends to aid the cutting-plane AL method in achieving faster convergence in the number of queries. This is the case, for example, for the spiral dataset in our experiment (see Appendix H.4). However, for simpler tasks, such as the quadratic regression prediction, adding regularization could slow down the convergence (see Appendix H.5). Therefore, in our experiments, we emphasize that we have used the **best** cutting-plane AL method, chosen from the version with or without the final convex solve. This is explicitly noted in the supplementary sections, namely Appendix H.4 for the spiral experiment and Appendix H.5 for the quadratic regression task. We also remark that due to the limitation in theory on the convexification of neural networks, such a convex solve is only viable up to a three-layer ReLU network.

## H.3 ACTIVE LEARNING BASELINES

In this section, we introduce the various baselines used in the spiral task and the quadratic regression task in Section 7. We also elaborate on the implementation details of each baseline discussed.

In addition to random sampling, which serves as a baseline for all other AL methods, and the linear cutting-plane AL method (Algorithm 3 for classification and Algorithm 6 for regression), which provides a baseline for our cutting-plane NN algorithm (Algorithm 4), demonstrating our extension from linear to nonlinear decision boundaries in the cutting-plane AL framework, we also survey popular AL algorithms from the `scikit-activeml` library (Kottke et al., 2021) and the `DeepAL` package (Huang, 2021).

We first introduce the baselines we implemented from the `DeepAL` package.

- *Entropy Sampling* (Settles, 2009). This technique selects samples for labeling based on the entropy of the predicted class probabilities. Recall that the entropy for a sample $x$ with predicted class probabilities $p(y|x)$ is given by:

$$H(y|x) = -\sum_c p(y = c|x) \log p(y = c|x)$$

  Higher entropy indicates greater uncertainty, making such samples good candidates for active learning.

- *Bayesian Active Learning Disagreement (BALD) with Dropout* (Gal et al., 2017a). BALD aims to choose samples that maximize the mutual information $I[y, \theta|x, \mathcal{D}_{train}]$ between predictions $y$ and model parameters $\theta$, given a sample $x$ and training data $\mathcal{D}_{train}$. Mathematically, this is expressed as:

$$I[y, \theta|x, \mathcal{D}_{train}] = H[y|x, \mathcal{D}_{train}] - \mathbb{E}_{p(\theta|\mathcal{D}_{train})}[H[y|x, \theta]]$$

  BALD with dropout extends this approach by using dropout during inference to approximate Bayesian uncertainty, allowing for efficient estimation of uncertainty in deep learning models through Monte Carlo dropout.

- *Least Confidence* (Lewis & Gale, 1994a). This strategy selects the samples where the model is least confident in its most likely prediction. For a given sample $x$, it is measured by the confidence of the predicted class $\hat{y}$, as follows:

$$\text{LC}(x) = 1 - \max_c p(y = c|x)$$

  Samples with lower confidence values are considered more uncertain and thus more informative for labeling.

Next, we introduce the baselines surveyed from `scikit-activeml`.

- *Query by Committee (QBC)* (Seung et al., 1992). The Query-by-Committee strategy uses an ensemble of estimators (a "committee") to identify samples on which there is disagreement. The committee members vote on the label of each sample, and the sample with the highest disagreement is selected for labeling. This disagreement is often quantified using measures like vote entropy:

$$H_{vote}(x) = -\sum_c \frac{v_c}{C} \log \frac{v_c}{C}$$

  where $v_c$ is the number of votes for class $c$ and $C$ is the total number of committee members. This strategy focuses on reducing uncertainty by selecting instances where committee members are in conflict.

- *Greedy Sampling on the Feature Space (GreedyX)* (Wu et al., 2019). GreedyX implements greedy sampling on the feature space, aiming to select samples that increase the diversity of the feature space the most. The method iteratively selects samples that maximize the distance between the newly selected sample and the previously chosen ones, ensuring that the selected subset of samples represents diverse regions of the feature space. This is often formulated as:

$$\max_{x_i \in \mathcal{D}} \min_{x_j \in \mathcal{Q}} \|x_i - x_j\|^2$$

  where $\mathcal{D}$ is the set of all data points and $\mathcal{Q}$ is the set of already selected query points.

- *Greedy Sampling on the Target Space (GreedyT)* (Wu et al., 2019). This query strategy initially selects samples to maximize diversity in the feature space and then shifts to maximize diversity in both the feature and target spaces. Alternatively, it can focus solely on the target space (denoted as GSy). This approach attempts to increase the representativeness of the selected samples in terms of both input features and target values. The optimization can be formulated as:

$$\max_{x_i \in \mathcal{D}} \min_{x_j \in \mathcal{Q}} \|x_i - x_j\|^2 \quad \text{and/or} \quad \max_{y_i \in \mathcal{D}} \min_{y_j \in \mathcal{Q}} \|y_i - y_j\|^2$$

  where $\mathcal{D}$ is the set of all data points and $\mathcal{Q}$ is the set of already selected query points, applied either to the feature space, the target space, or both.

- *KL Divergence Maximization (Kldiv)* (Elreedy et al., 2019). This strategy selects samples that maximize the expected Kullback-Leibler (KL) divergence between the predicted and true distributions of the target values. In this method, it is assumed that the target probabilities for different samples are independent. The KL divergence is a measure of how one probability distribution diverges from a second, reference distribution, and is given by:

$$D_{KL}(P \parallel Q) = \sum_i P(x_i) \log \frac{P(x_i)}{Q(x_i)}$$

  where $P$ is the true distribution and $Q$ is the predicted distribution. This method balances exploration and exploitation by focusing on the areas where the current model's uncertainty is greatest.

We used Query-By-Committee and Greedy Sampling on the Feature Space for both the spiral experiment and the quadratic regression experiment, and used Greedy Sampling on the Target and KL Divergence Maximization only for the regression task. As the readers may see in the descriptions provided above, both KL Divergence Maximization and Greedy Sampling on the Target Space are primarily suited for regression tasks due to the way they handle target distributions and diversity in the target space, which are more relevant in regression scenarios where the output values are continuous. Therefore, we have only included them for the regression task to offer a more robust comparison to our AL method.

---

**Algorithm 9** Deep Active Learning Baseline

---

1: $\mathcal{D}_L \leftarrow \text{SAMPLE}(\mathcal{D}, n_{\text{init}})$
2: $\mathcal{D}_U \leftarrow \mathcal{D} \setminus \mathcal{D}_L$
3: $\theta^0 \leftarrow \text{TRAIN}(\mathcal{D}_L)$
4: $t \leftarrow 0$
5: **repeat**
6: $\quad E_U \leftarrow \text{EMBEDDINGS}(\mathcal{D}_U, \theta^t)$
7: $\quad \{x_{n_t}, y_{n_t}\} \leftarrow \text{QUERY}(E_U, \mathcal{D}_U)$
8: $\quad \mathcal{D}_L \leftarrow \text{ADD}(\mathcal{D}_L, \{x_{n_t}, y_{n_t}\})$
9: $\quad \mathcal{D}_U \leftarrow \mathcal{D}_U \setminus \{x_{n_t}, y_{n_t}\}$
10: $\quad \theta^{t+1} \leftarrow \text{TRAIN}(\mathcal{D}_L)$
11: $\quad t \leftarrow t + 1$
12: **until** $|\mathcal{D}_L| \geq n_{\text{budget}}$
13: **return** $\theta^t$

---

We now discuss the implementation details for each of the aforementioned baselines. We implement random sampling and all `DeepAL` baselines using the `DeepAL` pipeline, while the `scikit-activeml` baselines are implemented within their respective framework. We note that the original `DeepAL` framework given by Huang (2021) only implements active learning for classification tasks. Thus, we modified the pipeline to allow AL methods surveyed from `DeepAL` to also be able to handle regression tasks. Since the modifications are minor, such as changing the loss criterion from cross-entropy to root mean square error loss, but not structural, we omit this distinction here and refer the readers to our submitted codes for details. Algorithm 9 outlines the general workflow of a deep active learning baseline. Here, $\mathcal{D}_L$ and $\mathcal{D}_U$ refer to the set of labeled and unlabeled training data respectively, and $n_{\text{init}}$ is the number of initial data to be randomly selected and labeled before the active learning loop.

It remains for us to discuss implementation details for `scikit-activeml` baselines. It is noteworthy that the default `scikit-activeml` active learning method for both classification and regression is implemented using its own classifiers or regressors, such as the mixture model classifiers for classification and the normal inverse Chi kernel regressor, instead of

trained using customizable deep neural nets, as in the case of `DeepAL`. For the sake of breadth in surveying popular active learning baselines, we use the Python package `skorch`, a scikit-learn compatible wrapper for PyTorch models (Viehmann et al., 2019), to integrate custom neural networks with `scikit-activeml` classifiers for the classification tasks, and used popular `scikit-activeml` active learning methods trained with its default regressor, the normal inverse Chi kernel regressor, for the regression tasks. In the classification case, we enforce the same ReLU architecture on all baselines for fairness. And in the regression case, to ensure the optimal performance of the `scikit-activeml` baselines for the robustness of comparison, we use the so-called "bagging regressor" in `scikit-activeml`, which is an ensemble method that fits multiple base regressors (in our case, 4) on random subsets of the original dataset using bootstrapping (i.e., sampling with replacement) to update the base normal inverse Chi kernel regressor. This method is known to improve query learning strategies by reducing variance and improving robustness, ensuring that the baselines perform optimally (Abe & Mamitsuka, 1998).

Further further implementation details, such as how we implement the QBC method in the classification case using `Skorch` by creating an ensemble of 5 classifiers using the exact same ReLU architecture with parameters in Table 2 but with different random state of 0-4 respectively, and additional details on the specific implementation, please refer to our submitted codes.

Finally, we would like to again emphasize that towards a fair and robust comparison, we select the best performing number of epochs (for a discussion on this, refer to Appendix H.4) and learning rate for AL baselines using deep neural networks and use enhancements such as bagging regressors for the `scikit-activeml` baselines trained on its default regressors.

## H.4 SYNTHETIC SPIRAL EXPERIMENT

In this section, we provide additional implementation details and supplementary results on the synthetic spiral binary classification experiment in Section 7.1.

**Data Generation.** We generate the two intertwined spiral used in Section 7 according to the following: the coordinates $(x_1, x_2)$ for the $i$-th data point of the spiral with label $y$ are generated as

$$x_1 = \frac{r\cos(\phi)y}{k_1} + k_2, \quad x_2 = \frac{r\sin(\phi)y}{k_1} + k_2, \quad r = k_3\left(\frac{k_4 - i}{k_4}\right), \quad \phi = k_5 i\pi,$$

where $k_1, \ldots, k_5$ are positive coefficients. In the implementation, index $i$ is normalized to lie within the range from 0 to $n_{\text{shape}} - 1$, where $n_{\text{shape}}$ controls the shape of the spiral, so that the shape is independent of the total number of data points generated. The bigger $n_{\text{shape}}$ is, the more complex the spiral (see for instance the spirals in Figure 5 versus Figure 6). In our experiment, we use the spiral dataset with $k_1 = 13, k_2 = 0.5, n_{\text{shape}} = 50$.

**AL Implementation Details.** To implement the cutting-plane AL method, we use 1000 simulations to randomly sample the number of partitions in the hyperplane arrangements (Definition 3). This determines the embedding size (or the number of neurons). Table 1 summarizes the statistics with respect to each seed. In our implementation of the deep AL baselines, we adjust the embedding size accordingly when using a different seed to ensure fair comparisons.

| Seed | 0 | 1 | 2 | 3 | 4 |
|---|---|---|---|---|---|
| Neurons | 623 | 607 | 605 | 579 | 627 |

Table 1: Number of neurons for each seed (0-4) sampled using 1000 simulations for the spiral dataset.

For all deep active learning baselines used in the Spiral case, we have set the hyper-parameters to be according to Table 2. We empirically selected learning rate 0.001 from the choices $\{0.1, 0.01, 0.001\}$ as it tends to give the best result among the three for all baselines in both the classification and regression tasks. We also choose the number of epochs to be 2000 as it also gives the best result among choices $\{20, 200, 2000\}$ and show significant improvement from both 20 and 200.

In fact, the performance of the deep AL baselines hinges much on the number of epochs in the train step (see Algorithm 9. Even when the AL method gains access to the full data, if the number of

| Epochs | Learning Rate | Train Batch Size | Test Batch Size | Momentum | Weight Decay |
|--------|---------------|------------------|-----------------|----------|--------------|
| 2000 | 0.001 | 16 | 10 | 0.9 | 0.003 |

Table 2: Hyper-parameters of deep AL baselines' training networks with the Stochastic Gradient Descent (SGD) optimizer

epochs is small, we would still not obtain satisfactory classification result. In the following, we explore the effect of number of epochs on classification accuracy for the baselines. We fix seed = 0 and use the same spiral data generation used in Section 7.1 with a 4:1 train/test split along with a budget of 20 queried points. Figure 8 to Figure 13 show the classification boundary with varying epochs = 20, 200, and 2000.

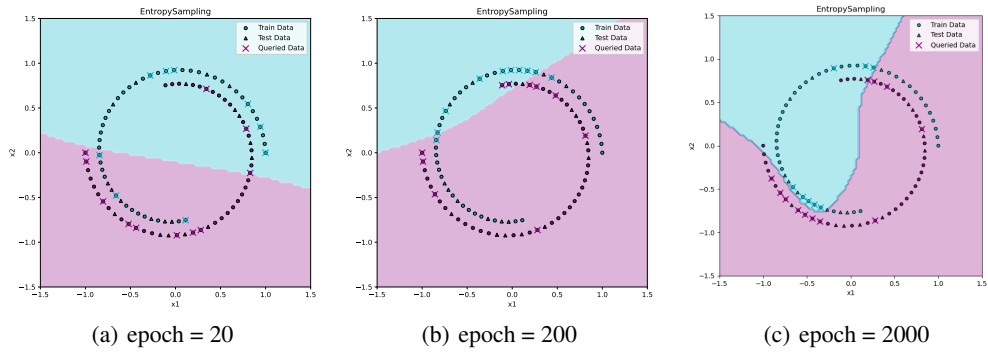

| (a) epoch = 20 | (b) epoch = 200 | (c) epoch = 2000 |

Figure 8: Entropy Sampling (20 Queries)

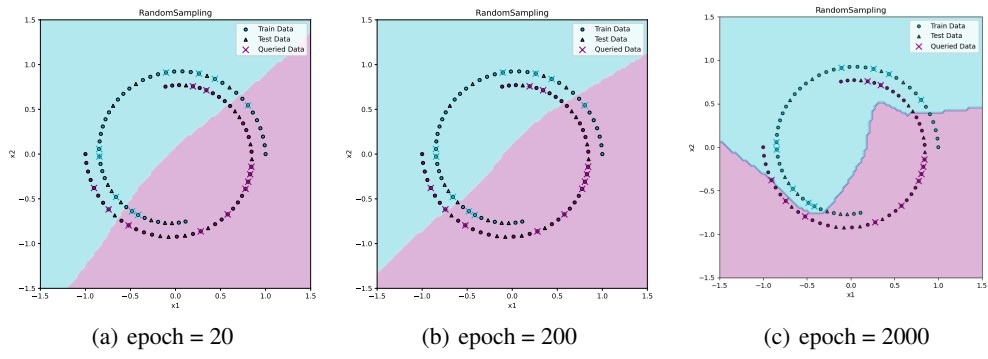

| (a) epoch = 20 | (b) epoch = 200 | (c) epoch = 2000 |

Figure 9: Random Sampling (20 Queries)

For most baselines, we observe a significant improvement in the decision boundary between 200 and 2000 epochs. To ensure a robust comparison across methods, we set the number of epochs to 2000. This mini experiment also underscores the inefficiency of gradient-based training, as the accuracy rate heavily depends on the number of training epochs, highlighting the need for a large number of iterations to achieve satisfactory results.

**Experiment Results.** We now give the deferred supplementary experiment results in Section 7.1. To start, per discussion in Appendix H.2, we would like to discuss the inclusion of regularization through the final convex solve of the equivalent convex program of the two-layer ReLU networks.

Figure 14 shows the final decision boundary with (left) and without (right) the convex solve with the same data and setup as in the spiral experiment in Section 7.1. We abbreviate "after final solve" as

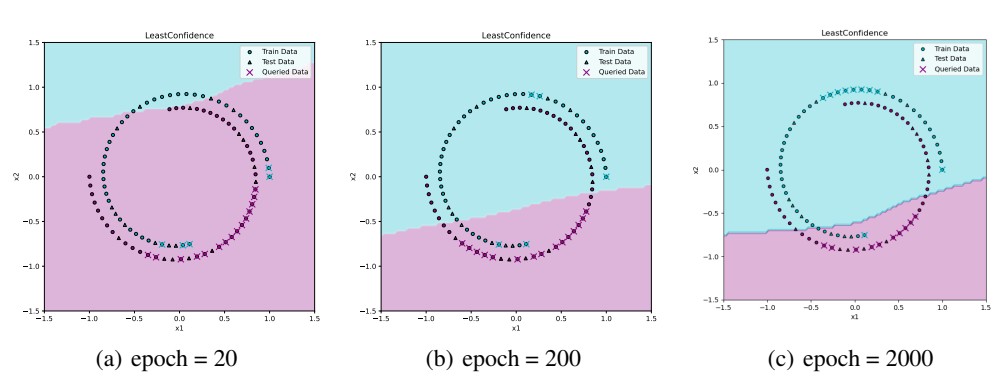

(a) epoch = 20          (b) epoch = 200          (c) epoch = 2000

Figure 10: Least Confidence (20 Queries)

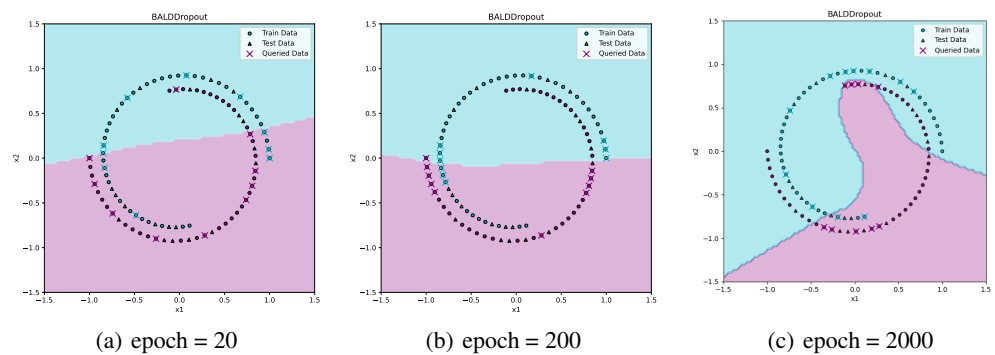

(a) epoch = 20          (b) epoch = 200          (c) epoch = 2000

Figure 11: BALD with Dropout (20 Queries)

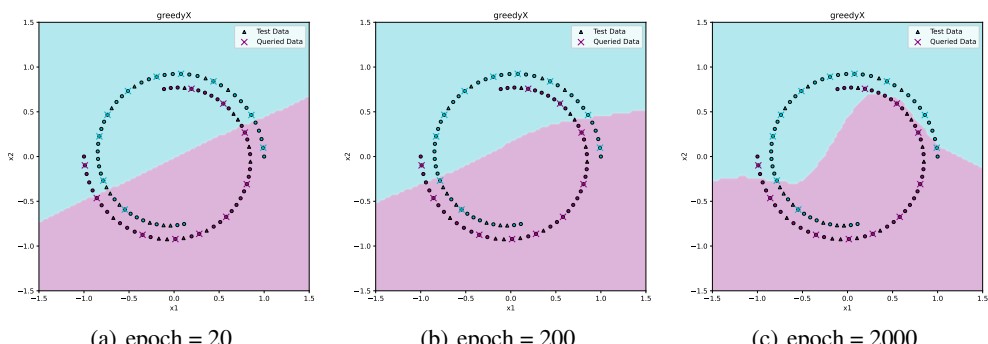

(a) epoch = 20          (b) epoch = 200          (c) epoch = 2000

Figure 12: GreedyX (20 Queries)

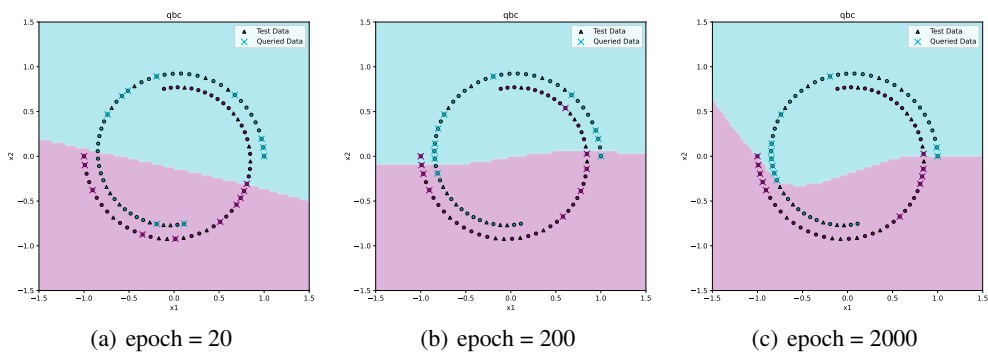

(a) epoch = 20        (b) epoch = 200        (c) epoch = 2000

Figure 13: QBC (20 Queries)

AFS and "before final solve" as BFS. It is evident that the inclusion of regularization improves the performance of our cutting-plane active learning method, increasing the accuracy rate on train/test set from 0.84/0.60 to 1.00/1.00. This is therefore the method we use for our cutting-plane AL algorithm. We will see that regularization with the final convex solve does not always speed up the convergence of our cutting-plane AL method in the number of queries. It tends to work less optimally when used on relatively simple dataset, as we will see in Appendix H.5.

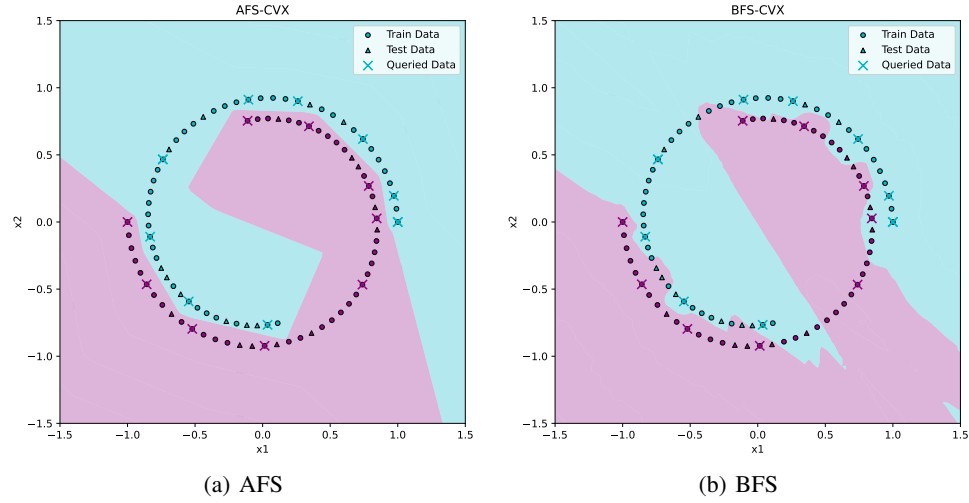

(a) AFS        (b) BFS

Figure 14: Decision boundary of cutting-plane AL via the two-layer ReLU network with (left) and without (right) final convex solve.

Table 3 presents the train and test accuracies on the binary spiral dataset. This corroborates the decision boundaries shown in Figure 2. Finally, to demonstrate the consistency of optimal performance

| Method | Train Accuracy | Test Accuracy |
|---|---|---|
| **Cutting-plane (ours)** | **1.00** | **1.00** |
| Linear Cutting-plane | 0.50 | 0.50 |
| Random Sampling | 0.73 | 0.70 |
| Entropy Sampling | 0.76 | 0.70 |
| BALD with Dropout | 0.71 | 0.65 |
| Least Confidence Sampling | 0.64 | 0.55 |
| Greedy Sampling (GreedyX) | 0.59 | 0.60 |
| Query By Committee (qbc) | 0.59 | 0.60 |

Table 3: Train and test accuracies of binary classification on the Spiral ($k_1 = 12$, $k_2 = 0.5$, $n_{shape} = 50$) dataset for cutting-plane AL via the 2-layer ReLU NN and various deep AL baselines using seed = 0.

of our proposed method and to highlight its fast convergence, we plot the mean train/test accuracy rates against the number of queries for our method and the baselines with error-bar generated by running 5 experiments on seeds 0-4. Figure 15 demonstrates the result.

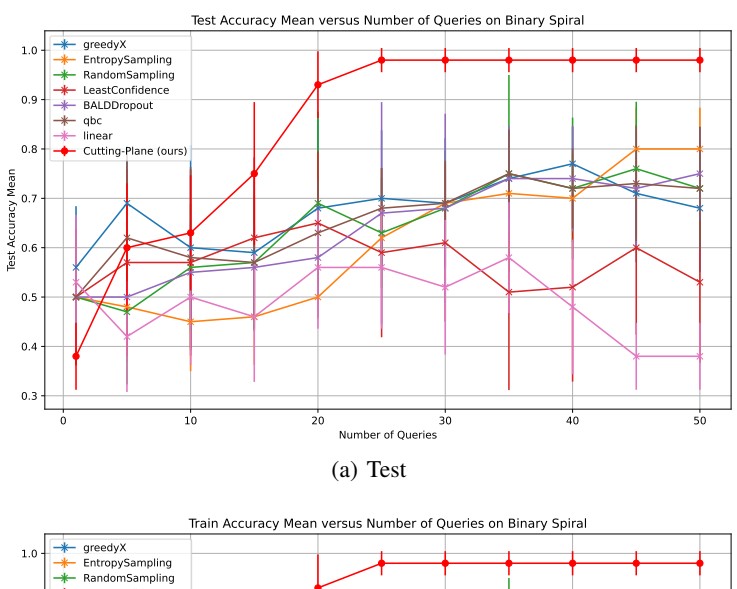

(a) Test

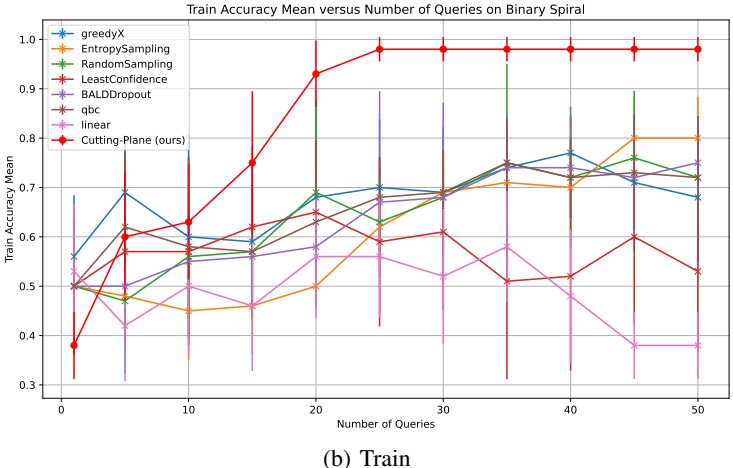

(b) Train

Figure 15: Mean test and train accuracy rate across seeds (0-4) versus the number of queries for the 2-layer cutting-plane AL and various baselines.

### H.5 QUADRATIC REGRESSION EXPERIMENT

**Data Generation and AL Implementation Details.** We generate the experiment dataset in this section simply according to the quadratic equation $y = x^2$ without adding any noise. Since the dimension of the regression dataset goes down from $d = 3$ in the spiral dataset (with the third dimension acting as bias for the ReLU networks) to $d = 2$ in the regression dataset, we increase the number of simulations to 2000 to randomly sample partitions for hyperplane arrangements to maintain an adequate embedding size. As in the spiral task, Table 4 summarizes the number of neurons sampled with respect to each seed. Once again, in our implementation of the deep AL baselines, we adjust the embedding size accordingly when using a different seed to ensure fair comparisons. As in the spiral experiment, we set the hyper-parameters for all deep active learning

| Seed | 0 | 1 | 2 | 3 | 4 |
|------|-----|-----|-----|-----|-----|
| Neurons | 160 | 160 | 157 | 159 | 160 |

Table 4: Number of neurons for each seed (0-4) sampled using 2000 simulations for the quadratic regression dataset.

baselines according to Table 2. In the quadratic regression task, we similarly observe improved

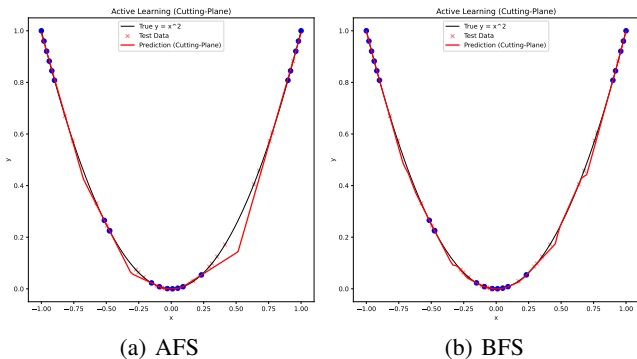

(a) AFS                    (b) BFS

Figure 16: Prediction of quadratic regression with cutting-plane AL via a two-layer ReLU network with (left) and without (right) convex solve.

performance in deep AL baselines with a higher number of training epochs and a lower learning rate of $1 \times 10^{-3}$, ensuring that the baselines provide a robust comparison. For baselines from `scikit-activeml`, we use the bagging regressor along with the default regressor to enhance their performance (see Appendix H.3). For our cutting-plane AL method (Algorithm 6) and the linear cutting-plane AL method (Algorithm 7), we choose the threshold value $\epsilon = 1 \times 10^{-3}$.

**Experiment Results.** We now present the deferred experiment results in the quadratic regression experiment. To start, we discuss the performance of our cutting-plane AL method with and without convex solve. While using the full 80 training data, the stand-alone convex solve achieves perfect prediction of quadratic regression (see Figure 7), Figure 16 shows that including the regularization in our cutting-plane AL method with a query budget of 20 leads to slightly suboptimal predictions compared to the non-regularized version. This could be because of the reduced complexity in the quadratic regression task, where the relationship between the features and the target is simple and well-behaved and the model is less likely to overfit. Therefore, the incorporation of regularization could slow down the parameter updates, leading to slower convergence.

| Method | Train RMSE | Test RMSE |
|---|---|---|
| **Cutting-plane (ours)** | **0.0111** | **0.0100** |
| Linear Cutting-plane (*infeasible*) | 0.7342 | 0.7184 |
| Random Sampling | 0.0824 | 0.0483 |
| Entropy Sampling | 0.0599 | 0.0529 |
| BALD with Dropout | 0.0568 | 0.0408 |
| Least Confidence Sampling | 0.0599 | 0.0529 |
| Greedy Sampling (GreedyX) | 0.0745 | 0.0447 |
| Query By Committee (qbc) | 0.1245 | 0.1366 |
| KL Divergence Maximization (kldiv) | 0.1356 | 0.0811 |
| Greedy Sampling Target (GreedyT) | 0.3296 | 0.3106 |

Table 5: Train and test RMSE for the quadratic regression task ($y = x^2$) using the cutting-plane AL with a 2-layer ReLU neural network and various deep AL baselines with seed = 0.

The trend plot in Figure 17 clearly illustrates the faster convergence of our cutting-plane AL method without regularization (BFS), compared to the regularized version (AFS), and significantly outpacing all other baselines. Therefore, for the quadratic regression task, we use the cutting-plane AL method without final solve (BFS). Nevertheless, it is noteworthy that both AFS and BFS cutting-plane AL method significantly outperforms all baselines and converging to the optimal prediction at a considerably faster rate.

Now we present the deferred results for the quadratic regression experiment in Section 7.1. Table 5 documents the train and test RMSE for our proposed cutting-plane AL against various deep AL baselines. In addition, Figure 18 presents the corresponding prediction made by our proposed method and the complete set of surveyed baselines.

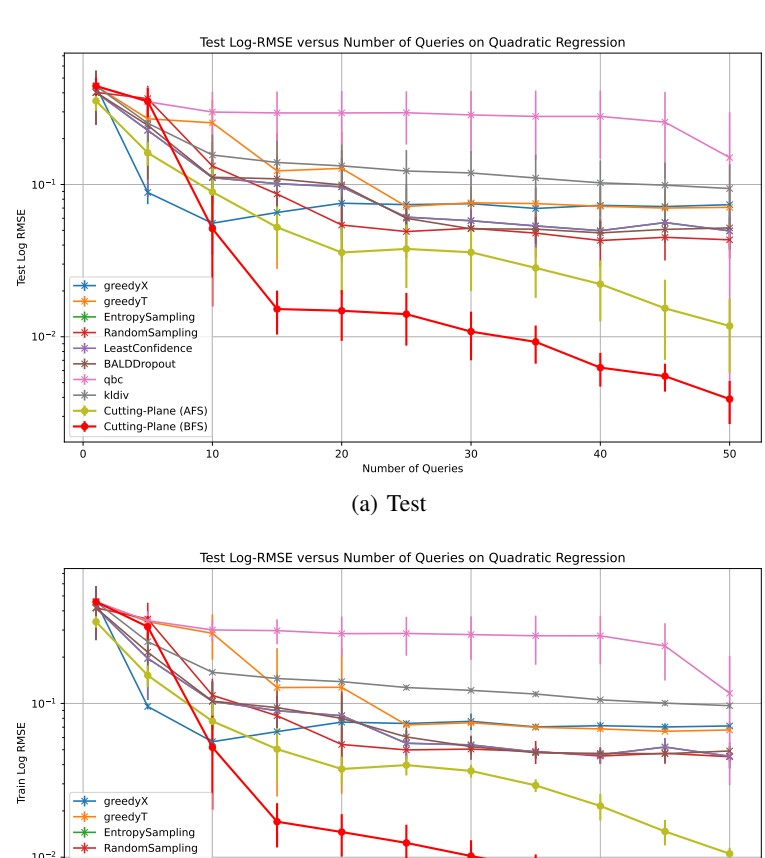

(a) Test

(b) Train

Figure 17: Logarithm of mean test and train RMSE across seeds (0-4) versus the number of queries for the 2-layer cutting-plane AL and various baselines. This is an augmented error-bar plot as right of Figure 3, with an additional distinguishment between the AFS and BFS cutting-plane AL method.

## H.6 IMDB DATA EXAMPLES

| Examples | Content | Label |
|---|---|---|
| Review 1 | "If you like original gut wrenching laughter you will like this movie. If you are young or old then you will love this movie, hell even ..." | positive |
| Review 2 | "An American Werewolf in London had some funny parts, but this one isn't so good. The computer werewolves are just awful..." | negative |
| Review 3 | ""Ardh Satya" is one of the finest film ever made in Indian Cinema. Directed by the great director Govind Nihalani..." | positive |

Table 6: Examples of IMDB movie reviews

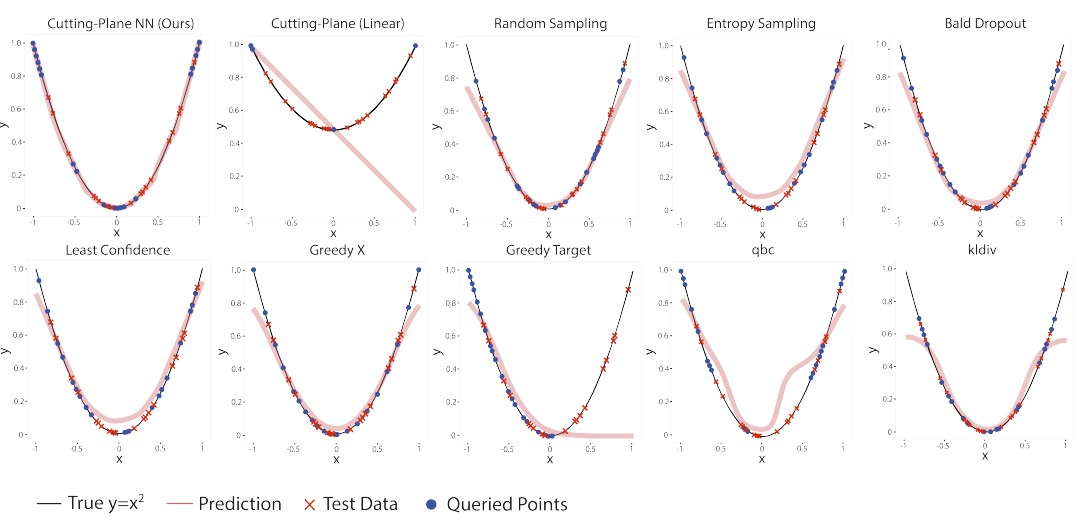

Figure 18: Predictions of various AL algorithms for quadratic regression task using the cutting-plane AL with a two-layer ReLU neural network and various deep AL baselines (complete).

