# OpenReview forum: "Active Learning of Deep Neural Networks via Gradient-Free Cutting Planes"
_ICLR.cc/2025/Conference — Submitted to ICLR 2025_

### Official Review · Reviewer_3D5d · 2024-10-23

**Soundness:** 2
**Presentation:** 2
**Contribution:** 2
**Rating:** 3
**Confidence:** 4

**Summary:**

This paper introduces a scheme for simultaneous training and query selection of data for a (ReLU) deep neural network within an active learning context. The paper map the training of a ReLU DNN to a linear program and derive cutting plane algorithms to solve this program. This further leads to an active learning selection scheme. Finally, the paper validates the framework on several synthetic datasets.

**Strengths:**

The problem of using convex solvers for training neural networks is timely and relevant, and the theoretical results seem reasonable. The paper is detailed with several algorithms for various settings and numerical experiments.

**Weaknesses:**

This paper does not include much of the rich related literature around MIP formulations of neural networks (See [1]-[5] and all of their related references). For instance, the problem of training a ReLU neural network with 1 hidden layer to global optimality via polynomial complexity was addressed in [5]. Moreover, MIP/cutting planes/bisection search formulations of active learning for deep neural networks has also been well-studied (see [6]-[8]).

Furthermore, the numerical experiments focus primarily on synthetic experiments on the spiral data set & quadratic regression, giving some analysis on IMDB. Most importantly, these experiments all rely on small 2 or 3-layer ReLU, which limits analysis of the scaling of the algorithms to larger, more realistic neural networks.


[1] Anderson, Ross, et al. "Strong mixed-integer programming formulations for trained neural networks." Mathematical Programming 183.1 (2020): 3-39.

[2] Toro Icarte, Rodrigo, et al. "Training binarized neural networks using MIP and CP." Principles and Practice of Constraint Programming: 25th International Conference, CP 2019, Stamford, CT, USA, September 30–October 4, 2019, Proceedings 25. Springer International Publishing, 2019.

[3] Fischetti, Matteo, and Jason Jo. "Deep neural networks and mixed integer linear optimization." Constraints 23.3 (2018): 296-309.

[4] Bienstock, Daniel, Gonzalo Muñoz, and Sebastian Pokutta. "Principled deep neural network training through linear programming." Discrete Optimization 49 (2023): 100795.

[5] Arora, Raman, et al. "Understanding deep neural networks with rectified linear units." arXiv preprint arXiv:1611.01491 (2016).

[6] Sener, Ozan, and Silvio Savarese. "Active learning for convolutional neural networks: A core-set approach." arXiv preprint arXiv:1708.00489 (2017).

[7] Mahmood, Rafid, Sanja Fidler, and Marc T. Law. "Low-Budget Active Learning via Wasserstein Distance: An Integer Programming Approach." International Conference on Learning Representations.

[8] Ducoffe, Mélanie. Active learning and input space analysis for deep networks. Diss. COMUE Université Côte d'Azur (2015-2019), 2018.

**Questions:**

Please see weaknesses.

---

### Official Review · Reviewer_vRgw · 2024-11-03

**Soundness:** 2
**Presentation:** 2
**Contribution:** 2
**Rating:** 3
**Confidence:** 4

**Summary:**

The paper describes an extension of the work of Louch and Ralaivola (2005) on using cutting plane methodology for active learning (AL) to Nonlinear cases using Relu networks. The work contributes some ingredients necessary to implement into Relu NNs using linear programming, provides justification for that, and also provide convergence guarantees. Some empirical validation is provided as for the methods capabilities with some standard and basic AL benchmarks.

**Strengths:**

1.	Novelty in terms of the extension to Relu networks using the linear programming
2.	Provide justification to the linear programming approach.
3.	convergence analysis (mostly relying on the work of L. and R (2005))

**Weaknesses:**

There are several major weaknesses in the level of contribution of this paper, as well as in the empirical validation of the method. There are also various issues of clarity. Altogether  that makes this paper problematic for acceptance.

1.	Effectiveness of the proposed methodology: Since the paper of L and R (2005) active learning and in particular deep active learning has made quite a progress. The active learning problem itself is meant to tackle the scalability of data labeling.

a.	 Much of Deep AL today work son massive data sets, in batch mode querying. These doesn’t not exist in this paper’s empirical validation. Instead the authors show their case on a simple 2D spiral binary classification and regressions problem and on some 50K data set of text data. By all means this is not a convincing demonstration of solving the scalability issues that AL tackles. More data sets are needed, and it would be also desired to test again R and L, on the data sets they used, even just as an ablation study (which is missing here too.)

b.	Multiclass extensions and batch mode querying are not touched here

c.	The volumetric stopping criterion is also not even touched upon, in particular its dependence on the dimension of the hypothesis space. How would a user designate it?

d.	Im missing some standard baselines for empirical comparison with active DL, such as Badge by J Ash et al, or Coreset by Sener et. al., other graph based Deep AL that uses graph built in the feature space and can address non-linearity  as such (e.g. Diffusion-based Deep Active Learning Kushnir et. al.)

2.	Contribution wise the convergence analysis is too similar to L and R (2005). It would be at least appropriate to address that Theorem 6.1 is similar to L and R.  One should also address the initial work of Tong and Kohler (JMLR, 2002) on cutting planes type methodology in AL. Also the effectiveness of the proposed method inlight of the the work done so far on deep AL is marginal in my view, and is not proved in this paper on enough examples and in terms of addressing key factors on scalability of AL

3.	Clarity:

a.	 the images, their label axes and their text fonts are reduced in size and sometime also not readable.

b.	Image 1 doesn’t correspond to the notation in the paper

c.	Statements such as line 93: “the size of parameters set reflects level of uncertainty” need to be made clear

d.	It is confusing to address theta as input in the context of DL

e.	Equation in line 163 definition is not clear: is it the space of all such diagonals matrix X? or the space of all vectors activated by the positive product?

f.	What is the benchmark accuracy for the IMDB? lines are quite flat. How does it compare with the R & L (2005) performance?

4.	The reader is redirected for many important details to the appendix, it disrupts the flow of the paper.

**Questions:**

see above

---

### Official Review · Reviewer_5g7m · 2024-11-03

**Soundness:** 3
**Presentation:** 4
**Contribution:** 3
**Rating:** 5
**Confidence:** 2

**Summary:**

This paper introduces a novel gradient-free approach for training deep neural networks using cutting-plane methods, along with an innovative active learning scheme. The authors successfully extend traditional cutting-plane methods, previously limited to linear models, to deep neural networks by reformulating ReLU network training as linear programming. The authors validate their method through experiments on both synthetic datasets and real-world sentiment classification tasks, showing superior performance compared to existing active learning baselines. The paper bridges the gap between classical optimization techniques and modern deep learning, offering a theoretically grounded alternative to gradient-based training methods.

**Strengths:**

1. The paper presents a significant theoretical contribution by providing the first convergence guarantees for deep active learning, addressing a crucial gap in the literature. The mathematical foundations are rigorous and well-explained.

2. The proposed method effectively eliminates common challenges associated with gradient-based methods, such as hyperparameter sensitivity and slow convergence, while maintaining competitive performance.

3. The method shows promising results in both classification and regression tasks when using small-scale datasets, indicating potential for specific use cases.

**Weaknesses:**

1. The method fundamentally lacks scalability due to its heavy reliance on CPU-based convex solvers and high computational complexity. The computational and memory requirements grow significantly with model size and data dimensionality, making it impractical for most real-world applications. This severely limits its practical utility in modern deep learning applications.

2. The method relies on a random sampling approach to collect ReLU activation patterns, which is not exhaustive and lacks theoretical guarantees for pattern coverage. This sampling process could miss important activation patterns, especially in high-dimensional data, potentially leading to suboptimal solutions. The paper does not provide sufficient analysis of how this limitation affects the model's performance and what patterns might be missed.

3. The method is extremely limited in its applicability, as it can only be used with neural networks using linear layers and ReLU activation functions. This restriction makes it incompatible with modern architectures such as transformers, convolutional neural networks, or networks using other activation functions, significantly limiting its practical impact in the field.

**Questions:**

1. How does the proposed method's computational complexity scale with the number of neurons and input dimensions? Could the authors provide a detailed analysis?

2. Have the authors considered extending this framework to other neural network architectures beyond ReLU networks, such as transformers or convolutional networks?

3. Could the authors elaborate on potential approaches to improve the scalability of their method, particularly in addressing the CPU-based solver limitation?

4. Is there a potential way to combine the benefits of both gradient-based and cutting-plane methods in a hybrid approach that could leverage the strengths of both paradigms?

---

### Official Review · Reviewer_Asky · 2024-11-03

**Soundness:** 3
**Presentation:** 2
**Contribution:** 2
**Rating:** 5
**Confidence:** 3

**Summary:**

This paper utilizes the cutting planes for active learning with two or multiple layers of MLP having ReLU activation functions. The cutting plane approach is free from the gradient and is usually used in the SGD, etc.  Cutting plane is the procedure to make the hyper-plane region smaller for the parameter estimation, along with the observed data points.  The cutting plane has been used in the linear binary classification. However, they show that the MLP with a ReLU in binary classification can be converted into linear programming with a constraint. From this point, they proposed a new algorithm, addressing the application of cutting plane to DNN, and active learning.

Theoretical validation based on linear programming is shown, and various experiments were done. On synthetic and IDBM datasets, the active learning of the proposed algorithm is dominant over Random, Entropy, Least confidence, BALD dropout, QBC, etc.

**Strengths:**

The proposed algorithm is interesting, and it has a solid theoretical background, especially for the separable cases. The algorithm is simple, and intuitive in the application aspects.

**Weaknesses:**

$\bullet$ The first issue is the applicability in real-world active learning with DNN. Many studies of active learning with DNN consider the CNN layer, ResNet architecture, transformer, etc in the image classification, natural langauge processing, and medical image analsys.  The linear layers with ReLU activation is limited in the use of active learning.

$\bullet$ The second issue is the range of experiments in the compared alg., and datasets. The datasets are too small compared to other active learning studies (usually considering 5~10 datasets such as tiny image-net or CIFAR100). In the aspects of algorithms, recent algorithms such as BADGE, BAIT, LDM-based, etc. are not considered. Especially, the BADGE is based on the gradient. The comparison of BADGE and cutting-plane can reveal the advantages or disadvantages of two algorithms in the overall performance and fast convergence in the early steps w/wo gradients.  It can highlight the proposed algorithm compared to gradient-based query algorithm.

$\bullet$ Furthermore, the effect of the depth of layers can be an interesting issue. The advanced issues are not thoroughly studied.

**Questions:**

$\bullet$
I think that the one of advantages of cutting-plane can be computational costs. Can you provide any discussion about this issue? In detail,  can you provide empirical runtime comparisons between the proposed method and gradient-based approaches on the datasets tested. This would help quantify any computational advantages.

$\bullet$
Is there any issue in the selection of querying (unlabeled) data points when multiple data points are targeted on a query?  How the method handles batch-mode active learning, where multiple samples are selected in each iteration, and whether there are any theoretical guarantees in that scenario.

---

### Official Review · Reviewer_4dmm · 2024-11-04

**Soundness:** 2
**Presentation:** 3
**Contribution:** 2
**Rating:** 3
**Confidence:** 3

**Summary:**

The paper proposes a novel neural network training method by using the cutting plane algorithm. Extending from previous cutting plane methods to multi-layer ReLU network, the authors decompose the problem of data fitting to a set of linear programming problems. The authors also introduce a new active learning strategy that queries the most confident samples, hoping it finds misclassified samples to shrink the parameter set significantly.

**Strengths:**

1. The paper introduces a novel perspective of neural network training by leveraging cutting plane algorithms.
2. The algorithm also has significant theoretical guarantees in its convergence and effectiveness.
3. Experiments are conducted to showcase the effectiveness of the cutting plane approach

**Weaknesses:**

I have several concerns, especially towards the scalability of the approach.
1. My biggest fear is the method will not work well with larger scale neural networks. This is because neural networks, with moderate capacity, are universal approximaters. For example, with a small ResNet-18, the model can fit arbitrary set of labels for the CIFAR-10 dataset with perfect accuracy. This suggests that in most vision/language applications involving large scale neural networks, the activation pattern $D$ of the last layer can simply be the set of $diag(\{0, 1\}^n)$. In such a case, cutting algorithms are not effective in training neural networks, nor doing active learning.
2. The active learning experiments all have extremely small budgets. This is potentially because the active learning strategy in this paper is somewhat counterintuitive. The algorithm queries the most confident samples, and hope for the model to be misclassifying them. However, for any neural network trained with a moderate amount of labeled samples and with gradient-based approach, the most confident samples are almost always correctly classified. I suspect the reason existing algorithms (with gradient based training) underperform is because, under limited budget, it is very easy for neural networks to make mistakes. However, in almost all deep learning applications, the model is only effective with high accuracies, i.e., the model needs to learn about uncertain samples. I also wonder if the gain from the cutting plane active learning algorithm only comes from the gradient-free training, instead of the active learning strategy.
3. The paper primarily relies on the property of ReLU activation, which is not generalizable to modern transformer architectures, where training the model is definitely not a linear program.

**Questions:**

See weakness section.

---

### Meta-Review · Area_Chair_U1Zb · 2024-12-08

**Metareview:**

a:

This paper proposes a gradient-free cutting-plane training method for ReLU networks, by utilzing the insight that MLP with a ReLU in binary classification can be converted into linear programming. The paper also proposes an active learning scheme. Convergence guarantee of the training method is shown as a Theorem. Experiments with synthetic and real-world datasets show the effectiveness of the proposed approach.

b:

A novel method is proposed which is a gradient-free cutting-plane training method for neural nets. Experiments show the effectiveness of the proposed method. Theoretical convergence analysis is shown. Intuition/motivation is clear.

c:

Reviewers discussed weaknesses in terms of effectiveness/practicality of the proposed methods (such as the problem/task and architecture), lack of significance of the convergence analysis, and scalability issues.

d:

Although the paper works on an interesting problem and it seems that substantial effort went into preparing the paper, a rebuttal to the reviews were not provided during the rebuttal period. This left concerns of the reviewers unaddressed.

**Additional Comments On Reviewer Discussion:**

Rebuttal was not provided.

---

### Decision · Program_Chairs · 2025-01-22

Reject